# THE LOGICAL EXPRESSIVENESS OF TOPOLOGICAL NEURAL NETWORKS

**Amirreza Akbari**
Aalto University
amirreza.akbari@aalto.fi

**Amauri H. Souza**
Federal Institute of Ceará, $2\delta$ AI
amauriholanda@ifce.edu.br

**Vikas Garg**
Aalto University and YaiYai Ltd
vgarg@csail.mit.edu

## ABSTRACT

Graph neural networks (GNNs) are the standard for learning on graphs, yet they have limited expressive power, often expressed in terms of the Weisfeiler-Leman (WL) hierarchy or within the framework of first-order logic. In this context, topological neural networks (TNNs) have recently emerged as a promising alternative for graph representation learning. By incorporating higher-order relational structures into message-passing schemes, TNNs offer higher representational power than traditional GNNs. However, a fundamental question remains open: *what is the logical expressiveness of TNNs?* Answering this allows us to characterize precisely which binary classifiers TNNs can represent. In this paper, we address this question by analyzing isomorphism tests derived from the underlying mechanisms of general TNNs. We introduce and investigate the power of higher-order variants of WL-based tests for combinatorial complexes, called $k$-CCWL test. In addition, we introduce the topological counting logic ($\text{TC}_k$), an extension of standard counting logic featuring a novel pairwise counting quantifier $\exists^N (x_i, x_j)\, \varphi(x_i, x_j)$, which explicitly quantifies pairs $(x_i, x_j)$ satisfying property $\varphi$. We rigorously prove the exact equivalence: k-CCWL $\equiv$ $\text{TC}_{k+2}$ $\equiv$ Topological $(k+2)$-pebble game. These results establish a logical expressiveness theory for TNNs.

## 1 INTRODUCTION

Graph neural networks (GNNs) (Scarselli et al., 2009; Bruna et al., 2014; Gilmer et al., 2017; Hamilton et al., 2017) have become the *de facto* paradigm for predictive tasks on graph-structured data, driving progress in several applications such as drug repurposing (Stokes et al., 2020), simulation of physical systems (Sanchez-Gonzalez et al., 2020), algorithmic reasoning (Jurss et al., 2023), and recommender systems (Ying et al., 2018). Nonetheless, most GNNs are at most as powerful as the 1-Weisfeiler–Lehman (1-WL) test (or the color refinement algorithm) (Weisfeiler & Leman, 1968) in distinguishing non-isomorphic graphs (Xu et al., 2019; Morris et al., 2019). Consequently, GNNs fail to capture fundamental structural information (e.g., cycles or connected components) and decide basic graph properties (Loukas, 2020; Garg et al., 2020; Chen et al., 2020). Importantly, this limitation has also been examined through the lens of first-order logic, which provides a precise characterization of the node-level binary classifiers that GNNs can express (Barceló et al., 2020).

Naturally, this limited expressivity has propelled the development of more expressive GNNs (Sato et al., 2021; Wang et al., 2022; Horn et al., 2022; Verma et al., 2024; Immonen et al., 2023). Among these, topological neural networks (TNNs) (Hajij et al., 2023; Bodnar et al., 2021a; Papillon et al., 2025) have recently emerged as the new frontier for relational learning (Papamarkou et al., 2024). Akin to GNNs, typical TNNs employ message-passing layers where each element of the input (e.g., nodes or cells) updates its representation (or features) based on those of its topological neighbors. These TNNs operate on the so-called (uniform) combinatorial complexes (Hajij et al., 2023): a flexible relational notion that accommodates both non-hierarchical domains (e.g., hypergraphs)

and hierarchical ones (e.g., cell complexes). Unraveling the representation capabilities of TNNs is imperative to understanding their strengths and pitfalls, and designing better, more nuanced models that are theoretically well-grounded. However, although the logical expressivity of GNNs, and even their higher-order variants (Morris et al., 2019; 2024), is well understood within the framework of first-order logic (Barceló et al., 2020; Geerts & Reutter, 2022), the logical characterization of TNNs remains largely uncharted.

To address this gap, this paper investigates the logical expressiveness of a general framework for topological neural networks and their higher-order variants, here denoted as $k$-TNNs. As a first step, we introduce extensions of the classical color refinement algorithm tailored to TNNs: the *combinatorial complex WL* (CCWL) test and its higher-order counterpart, $k$-CCWL. These refinements are not just abstract gadgets: they capture the expressive power of a broad class of existing TNN architectures. In particular, 1-CCWL subsumes the expressivity of most message-passing TNNs on cell complexes, including CW Networks (e.g., Cell Isomorphism Networks, CIN) (Bodnar et al., 2021a), message-passing simplicial networks (Bodnar et al., 2021b), and higher-order GNNs such as UniGCN and UniGIN (Huang & Yang, 2021), among many others summarized in (Papillon et al., 2024). More generally, any model with invariant update functions under the $k$-CCWL refinement cannot distinguish ACCs that $k$-CCWL cannot separate. For $k = 1$, this reduces to standard message passing on combinatorial complexes; for $k > 1$, the architecture operates on $k$-tuples of cells and updates labels via the same mechanism as $k$-CCWL. Notably, $(k$-$)$CCWL can be used as a standard tool for analyzing the expressivity of TNNs, and it leads to a strict hierarchy as $k$ grows.

To formally analyze the expressive power of TNNs, we introduce a new fragment of first-order logic, the *topological counting logic* ($\mathsf{TC}_k$), which is inspired by classical counting logic (Ebbinghaus & Flum, 1999; Otto, 2017; Libkin, 2004; Immerman, 1998). The standard counting quantifier can only count single-variable instantiations; however, $\mathsf{TC}_k$ extends this by introducing a *pairwise counting quantifier* $\exists^N(x_i, x_j)\,\varphi(x_i, x_j)$ that allows reasoning about pairs of cells in a topological structure. Intuitively, unlike in graphs, where edges directly connect two vertices, many relations between cells are *mediated* by other cells (e.g., two edges are adjacent because they share a vertex, or they both lie in the same face). Many interesting patterns also come from how a cell's boundary (its faces) and coboundary (its cofaces) interact, which is naturally described by counting pairs of cells that meet in some shared neighbor. This extension is motivated by the setting of TNNs aggregation: When updating a cell's label through its upper and lower neighborhoods, the update necessarily involves information from intermediary face and co-face cells, which means that TNNs aggregate *pairs* of relational signals rather than single neighbors. The pairwise quantifier thus provides the correct logical abstraction for capturing these higher-order interactions.

In parallel, we define the *topological $k$-pebble game*, which generalizes the classical pebble games of Immerman (1982); Immerman & Lander (1990) and the earlier work of Ehrenfeucht (1961); Fraïssé (1954), and serves as the game-theoretic counterpart of $\mathsf{TC}_k$. Unlike the classical version, however, the topological game must be designed to capture the semantics of the new *pairwise counting quantifier*: Instead of only reasoning about single-variable placements, the game needs to account for pairs of pebbled elements and their topological relations. The main challenge lies in extending the rules so that duplicator–spoiler interactions reflect the ability of $\mathsf{TC}_k$ to distinguish structures based on the number of cell pairs satisfying a given property; therefore, we need to precisely align the expressive power of the game with the expressive power of the logic.

These tools create a unified analysis of $k$-TNNs from three sides: algorithmic ($k$-CCWL), logical ($\mathsf{TC}_k$), and game-theoretic (the topological pebble game). Our main contribution shows that these three characterizations are *equivalent*, in direct analogy to the classical $k$-WL / counting logic / pebble games correspondence (Cai et al., 1992), which provides the first logic–game–algorithm triad for $k$-TNNs and a precise description of how their expressive power scales with $k$.

## 1.1 CONTRIBUTIONS

In this paper, we extend the classic triad "$k$-WL $\leftrightarrow$ $C_{k+1}$ counting logic $\leftrightarrow$ $(k+1)$-pebble game from graphs to attributed combinatorial complexes (ACCs), and use it to analyze the expressive power of topological neural networks (TNNs):

Section 3 *Algorithm.* We introduce $k$-CCWL, a WL-style refinement on ACCs that respects boundary, coboundary, lower and upper adjacencies. On uniform ACCs, a simple broadcast-

anchor gadget gives an *identical-vs-disjoint* global color property (Proposition 3.1), and for every $k \in \mathbb{N}$, we provide non-isomorphic ACCs that are distinguished by $k$-CCWL but not by $(k-1)$-CCWL (Theorem 3.2). We also show $k$-CCWL has the same expressive power as $k$-WL in terms of distinguishing non-isomorphic 1-dimensional ACCs (Theorem D.2).

Section 4 *Logic.* We define a new topological counting logic $\mathsf{TC}_k$ with *a pairwise counting quantifier*, $\exists^N(x_i, x_j)\, \varphi(x_i, x_j)$, which can capture the aggregation patterns of TNNs. We show that $\mathsf{TC}_{k+2}$ *refines* $k$-CCWL in distinguishing non-isomorphic ACCs (Theorem 4.1).

Section 5 *Game.* We introduce a topological $k$-pebble game, whose winning strategies for Player II characterize $\mathsf{TC}_k$-equivalence (Theorem 5.1). As the final piece of the triad, we then show that $k$-CCWL-indistinguishability implies a winning strategy for Player II (Theorem 5.2). All together, we have our main result[1](Corollary 5.3):

$$k\text{-CCWL} \; \equiv \; \mathsf{TC}_{k+2} \; \equiv \; \text{topological } (k+2)\text{-pebble game},$$

so the hierarchy in $k$ is strict.

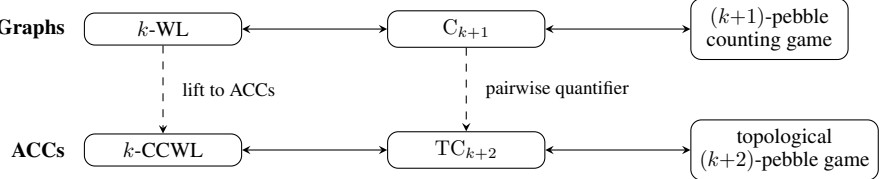

## 2 BACKGROUND

This section overviews combinatorial complexes, message-passing topological neural networks, and the logical expressivity of GNNs.

**Combinatorial Complexes.** Hajij et al. (2023) introduced combinatorial complexes as a relational structure to accommodate both hierarchical topological domains (e.g., simplicial complexes) and non-hierarchical ones (e.g., hypergraphs). Formally, a combinatorial complex (CC) is a triple $(S, \mathcal{X}, \mathrm{rk})$ consisting of a set of vertices $S$, a set of cells $\mathcal{X} \subseteq \mathcal{P}(S) \setminus \{\emptyset\}$, and a rank function $\mathrm{rk} : \mathcal{X} \to \mathbb{Z}_{\geq 0}$ such that $\forall s \in S,\ \{s\} \in \mathcal{X}$; and $\forall x, y \in \mathcal{X},\ x \subseteq y \Rightarrow \mathrm{rk}(x) \leq \mathrm{rk}(y)$. The set of $k$-cells of a combinatorial complex, denoted $\mathcal{X}(k)$, consists of all cells with rank exactly $k$.

We can exploit rank functions to specify local neighbors for each cell. Analogously to Bodnar et al. (2021a), we define four neighborhood structures:

- Boundary adjacency: $\mathcal{N}_B(x) = \{y \subset x : \mathrm{rk}(y) = \mathrm{rk}(x) - 1\}$
- Co-boundary adjacency: $\mathcal{N}_C(x) = \{y \supset x : \mathrm{rk}(y) = \mathrm{rk}(x) + 1\}$
- Upper adjacency: $\mathcal{N}_\uparrow(x) = \{y : \exists \delta \text{ such that } x \subset \delta, y \subset \delta \text{ and } \mathrm{rk}(\delta) - 1 = \mathrm{rk}(y) = \mathrm{rk}(x)\}$
- Lower adjacency: $\mathcal{N}_\downarrow(x) = \{y : \exists \delta \text{ such that } \delta \subset x, \delta \subset y \text{ and } \mathrm{rk}(\delta) + 1 = \mathrm{rk}(y) = \mathrm{rk}(x)\}$

We are interested in attributed combinatorial complexes. An attributed combinatorial complex (ACC) is a tuple $(S, \mathcal{X}, \mathrm{rk}, c)$ where $c : \mathcal{X} \to \Sigma$ assigns a label $c(x)$ from a domain $\Sigma$ to each cell $x$. The *order* of an ACC is $\max\{\mathrm{rk}(x) : x \in \mathcal{X}\}$. Moreover, we call an ACC *uniform* if for every cell $x$ of rank $r > 0$ there exists a finite sequence of cells $x = x_0, x_1, \ldots, x_t$ such that $x_t$ is a cell of rank 0 and, for every $0 \leq i < t$, there exists $\mathcal{N}_i \in \{\mathcal{N}_B, \mathcal{N}_C, \mathcal{N}_\downarrow, \mathcal{N}_\uparrow\}$ with

$$x_i \in \mathcal{N}_i(x_{i+1}).$$

In other words, from any cell, there is a finite sequence of cells to a cell of rank 0 by following the neighborhood structures. Further, we assume binary features, i.e., $\Sigma = \{0, 1\}^\ell$ for some constant $\ell$.

**CC-Isomorphism.** Papillon et al. (2025) introduced the notion of CC-isomorphism. Let $\Gamma_1 = (V_1, C_1, \mathrm{rk}_1, c_1)$ and $\Gamma_2 = (V_2, C_2, \mathrm{rk}_2, c_2)$ be combinatorial complexes equipped with neighborhood structures $\mathcal{N}_1$ and $\mathcal{N}_2$, respectively. A *CC-isomorphism induced by* $(\mathcal{N}_1, \mathcal{N}_2)$ is a bijective function $f : C_1 \to C_2$ such that

---

[1]In case $k = 1$, we show 1-CCWL has the same expressive power as guarded fragment of $\mathsf{TC}_3$.

1. For all $\sigma, \tau \in C_1$, if $\tau \in \mathcal{N}_1(\sigma)$ then $f(\tau) \in \mathcal{N}_2(f(\sigma))$,
2. The inverse $f^{-1}$ is also a CC-homomorphism induced by $(\mathcal{N}_2, \mathcal{N}_1)$, and
3. If $c_1(\sigma) = c_1(\tau)$, then $c_2(f(\sigma)) = c_2(f(\tau))$.

**Topological neural networks (TNNs)** (Bodnar et al., 2021a; Papillon et al., 2025; 2024) employ message-passing layers to obtain cell-level representations. In particular, let $\mathcal{N}_{\text{all}} = \{\mathcal{N}_B, \mathcal{N}_C, \mathcal{N}_\downarrow, \mathcal{N}_\uparrow\}$ be the set of neighborhood structures. In its general form, starting from $h_x^0 = c(x)$ for all cells $x$, message-passing TNNs recursively compute:

$$h_x^{\ell+1} = \varphi\left(h_x^\ell, \bigotimes_{\mathcal{N} \in \mathcal{N}_{\text{all}}} \text{Agg}_\ell\left(\mathcal{M}_{\mathcal{N},x}^\ell\right)\right) \tag{1}$$

where $h_x^\ell$ is the embedding of $x$ at layer $\ell$, $\bigotimes$ and $\text{Agg}_\ell$ are inter- and intra-neighborhood aggregation functions, respectively, $\varphi$ is an update function (e.g., MLP), and $\mathcal{M}_{\mathcal{N},x}^\ell$ denotes a multiset of messages. Let us define $\mathcal{N}_C(x,y) = \mathcal{N}_C(x) \cap \mathcal{N}_C(y)$ and $\mathcal{N}_B(x,y) = \mathcal{N}_B(x) \cap \mathcal{N}_B(y)$. In the literature, two main variants of message computation have been introduced:

$$\mathcal{M}_{\mathcal{N},x}^\ell = \begin{cases} \{\!\!\{\phi_{\ell,\mathcal{N}}(h_y^\ell, h_x^\ell, h_\delta^\ell) : y \in \mathcal{N}(x), \delta \in \mathcal{N}_B(x,y)\}\!\!\}, & \text{if } \mathcal{N} = \mathcal{N}_\downarrow \\ \{\!\!\{\phi_{\ell,\mathcal{N}}(h_y^\ell, h_x^\ell, h_\delta^\ell) : y \in \mathcal{N}(x), \delta \in \mathcal{N}_C(x,y)\}\!\!\}, & \text{if } \mathcal{N} = \mathcal{N}_\uparrow \\ \{\!\!\{\phi_{\ell,\mathcal{N}}(h_y^\ell, h_x^\ell) : y \in \mathcal{N}(x)\}\!\!\}, & \text{otherwise.} \end{cases} \tag{2}$$

Similar to the equivalence between 1-WL and GNNs, we can define corresponding isomorphism tests for TNNs. The key idea consists of adopting injective (or hash) functions for the maps $\phi, \bigotimes, \text{Agg}, \varphi$ in Equations (1) and (2).

**The $k$-WL Algorithm.** The $k$-dimensional Weisfeiler–Leman algorithm ($k$-WL) extends the classical color refinement to $k$-tuples of vertices (Grohe & Otto, 2015; Grohe, 2017; Cai et al., 1992). Let $G = (V, E, c)$ be a vertex-colored graph. For $\mathbf{v} = (v_1, \ldots, v_k) \in V^k$, the *atomic type* $\text{atp}_{G,k}(\mathbf{v})$ encodes the equality pattern among coordinates, the adjacency relations $(v_i, v_j) \in E$, and the initial colors $c(v_i)$. Initialization is given by

$$\text{wl}_k^{(0)}(\mathbf{v}) := \text{atp}_{G,k}(\mathbf{v}). \tag{3}$$

At iteration $t + 1$, the label of $\mathbf{v}$ is refined using substitutions by all $u \in V$:

$$\text{wl}_k^{(t+1)}(\mathbf{v}) := \left(\text{wl}_k^{(t)}(\mathbf{v}), \left\{\!\!\left\{\left(\text{atp}_{k+1}(\mathbf{v}, u), \text{wl}_k^{(t)}(\mathbf{v}[u/1]), \ldots, \text{wl}_k^{(t)}(\mathbf{v}[u/k])\right) : u \in V\right\}\!\!\right\}\right) \tag{4}$$

Here $\mathbf{v}[u/i]$ denotes the $k$-tuple obtained by replacing $v_i$ with $u$. The process stabilizes after finitely many steps, yielding the stable coloring $\text{wl}_k^{(\infty)}$. Two graphs $G, H$ are distinguished by $k$-WL if their multisets of stable colors differ.

Morris et al. (2019) use a slightly different method as $k$-WL. To avoid confusion, we follow the method used in Cai et al. (1992) as $k$-WL, and recall the latter method as oblivious $k$-WL, or $k$-OWL.

**Logical expressivity and counting logic.** The $k$-variable counting logic $\mathsf{C}_k$ is the fragment of first-order logic restricted to variables $\{x_1, \ldots, x_k\}$ and extended with counting quantifiers. For any $N \in \mathbb{N}$, formulas are generated by

$$\varphi := (x_i = x_j) \mid P_s(x_i) \mid E(x_i, x_j) \mid \neg\psi \mid (\psi_1 \wedge \psi_2) \mid \exists^N x_i\, \psi,$$

where $P_s$ (for $s \in [\ell]$) are unary predicates encoding node attributes, and $E$ is the adjacency relation.

Semantics is defined over a vertex-colored graph $G = (V_G, E_G, c_G)$. Let $C_s = \{v \in V_G : (c_G(v))_s = 1\}$ be the set of vertices that their $s$-th bit of attribute is 1. For a (partial) valuation $\mu : \{x_1, \ldots, x_k\} \rightharpoonup V_G$, the satisfaction relation $G, \mu \models \phi$ is defined as follows:

- $G, \mu \models P_s(x_i)$ iff $\mu(x_i) \in C_s$.
- $G, \mu \models E(x_i, x_j)$ iff $(\mu(x_i), \mu(x_j)) \in E_G$.
- $G, \mu \models (x_i = x_j)$ iff $\mu(x_i) = \mu(x_j)$.
- $G, \mu \models \neg\psi$ iff $G, \mu \not\models \psi$.
- $G, \mu \models (\psi_1 \wedge \psi_2)$ iff $G, \mu \models \psi_1$ and $G, \mu \models \psi_2$.
- $G, \mu \models \exists^N x_i\, \psi$ iff there exist at least $N$ distinct elements $c \in V_G$ such that $G, \mu[x_i \mapsto c] \models \psi$.

A key result of Cai et al. (1992) shows that $C_{k+1}$ has exactly the same distinguishing power as the $k$-dimensional Weisfeiler–Leman algorithm ($k$-WL). Moreover, two graphs can be distinguished by $k$-WL if and only if they can be separated by some $C_{k+1}$ formulas. Therefore, the iterative refinement of $k$-WL captures precisely those properties which can be defined with $k+1$ variables and counting quantifiers. This explains why increasing $k$ strictly increases expressivity, and why the power of GNNs and their higher-order variants can be analyzed by counting logic. **Pebble Game.** The $C_k$-pebble game, introduced by Immerman & Lander (1990), gives a game-theoretic view of $C_k$. The game is played between two players on two graphs $A$ and $B$ with $k$ pairs of pebbles $(a_1, b_1), \ldots, (a_k, b_k)$. At any moment, the current pebble placements define a partial mapping from vertices of $A$ to vertices of $B$. In each round, Player I selects a subset $X \subseteq B$; Player II must respond with a subset $Y \subseteq A$ of equal size. Then, Player I chooses an element of $Y$ to place a pebble $a_i$ there, and Player II must place the matching $b_i$ on an element of $X$. Player I wins immediately if the partial mapping defined by the pebbles is not a partial isomorphism (i.e., fails to preserve adjacency relations or equality).

Player II has a winning strategy for $q$ rounds if she can always maintain a partial isomorphism for that many rounds. In this case, $A$ and $B$ satisfy the same $C_k$-sentences of quantifier depth at most $q$. Indeed, the following are equivalent: $A$ and $B$ agree on all formulas in $C_k$; $A$ and $B$ are indistinguishable by $k$-WL; and Player II has a winning strategy in the $k$-pebble game on $A$ and $B$.

## 3 HIGHER-ORDER TOPOLOGICAL ISOMORPHISM TESTS

This section introduces higher-order isomorphism tests associated with TNNs (using injective aggregate and update functions). In other words, we naturally extend the tuple-based procedure of $k$-WL to combinatorial complexes.

The construction mirrors $k$-WL on graphs: we start from an *atomic type* for $k$-tuples that encodes equality, ranks, colors, and the four topological adjacencies; we then iteratively refine by "probing" all possible substitutions and aggregating the resulting colors. The key novelty is the *double shift* sequence, which simultaneously tracks two substitutions per position; this is exactly what will force a $(k+2)$-variable requirement on the logic side in Section 4.

---

**$k$-CCWL isomorphism test ($k \geq 2$)**

Let $\Gamma = (S, \mathcal{X}, \mathrm{rk}, \mathrm{c})$ be an ACC, and let $\mathbf{x} = (x_1, \ldots, x_k) \in \mathcal{X}^k$ be a $k$-tuple of cells. The *rank sequence* of $\mathbf{x}$ is defined as $\mathrm{rank}(\mathbf{x}) = (\mathrm{rk}(x_1), \ldots, \mathrm{rk}(x_k))$. The *atomic type* of a $k$-tuple $\mathbf{x}$, denoted $\mathrm{atp}_k(\mathbf{x})$, encodes structural and attribute-level information about the tuple. It consists of the rank sequence of $\mathbf{x}$, the initial colors $\mathrm{c}(x_i)$ for $i = 1, \ldots, k$, and five binary vectors that describe all pairwise relations among components of $\mathbf{x}$: equality, boundary adjacency, coboundary adjacency, lower adjacency, and upper adjacency. Each relation vector shows if the corresponding pair of cells satisfies the relation.

1. **Initialization**: For each $\mathbf{x} \in \mathcal{X}^k$, we define the initial label: $\chi_k^{(0)}(\mathbf{x}) = \mathrm{atp}_k(\mathbf{x})$.

2. **Update**: For $t \geq 1$, the label of each $\mathbf{x} \in \mathcal{X}^k$ is updated as: $\chi_k^{(t)}(\mathbf{x}) = \mathrm{HASH}\big(\chi_k^{(t-1)}(\mathbf{x}), \ \mathcal{D}_k^{(t)}(\mathbf{x})\big)$, where $\mathcal{D}_k^{(t)}(\mathbf{x})$ is a multiset of refinement contexts, which is defined as follows:

$$\mathcal{D}_k^{(t)}(\mathbf{x}) = \left\{\!\!\left\{ \Big( \mathrm{atp}_{k+2}(\mathbf{x}\alpha\beta), \ \Delta_k^{(t-1)}(\mathbf{x}; \alpha, \beta) \Big) \ \Big| \ \alpha, \beta \in \mathcal{X} \right\}\!\!\right\}.$$

Here, $\Delta_k^{(t-1)}(\mathbf{x}; \alpha, \beta)$ denotes the *double shift sequence*:

$$\Delta_k^{(t-1)}(\mathbf{x}; \alpha, \beta) = \left\langle \begin{pmatrix} \chi_k^{(t-1)}(\mathbf{x}[\alpha/i]) \\ \chi_k^{(t-1)}(\mathbf{x}[\beta/i]) \end{pmatrix} \right\rangle_{i=1}^k,$$

3. **Termination**: The process continues until convergence: $\chi_k^{(t+1)}(\mathbf{x}) = \chi_k^{(t)}(\mathbf{x})$ for all $\mathbf{x} \in \mathcal{X}^k$. The final stable labeling is denoted $\chi_k^{(\infty)}(\mathbf{x})$.

---

In both CCWL and $k$-CCWL, after each refinement step, we aggregate the tuple labels to form a global label for the complex. For an ACC $\Gamma = (S, \mathcal{X}, \mathrm{rk}, \mathrm{c})$ and iteration $t$, this is defined as

$$\chi_{k,\Gamma}^{(t)} = \left\{\!\!\left\{ \chi_k^{(t)}(\mathbf{x}) \mid \mathbf{x} \in \mathcal{X}^k \right\}\!\!\right\}.$$

Given two ACCs $A = (S_A, \mathcal{X}_A, \mathrm{rk}_A, \mathrm{c}_A)$ and $B = (S_B, \mathcal{X}_B, \mathrm{rk}_B, \mathrm{c}_B)$, the $k$-CCWL procedure *distinguishes* $A$ and $B$ if their stable global signatures differ, i.e.,

$$\chi_{k,A}^{(\infty)} \neq \chi_{k,B}^{(\infty)}.$$

**Complexity.** Let $n = |\mathcal{X}|$. Since we work with finite ACCs, the refinement process induced by $k$-CCWL stabilizes after finitely many iterations (at most $n^k - 1$), so the stable coloring is well-defined (see Lemma B.1). Additionally, we can use $O(n^{\max(2,k)})$ space to encode all adjacencies, ranks, and attributes. Updating the colors of all $k$-tuples can be done in $O(n^{k+2})$. Thus, the time complexity of $k$-CCWL is $O(n^{2k+2})$.

For isomorphism testing, we augment each complex by introducing a fresh 0-cell connected to every existing 0-cell via new 1-cells, all initialized with a distinct new color. We then run the $k$-CCWL refinement on these augmented complexes and compare their stable global signatures. Intuitively, the fresh 0-cell acts as a *broadcast anchor*: any local color difference between the two complexes is propagated through this anchor to all other cells during the refinement process. Importantly, adding the broadcast-anchor cells does not change whether the original complexes are CC-isomorphic; we only mark the anchor and its incident 1-cells as new elements by choosing distinct colors from the rest of the complex. As a consequence, once stabilization is reached, the global signatures of the two augmented complexes cannot partially overlap: *they are either exactly the same or completely disjoint*.

**Proposition 3.1** (Identical-vs-disjoint). *Let $A = (S_A, \mathcal{X}_A, \mathrm{rk}_A, \mathrm{c}_A)$ and $B = (S_B, \mathcal{X}_B, \mathrm{rk}_B, \mathrm{c}_B)$ be two uniform ACCs, and let $k \geq 1$. Suppose the $k$-CCWL algorithm is applied to both $A$ and $B$. Then exactly one of the following holds:*

$$\chi_{k,A}^{(\infty)} = \chi_{k,B}^{(\infty)} \quad or \quad \chi_{k,A}^{(\infty)} \cap \chi_{k,B}^{(\infty)} = \emptyset.$$

The proof of Proposition 3.1 is provided in Section B of Appendix. Our next result shows that $k$-CCWL is not less expressive than $(k-1)$-CCWL.

**Theorem 3.2.** *For any $k \in \mathbb{N}$, there exist pairs of non-isomorphic ACCs that are distinguishable by $k$-CCWL but not by $(k-1)$-CCWL.*

The proof of Theorem 3.2 is provided in Section D of Appendix.

## 4 TOPOLOGICAL COUNTING LOGIC

In what follows, we consider ACCs of the form $\Gamma = (S, \mathcal{X}, \mathrm{rk}, \mathrm{c})$, where $\mathrm{c} : \mathcal{X} \to \{0,1\}^\ell$ assigns binary attribute vectors and $\mathrm{rk} : \mathcal{X} \to \{0, 1, \ldots, \rho\}$ assigns a rank to each cell. We define the $k$-variable fragment $\mathsf{TC}_k$ of first-order logic extended with a pairwise counting quantifier. Formulas $\phi \in \mathsf{TC}_k$ are built over the variable set $\{x_1, \ldots, x_k\}$ and follow the grammar:

$$\phi := (x_i = x_j) \mid R_r(x_i) \mid P_s(x_i) \mid E^{\mathcal{N}}(x_i, x_j) \mid \psi_1 \wedge \psi_2 \mid \neg\psi \mid \exists^N(x_i, x_j)\,\psi, \tag{5}$$

where $i, j \in \{1, \ldots, k\}$ are distinct. The symbols $R_r$ for $r \in \{0, 1, \ldots, \rho\}$ and $P_s$ for $s \in \{1, \ldots, \ell\}$ are unary predicates encoding rank and bitwise color attributes, respectively. The binary predicate $E^{\mathcal{N}}$, for $\mathcal{N} \in \{\mathcal{N}_B, \mathcal{N}_C, \mathcal{N}_\uparrow, \mathcal{N}_\downarrow\}$, captures boundary, coboundary, lower, and upper adjacency relations.

The set of free variables in a formula $\phi \in \mathsf{TC}_k$, denoted $\mathrm{free}(\phi)$, determines the variables that are not bound by quantifiers and is defined inductively as follows:

1. $\mathrm{free}(x_i = x_j) = \mathrm{free}(E^{\mathcal{N}}(x_i, x_j)) = \{x_i, x_j\}$

2. $\mathrm{free}(R_r(x_i)) = \mathrm{free}(P_s(x_i)) = \{x_i\}$       3. $\mathrm{free}(\phi_1 \wedge \phi_2) = \mathrm{free}(\phi_1) \cup \mathrm{free}(\phi_2)$

4. $\mathrm{free}(\neg\phi) = \mathrm{free}(\phi)$       5. $\mathrm{free}(\exists^N(x_i, x_j)\,\phi) = \mathrm{free}(\phi) \setminus \{x_i, x_j\}$

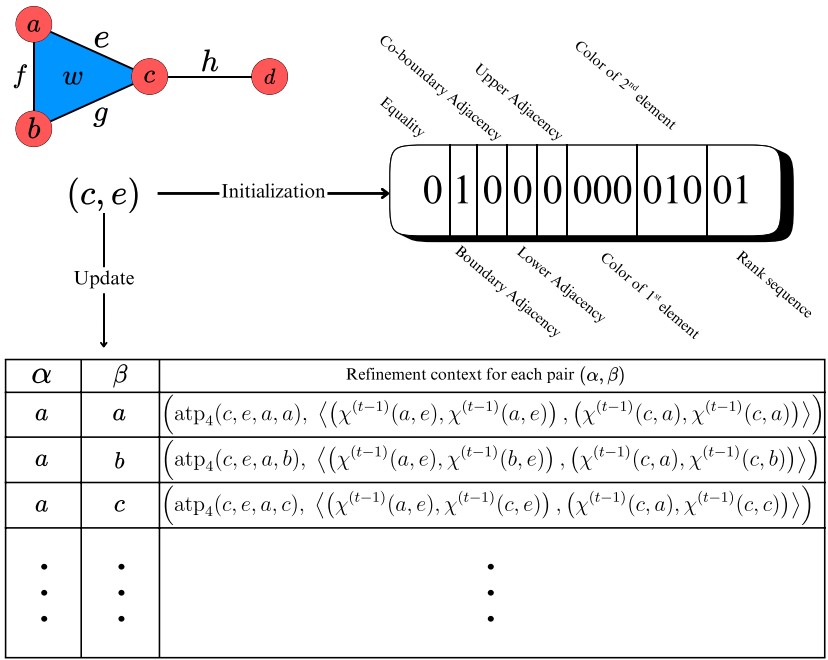

Figure 1: Example of one refinement round in the **2-CCWL** on an ACC where all vertices share the same color, all edges share the same color, and there is a single 2-cell (the triangular face). The chosen tuple $(c, e)$, consisting of a vertex $c$ and an edge $e$, is first assigned its *atomic type*, which includes pairwise equality, adjacency relations, colors, and the rank sequence. During the *update step*, all substitutions $(\alpha, \beta)$ of cells into the tuple are considered, and for each pair the *refinement context* is formed from the extended atomic type and the double shift sequence. The resulting collection of contexts is then hashed with the initial label to produce the updated label at round $t$.

A partial function $\mu : \{x_1, \ldots, x_k\} \rightharpoonup \mathcal{X}$ has a domain $\mathrm{dom}_\mu \subseteq \{x_1, \ldots, x_k\}$ on which each $x \in \mathrm{dom}_\mu$ maps to exactly one cell of $\Gamma$. The semantics of formulas in $\mathsf{TC}_k$ is defined in terms of interpretations relative to a given ACC $\Gamma$ and a partial valuation $\mu$. This interpretation determines whether a formula $\phi$ is satisfied by a given ACC $\Gamma$ and a partial valuation $\mu$ with $\mathrm{free}(\phi) \subseteq \mathrm{dom}_\mu$, denoted as $\Gamma, \mu \models \phi$. To be more precise, we define:

- $\Gamma, \mu \models R_r(x_i)$ if and only if $\mathrm{rk}(\mu(x_i)) = r$.
- $\Gamma, \mu \models P_s(x_i)$ if and only if the $s$-th bit of $\mathrm{c}(\mu(x_i))$ is 1.
- $\Gamma, \mu \models E^{\mathcal{N}}(x_i, x_j)$ if and only if $\mu(x_j) \in \mathcal{N}(\mu(x_i))$.
- $\Gamma, \mu \models (x_i = x_j)$ if and only if $\mu(x_i) = \mu(x_j)$.
- $\Gamma, \mu \models (\phi \wedge \psi)$ if and only if $\Gamma, \mu \models \phi$ and $\Gamma, \mu \models \psi$.
- $\Gamma, \mu \models \neg\phi$ if and only if $\Gamma, \mu \not\models \phi$.
- $\Gamma, \mu \models \exists^N (x_i, x_j)\, \psi$ if and only if $\left| \{(c_1, c_2) \in \mathcal{X}^2 \mid \Gamma, \mu[x_i \mapsto c_1, x_j \mapsto c_2] \models \psi\} \right| \geq N$.

In the last case, $\mu[x_i \mapsto c_1, x_j \mapsto c_2]$ denotes the partial valuation obtained by extending $\mu$ to include the assignments $x_i$ to $c_1$ and $x_j$ to $c_2$. Moreover, when $\mathrm{free}(\phi) = \emptyset$, we simply write $\Gamma \models \phi$ instead of $\Gamma, \mu \models \phi$, since the choice of valuation $\mu$ is then immaterial.

**Limitations.** Note that $\mathsf{TC}_k$ is still a finite-variable counting logic and it inherits the usual locality restrictions of $\mathsf{C}_k$. Many global graph/topological properties need unbounded $k$ and cannot be expressed by $\mathsf{TC}_k$ for any fixed $k$, e.g.: (1) classic global graph properties that $k$-WL cannot capture for fixed $k$, such as connectivity, existence of Hamiltonian path or a perfect matching. (2) topological invariants that depend on the overall pattern of "holes" and voids across the entire complex, where the difference between two complexes may only appear in very large cycles that cannot be detected from any fixed-radius local neighborhood. Note that Eitan et al. (2025) shows that higher-order

message passing on CCs has "blindspots" (e.g., for diameter, orientability, planarity, homology). Indeed, our hierarchy is complementary: many such blindspots correspond to properties requiring unbounded quantification and thus lie outside $\mathsf{TC}_{k+2}$.

The $k$-CCWL procedure refines tuple colors step by step, and each update depends on how many pairs of cells $(\alpha, \beta)$ yield a given atomic type for the extended tuple and produce a specific color pattern. On the logic side, $\mathsf{TC}_{k+2}$ can describe exactly this situation: with $k$ variables for the tuple and two extra variables, it can state, "there are at least $N$ such pairs $(\alpha, \beta)$ making the extension satisfy property $\varphi$". Thus, each refinement step in $k$-CCWL can be seen as the logical counterpart of adding one more layer of quantifiers in $\mathsf{TC}_{k+2}$. Formally, we have

**Theorem 4.1.** *Let $A = (S_A, \mathcal{X}_A, \mathrm{rk}_A, \mathrm{c}_A)$ and $B = (S_B, \mathcal{X}_B, \mathrm{rk}_B, \mathrm{c}_B)$ be two ACCs, and let $u_A \in \mathcal{X}_A^k$, $u_B \in \mathcal{X}_B^k$ be $k$-tuples of cells from $A$ and $B$, respectively. If for every formula $\phi \in \mathsf{TC}_{k+2}$, we have*

$$A, u_A \models \phi \iff B, u_B \models \phi,$$

*then the $k$-CCWL colorings agree:*

$$\chi_{k,A}^{(\infty)}(u_A) = \chi_{k,B}^{(\infty)}(u_B).$$

**Proof sketch.** The proof proceeds by induction on the number of refinement steps $t$. For the base case ($t = 0$), we use the agreement on all quantifier-free formulas to show that the initial atomic types, and therefore, the initial colors, are the same. For the inductive step, we assume the claim holds up to step $n - 1$. If a difference appears at step $n$, we have two cases: (1) It happened at step $n - 1$ (contradicting the induction hypothesis) or (2) comes from the neighborhood aggregation. In the latter case, we can construct a formula of quantifier depth $n$ in $\mathsf{TC}_{k+2}$ that separates the two structures, which again contradicts the assumption. Therefore, the colors must agree at every step. The full proof of Theorem 4.1 is provided in Section C.4 of Appendix.

## 5 TOPOLOGICAL PEBBLE GAME

Let $A = (S_A, \mathcal{X}_A, \mathrm{rk}_A, \mathrm{c}_A)$ and $B = (S_B, \mathcal{X}_B, \mathrm{rk}_B, \mathrm{c}_B)$ be two ACCs, and let $u_A, u_B : \{x_1, \ldots, x_k\} \rightharpoonup \mathcal{X}_A, \mathcal{X}_B$ be partial valuations such that $\mathrm{dom}(u_A) = \mathrm{dom}(u_B)$. We say that $(A, u_A)$ and $(B, u_B)$ are $k$-*dimensionally structurally similar*, which is written as $(A, u_A) \sim_k (B, u_B)$, if for every pair of variables $x_i, x_j \in \mathrm{dom}(u_A)$, the following conditions hold:

1. $u_A(x_i) = u_A(x_j) \iff u_B(x_i) = u_B(x_j)$

2. $\mathrm{rk}_A(u_A(x_i)) = \mathrm{rk}_B(u_B(x_i))$

3. $\mathrm{c}_A(u_A(x_i)) = \mathrm{c}_B(u_B(x_i))$

4. $u_A(x_i) \in \mathcal{N}(A, u_A(x_j)) \iff u_B(x_i) \in \mathcal{N}(B, u_B(x_j))$ for all $\mathcal{N} \in \{\mathcal{N}_B, \mathcal{N}_C, \mathcal{N}_\uparrow, \mathcal{N}_\downarrow\}$

Note that the above conditions show a local CC-isomorphism between the subcomplexes of $A$ and $B$, which is induced by $u_A$ and $u_B$. In particular, $(A, u_A) \sim_k (B, u_B)$ confirms that the mapping $u_A(x_i) \mapsto u_B(x_i)$ preserves equalities between cells, their ranks, their labels, and all of their neighborhood relations. In other words, it satisfies exactly the requirements of a labeled CC-isomorphism restricted to the cells in the domain of the (partial) valuations. Therefore, we can take $k$-dimensional structural similarity as a local form of CC-isomorphism, defined by partial valuations. Here, $\mathcal{N}(A, x)$ denotes the neighborhood of cell $x$ in the combinatorial complex $A$, making explicit which complex the neighborhood is taken with respect to.

The topological pebble game in Box 1 below is designed to match exactly the semantics of $\mathsf{TC}_k$. Intuitively, the pairwise counting quantifier in the logic is reflected by the game rule where Player I selects a set of cell pairs in one complex and Player II must answer with a set of the same size in the other. To keep winning, Player II must choose the responses so that all ranks, colors, and neighborhood relations remain consistent between the two structures. This means that if Player II can survive for $t$ rounds without losing, then the two structures cannot be distinguished by any $\mathsf{TC}_k$ formula with quantifier depth at most $t$.

> **Box 1: Topological Pebble Game** $\mathsf{TC}_k$
>
> The topological pebble game is a two-player combinatorial game played between Player I (the *Spoiler*) and Player II (the *Duplicator*) on two ACCs $A = (S_A, \mathcal{X}_A, \mathrm{rk}_A, \mathrm{c}_A)$ and $B = (S_B, \mathcal{X}_B, \mathrm{rk}_B, \mathrm{c}_B)$. The state of the game is described by a pair of partial valuations $\mu_A, \mu_B : \{x_1, \ldots, x_k\} \rightharpoonup \mathcal{X}_A, \mathcal{X}_B$, which assign up to $k$ variables to cells in $A$ and $B$, respectively. Each round of the game proceeds as follows:
>
> **Step 1:** Player I chooses one of the complexes (say $A$) and selects a finite set of ordered pairs $P_A \subseteq \mathcal{X}_A^2$.
>
> **Step 2:** Player II must respond with a set $P_B \subseteq \mathcal{X}_B^2$ such that $|P_B| = |P_A|$.
>
> **Step 3:** Player I chooses a pair $(\sigma_B, \tau_B) \in P_B$ and updates the valuation $\mu_B$ by placing pebbles: $\mu_B[x_i \mapsto \sigma_B, \ x_j \mapsto \tau_B]$ for some distinct $i, j$.
>
> **Step 4:** Player II must choose a pair $(\sigma_A, \tau_A) \in \mathcal{X}_A^2$ and update $\mu_A[x_i \mapsto \sigma_A, \ x_j \mapsto \tau_A]$.
>
> After $r$ rounds, we obtain valuations $(\mu_A^{(r)}, \mu_B^{(r)})$, which show all pebble placements so far. Player II wins the $r$-round game if they preserve a $k$-dimensional structural similarity,
>
> $$(A, \mu_A^{(i)}) \sim_k (B, \mu_B^{(i)}), \quad \text{for all } i \leq r.$$
>
> Player II has a *winning strategy* if this similarity is preserved in every $r$-round game. Player I wins if Player II fails to preserve structural similarity in any round.

**Theorem 5.1.** *Let $A = (S_A, \mathcal{X}_A, \mathrm{rk}_A, \mathrm{c}_A)$ and $B = (S_B, \mathcal{X}_B, \mathrm{rk}_B, \mathrm{c}_B)$ be two ACCs, and let $u_A \in \mathcal{X}_A^k$, $u_B \in \mathcal{X}_B^k$ be $k$-tuples of cells from $A$ and $B$, respectively. If Player II has a winning strategy in the topological $(k)$-pebble game on $(A)$ and $(B)$ with initial configuration $(u_A, u_B)$, then for every formula $\phi \in \mathsf{TC}_k$,*

$$A, u_A \models \phi \iff B, u_B \models \phi.$$

**Proof sketch.** The proof proceeds by induction on the number of moves $t$ in the game. For the base case ($t = 0$), we prove that if Player II has a winning strategy, then $A, u_A$ and $B, u_B$ agree on all quantifier-free formulas. For the inductive step, we assume the claim holds up to $t = n - 1$. At $t = n$, the only important case we need to handle is when the formula is of the form $\exists^N (x, y)\phi(x, y)$. If Player I selects a set of candidate pairs in one structure, then Player II must respond with a set of equal size in the other. If no such response were possible, Player II would fail to maintain a winning strategy, which contradicts the assumption. Therefore, the result also holds for $t = n$. The full proof of Theorem 5.1 is provided in Section C.5 of Appendix.

The next step is to connect $k$-CCWL with the topological pebble game. Intuitively, if two tuples receive the same color at every step of $k$-CCWL, then Player II can always mirror Player I's moves in the $(k+2)$-pebble game without losing. To give more intuition, one can recall that in the classical Weisfeiler–Leman correspondence, where $k$-WL matches with the $(k+1)$-pebble game: the extra pebble serves as a cursor to resample one coordinate, while the other $k$ pebbles keep the rest of the tuple fixed. In our setting, however, each refinement step inspects *pairs* of extensions $(\alpha, \beta)$ via the double-shift sequence, and the logic quantifies over such pairs. As a result, we need two spare pebbles for $(\alpha, \beta)$, which leads to $(k+2)$ pebbles rather than $(k+1)$ pebbles. This correspondence is formalized in the following theorem.

**Theorem 5.2.** *Let $A = (S_A, \mathcal{X}_A, \mathrm{rk}_A, \mathrm{c}_A)$ and $B = (S_B, \mathcal{X}_B, \mathrm{rk}_B, \mathrm{c}_B)$ be two ACCs, and let $u_A \in \mathcal{X}_A^k$, $u_B \in \mathcal{X}_B^k$ be $k$-tuples of cells from $A$ and $B$, respectively. If the $k$-CCWL colorings of $u_A$ and $u_B$ agree, i.e., $\chi_{k,A}^{(\infty)}(u_A) = \chi_{k,B}^{(\infty)}(u_B)$, then Player II has a winning strategy in the topological $(k+2)$-pebble game on $(A)$ and $(B)$ with initial configuration $(u_A, u_B)$.*

**Proof sketch.** The proof shows that if $\chi_{k,A}^{(t)}(u_A) = \chi_{k,B}^{(t)}(u_B)$, then Player II has a winning strategy for the $t$-move $\mathsf{TC}_{k+2}$ game. The proof is by induction on $t$. For the base case ($t = 0$), equality of $k$-CCWL colors implies identical atomic types for $u_A$ and $u_B$, which satisfies the conditions needed for Player II to win the 0-move game. In the inductive step, assume the result holds for all $t < n$. For $t = n$, Player I's strongest move is to modify an unmodified pair $(x_{k+2}, x_{k+1})$. Player II responds by selecting matching valuations in the other structure, preserving atomic types and color

shifts. By the induction hypothesis, Player II can then continue to maintain a winning strategy for the remainder of the game. The full proof of Theorem 5.2 is provided in Section C.5 of Appendix.

Combining Theorems 4.1, 5.1 and 5.2 yields our three-way equivalence: equality of stable $k$-CCWL colors, Duplicator's winning strategy with $k+2$ pebbles, and agreement on all $\mathsf{TC}_{k+2}$ formulas.

**Corollary 5.3.** *Let $A = (S_A, \mathcal{X}_A, \mathrm{rk}_A, \mathrm{c}_A)$ and $B = (S_B, \mathcal{X}_B, \mathrm{rk}_B, \mathrm{c}_B)$ be two ACCs, and let $u_A \in \mathcal{X}_A^k$, $u_B \in \mathcal{X}_B^k$ be $k$-tuples of cells. Then the following statements are equivalent:*

1. *$\chi_{k,A}^{(\infty)}(u_A) = \chi_{k,B}^{(\infty)}(u_B)$*

2. *Player II has a winning strategy in the topological $(k+2)$-pebble game on $(A, u_A)$ vs. $(B, u_B)$*

3. *For every formula $\phi \in \mathsf{TC}_{k+2}$, we have: $A, u_A \models \phi \iff B, u_B \models \phi$*

Restricting the above corollary to the case of uniform ACCs, and using Proposition 3.1, we have:

**Theorem 5.4.** *The $k$-CCWL procedure and $\mathsf{TC}_{k+2}$ are equivalent in expressive power for distinguishing non-isomorphic uniform ACCs.*

The proof of Theorem 5.4 is provided in Section C.5

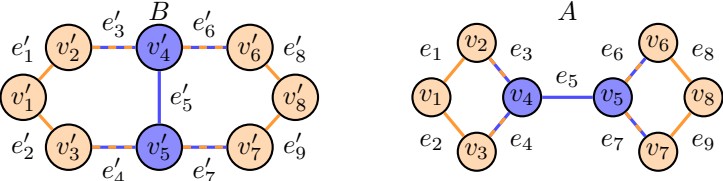

Figure 2: Two non-isomorphic simplicial complexes. If all vertices and edges are given the same initial color, then running SWL or 1-CCWL gives the final coloring shown above.

*Example* 5.5. Figure 2 is an example of two non-isomorphic simplicial complexes from Verma et al. (2024). In this example, we want to test both logical and game point of view for $k = 2$ to show that (1) $\mathsf{TC}_4$ can distinguish these two ACCs, and (2) Player I has a winning strategy in $\mathsf{TC}_4$ game.

- Logic: Consider formula $\phi \in \mathsf{TC}_4$ as follows:
$$\phi = \exists^{16}(x_1, x_2)\exists^1(x_3, x_4) \, \mathrm{CYCLE}(x_1, x_2, x_3, x_4)$$
where $\mathrm{CYCLE}(x_1, x_2, x_3, x_4) = E^{\mathcal{N}_\downarrow}(x_1, x_2) \wedge E^{\mathcal{N}_\downarrow}(x_2, x_3) \wedge E^{\mathcal{N}_\downarrow}(x_3, x_4) \wedge E^{\mathcal{N}_\downarrow}(x_4, x_1) \wedge \neg E^{\mathcal{N}_\downarrow}(x_1, x_3) \wedge \neg E^{\mathcal{N}_\downarrow}(x_2, x_4) \wedge \neg(x_1 = x_3) \wedge \neg(x_2 = x_4)$. In the ACC $A$, we have two copies of $C_4$ connected by an edge, and each $C_4$ gives us eight 4-tuples $u_A = (u_1, u_2, u_3, u_4)$ that $u_i$ are connected consecutively. Therefore, $A \models \phi$, however, we do not have such tuples in ACC $B$, therefore, $B \not\models \phi$.

- Game: Player I first chooses, in each copy of $C_4$ in $A$, all 16 pairs of adjacent edges as $P_A$. For any reply $P_B$, there is a pair not containing $e'_5$; Player I picks such a pair and modifies $u_B$. W.l.o.g., assume now Player II modifies the valuation $u_A$ such that $u_A(x_1) = e_1$ and $u_A(x_2) = e_2$. In the next round, Player I sets $P_A = \{(e_3, e_4)\}$. Whatever Player II chooses, they lose, since in $A$, the edges $e_1, e_2, e_3$ and $e_4$ are consecutively in each other's lower neighborhood.

## 6 CONCLUSION

We introduced the $k$-dimensional combinatorial complex Weisfeiler–Leman test ($k$-CCWL) together with a Topological Counting Logic ($\mathsf{TC}_k$) equipped with a pairwise counting quantifier and matched them with a topological $(k+2)$-pebble game. Together, these yield the first logic–game–algorithm triad for higher-order message passing on combinatorial complexes. Our main equivalence is

$$k\text{-CCWL} \equiv \mathsf{TC}_{k+2} \equiv \text{Topological } (k+2)\text{-pebble game.} \tag{6}$$

It follows that $k$-CCWL and $\mathsf{TC}_{k+2}$ have identical separation power, which *strictly* increases with $k$. This pins down exactly which binary classifiers higher-order architectures can realize and, via the pairwise-counting view, explains when moving to higher dimensions yields provable gains over standard message passing on graphs.

ETHICS STATEMENT

This work advances our understanding of the foundations of topological neural networks by establishing their logical expressivity. While it might pave way for design of more powerful topological models, we do not foresee any specific ethical concerns stemming from this work.

ACKNOWLEDGMENTS

This work was supported by the Quantum Doctoral Programme (QDOC) of the Finnish Quantum Flagship. VG also acknowledges Saab-WASP, Research Council of Finland, and the Jane and Aatos Erkko Foundation for their support. AS acknowledges the support by the Conselho Nacional de Desenvolvimento Científico e Tecnológico (CNPq) (408974/2025-7, 312068/2025-5).

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

## APPENDIX OVERVIEW

The appendix complements the main text by providing formal definitions, detailed proofs, and auxiliary results that underpin our theoretical contributions. We present complete proofs for the main results stated in the paper, including the identical-vs-disjoint property (Proposition 3.1) and the equivalence theorems connecting $k$-CCWL, $\mathsf{TC}_{k+2}$, and the topological pebble game.

## A    GENERAL NOTATION

The set of real numbers is denoted by $\mathbb{R}$, the set of integers is denoted by $\mathbb{Z}$, and the set of natural numbers is denoted by $\mathbb{N}$. Furthermore, we denote $[n] := \{1, 2, \ldots, n\}$. Moreover, for any set of numbers $S$, the following notations are defined: the union of $S$ and $\{0\}$ is denoted by $S_0 := S \cup \{0\}$, and for any $m \in S$, the subset $S_{\geq m} := \{x \in S \mid x \geq m\}$ contains all elements in $S$ greater than or equal to $m$.

In general, for functions $f, g : X \to Y$, $f$ is said to *refine* $g$ (denoted by $f \preceq g$) if for all $x, x' \in X$, whenever $f(x) = f(x')$, it follows that $g(x) = g(x')$. Functions $f$ and $g$ are considered *equivalent*

(denoted by $f \equiv g$) if $f \preceq g$ and $g \preceq f$, meaning that $f$ and $g$ assign the same values to related elements in $X$. Moreover, consider a subset $A \subseteq X$, the restriction of $f$ to $A$, denoted by $f|_A$, is the function $f|_A : A \to Y$ defined by

$$f|_A(x) = f(x) \quad \text{for all } x \in A.$$

A *partial function* $f$ from a set $A$ to a set $B$ is a relation $f \subseteq A \times B$ such that for every $x \in A$, there is *at most one* $y \in B$ with $(x, y) \in f$. In other words, a partial function is a function that may not be defined for all elements of its domain $A$. In this paper, we use the arrow notation $f : A \rightharpoonup B$ to show partial function $f$. The domain of $f$, denoted by $\text{dom}_f$, indicates the set of all $x \in A$ for which $f(x)$ is defined.

## B CCWL AND $k$-CCWL

First, we give the formal definition of the combinatorial complex Weisfeiler-Leman (CCWL) algorithm.

**Combinatorial complex Weisfeiler-Leman (CCWL).** Let $\Gamma = (S, \mathcal{X}, \text{rk}, c)$ be an attributed combinatorial complex. The CCWL algorithm iteratively computes the labeling of cells in the combinatorial complex. The process is defined as follows:

1. Initialize the cell labeling: for each cell $x \in \mathcal{X}$, set $\chi_1^{(0)}(x) = (c(x), \text{rk}(x))$.

2. For each iteration $t \geq 1$, update the cell labels using:

$$\chi_1^{(t)}(x) = \left( \chi_1^{(t-1)}(x), \mathcal{M}_\Gamma^{(t)}(x) \right),$$

where

$$\mathcal{M}_\Gamma^{(t)}(x) = \left( c^{B,(t)}(x), c^{C,(t)}(x), c^{\downarrow,(t)}(x), c^{\uparrow,(t)}(x) \right).$$

The colors of the boundary cells are denoted by $c^{B,(t)}(x)$, the colors of the co-boundary cells are denoted by $c^{C,(t)}(x)$, the colors of the lower cells are denoted by $c^{\downarrow,(t)}(x)$, and the colors of the upper cells are denoted by $c^{\uparrow,(t)}(x)$. These sets are defined as follows:

$$c^{B,(t)}(x) = \left\{\!\!\left\{ \chi_1^{(t-1)}(y) \,\middle|\, y \in \mathcal{N}_B(x) \right\}\!\!\right\},$$

$$c^{C,(t)}(x) = \left\{\!\!\left\{ \chi_1^{(t-1)}(y) \,\middle|\, y \in \mathcal{N}_C(x) \right\}\!\!\right\},$$

$$c^{\downarrow,(t)}(x) = \left\{\!\!\left\{ \left( \chi_1^{(t-1)}(y), \chi_1^{(t-1)}(z) \right) \,\middle|\, y \in \mathcal{N}_\downarrow(x), z \in \mathcal{N}_B(x \cap y) \right\}\!\!\right\}, \text{ and}$$

$$c^{\uparrow,(t)}(x) = \left\{\!\!\left\{ \left( \chi_1^{(t-1)}(y), \chi_1^{(t-1)}(z) \right) \,\middle|\, y \in \mathcal{N}_\uparrow(x), z \in \mathcal{N}_C(x \cap y) \right\}\!\!\right\}$$

Here, for the sake of simplifying the notation, we write $\mathcal{N}_B(x \cap y)$ instead of $\mathcal{N}_B(x) \cap \mathcal{N}_B(y)$, and similarly $\mathcal{N}_C(x \cap y)$ instead of $\mathcal{N}_C(x) \cap \mathcal{N}_C(y)$. Moreover, whenever we work with two ACCs $A$ and $B$ in the same argument, we make the complexes explicit in the notation: for example, $\mathcal{N}(A, x)$ denotes the neighborhood of $x$ in $A$, and $\chi_{1,A}^{(t)}$ denotes the 1-CCWL coloring on $A$ at round $t$. In addition, after each iteration, collect all cell labels to form a global label for the combinatorial complex:

$$\chi_{1,\Gamma}^{(t)} = \left\{\!\!\left\{ \chi_1^{(t)}(x) \,\middle|\, x \in X \right\}\!\!\right\}.$$

The stable labeling is reached at iteration $t$ when, for each $x \in \mathcal{X}$, $\chi_1^{(t+1)}(x) = \chi_1^{(t)}(x)$. The stable labeling is denoted by $\chi_1^{(\infty)}(x)$. The set of stable labels for all cells is denoted as $\chi_{1,\Gamma}^{(\infty)}$.

**Convergence.** Before proceeding, we start with a simple termination result for $k$-CCWL for $k \geq 2$, and the termination of CCWL ($k = 1$) can be proved similarly.

**Lemma B.1** (Termination of $k$-CCWL). *Let $\Gamma = (S, \mathcal{X}, \mathrm{rk}, \mathrm{c})$ be an ACC with finite cell set $\mathcal{X}$, and let $k \geq 1$. Consider the $k$-CCWL refinement sequence*

$$\left(\chi_k^{(t)} : \mathcal{X}^k \to \Sigma_t\right)_{t \in \mathbb{N}_0}$$

*as defined in Section 3. Then this sequence stabilizes after finitely many iterations. In particular, there exists $T \in \mathbb{N}_0$ such that*

$$\chi_k^{(T+1)}(\mathbf{x}) = \chi_k^{(T)}(\mathbf{x}) \qquad \text{for all } \mathbf{x} \in \mathcal{X}^k,$$

*so the stable labeling $\chi_k^{(\infty)}$ is well-defined.*

*Proof of Lemma B.1.* Since $\mathcal{X}$ is finite, say $|\mathcal{X}| = n < \infty$, the set of $k$-tuples $U := \mathcal{X}^k$ is also finite with $|U| = n^k$. For each iteration $t \in \mathbb{N}_0$, the labeling

$$\chi_k^{(t)} : U \to \Sigma_t$$

induces an equivalence relation $\sim_t$ on $U$ via

$$\mathbf{x} \sim_t \mathbf{y} \iff \chi_k^{(t)}(\mathbf{x}) = \chi_k^{(t)}(\mathbf{y}),$$

and hence a partition $\mathcal{P}_t$ of $U$ into color classes. By the $k$-CCWL update rule (see the definition in Section 3), we have

$$\chi_k^{(t+1)}(\mathbf{x}) = \mathrm{HASH}\left(\chi_k^{(t)}(\mathbf{x}), \mathcal{D}_k^{(t)}(\mathbf{x})\right),$$

where $\mathcal{D}_k^{(t)}(\mathbf{x})$ is the multiset of refinement contexts

$$\mathcal{D}_k^{(t)}(\mathbf{x}) = \left\{\!\left\{ \left(\mathrm{atp}_{k+2}(\mathbf{x}\alpha\beta), \Delta_k^{(t)}(\mathbf{x}; \alpha, \beta)\right) : \alpha, \beta \in \mathcal{X} \right\}\!\right\}.$$

By assumption, HASH is injective on its arguments (cf. the discussion preceding the CCWL and $k$-CCWL definitions), so

$$\chi_k^{(t+1)}(\mathbf{x}) = \chi_k^{(t+1)}(\mathbf{y}) \implies \left(\chi_k^{(t)}(\mathbf{x}), \mathcal{D}_k^{(t)}(\mathbf{x})\right) = \left(\chi_k^{(t)}(\mathbf{y}), \mathcal{D}_k^{(t)}(\mathbf{y})\right).$$

In particular,

$$\chi_k^{(t+1)}(\mathbf{x}) = \chi_k^{(t+1)}(\mathbf{y}) \implies \chi_k^{(t)}(\mathbf{x}) = \chi_k^{(t)}(\mathbf{y}),$$

which means that any two tuples that are *merged* at round $t+1$ must already have been in the same color class at round $t$. Equivalently, each color class of $\mathcal{P}_{t+1}$ is contained in some color class of $\mathcal{P}_t$, so $\mathcal{P}_{t+1}$ is a refinement of $\mathcal{P}_t$.

If $\mathcal{P}_{t+1} \neq \mathcal{P}_t$, then at least one color class of $\mathcal{P}_t$ is split into two or more classes in $\mathcal{P}_{t+1}$, and hence

$$|\mathcal{P}_{t+1}| > |\mathcal{P}_t|.$$

On the other hand, each $\mathcal{P}_t$ is a partition of the finite set $U$, so

$$|\mathcal{P}_t| \leq |U| = n^k \qquad \text{for all } t \in \mathbb{N}_0.$$

Therefore, the sequence of partitions $\mathcal{P}_0, \mathcal{P}_1, \mathcal{P}_2, \ldots$ can strictly increase in size at most $n^k - 1$ times. It follows that there exists some $T \leq n^k - 1$ with

$$\mathcal{P}_{T+1} = \mathcal{P}_T,$$

which is equivalent to

$$\chi_k^{(T+1)}(\mathbf{x}) = \chi_k^{(T)}(\mathbf{x}) \qquad \text{for all } \mathbf{x} \in U = \mathcal{X}^k.$$

For all $t \geq T$, the refinement is stationary, so the stable labeling

$$\chi_k^{(\infty)} := \chi_k^{(T)}$$

is well-defined, and the $k$-CCWL procedure converges on $\Gamma$. $\qquad\square$

**Complexity of $k$-CCWL.** To complete our understanding of $k$-CCWL algorithm, Let $n := |X|$ denote the number of cells in the ACC $(S, \mathcal{X}, \mathrm{c}, \mathrm{rk})$ and let $\rho$ denote its order (maximum rank). In each $k$-CCWL iteration, we refine labels on all $k$-tuples of cells and for each tuple $x$ we aggregate over all pairs $(\alpha, \beta) \in X^2$ via the refinement context $D_k^{(t)}(x)$. From the definition in Section 3 and Lemma B.1, we obtain the following bounds:

- The number of tuples is $|X^k| = n^k$.

- For a fixed $k$-tuple $x$, in order to compute $D_k^{(t)}(x)$, we require to scan all pairs $(\alpha, \beta) \in X^2$ and, for each such pair,
    1. Compute the $(k+2)$-atomic type $\mathsf{atp}_{k+2}(x\alpha\beta)$, and
    2. Read the previously computed colors $\chi_k^{(t)}(x[\alpha/i])$ and $\chi_k^{(t)}(x[\beta/i])$ for $i = 1, \ldots, k$ to compute the double-shift sequence $\Delta_k^{(t)}(x; \alpha, \beta)$.

For fixed $k$ and $\rho$, if we store adjacency, ranks, and attributes in precomputed data structures, both operations can be implemented in time $O(1)$ per pair $(\alpha, \beta)$. In practice, we can store adjacencies in memory of $O(n^2)$, whose entries show the topological adjacency relation between each pair of cells. In addition, we can store ranks and attributes in arrays of size $O(n)$. Therefore:

- *Per-iteration cost:* $O(n^k)$ tuples times $O(n^2)$ pairs per tuple.
- *Number of iterations:* according to Lemma B.1, the refinement process terminates after at most $T \leq n^k - 1$ iterations, since each iteration strictly refines a partition of $X^k$ and there are only $n^k$ tuples.

Therefore, we can obtain a worst-case bound

$$\mathrm{TIME}(k\text{-CCWL}) \leq O\big(T \cdot n^{k+2}\big) \subseteq O\big(n^{2k+2}\big),$$

with space complexity $O(n^{\max(2,k)})$ to store tuple colors. In addition, we need the space for the ACC itself (adjacencies, ranks, attributes), which is at most quadratic in $n$ for fixed rank. Therefore, the overall space complexity is $O(n^{\max(2,k)})$ for fixed $k$.

Note that for the case $k = 1$, we again store the ACC in $O(n^2)$ space, and the per-iteration cost for each cell $x$ is still in $O(n^2)$:

1. We need to aggregate colors from boundary and coboundary cells, which can be done in $O(n)$.
2. We need to aggregate at most $O(n^2)$ color pairs for corresponding lower and upper multisets in the refinement process.

We know that the refinement process terminates after at most $T \leq n - 1$, therefore,

$$\mathrm{TIME}(1\text{-CCWL}) \leq O\big(T \cdot n^3\big) \subseteq O\big(n^4\big),$$

## B.1 BROADCAST-ANCHOR CELL

We proceed as follows: first we analyze the identical-vs-disjoint construction for uniform ACCs, then we establish the disjoint-or-identical property (Proposition 3.1), and finally we prove the equivalence results in Theorem 4.1, Theorem 5.1, and Theorem 5.2. Our aim is to make explicit all intermediate arguments, ensuring that the logical–game–algorithm correspondence is completely rigorous.

For two uniform ACCs $A = (S_A, \mathcal{X}_A, \mathrm{rk}_A, \mathrm{c}_A)$ and $B = (S_B, \mathcal{X}_B, \mathrm{rk}_B, \mathrm{c}_B)$, the $k$-CCWL algorithm distinguishes the two uniform ACCs $A$ and $B$, if their stable global labels are different. In other words, $k$-CCWL distinguishes $A$ and $B$, if

$$\chi_{A,k}^{(\infty)} \neq \chi_{B,k}^{(\infty)}.$$

This can also be interpreted as the existence of a color $C$ such that

$$\left| \left\{ u \in \mathcal{X}_A^k \,\middle|\, \chi_{k,A}^{(\infty)}(u) = C \right\} \right| \neq \left| \left\{ v \in \mathcal{X}_B^k \,\middle|\, \chi_{k,B}^{(\infty)}(v) = C \right\} \right| \tag{7}$$

To perform a $k$-CCWL test on these two uniform ACCs via $k$-CCWL, we first add a fresh cell with rank 0 to each uniform ACC and then connect this fresh cell to all 0-ranked cells via 1-ranked cells. Then, we apply $k$-CCWL on these two modified uniform ACCs and check if their stable global labels are different. Formally, let $a^*$ and $b^*$ be the fresh 0-ranked cells, and define the two new uniform ACCs $A^*$ and $B^*$ as follows:

$$A^* = (S_{A^*}, \mathcal{X}_{A^*}, \mathrm{rk}_{A^*}, \mathrm{c}_{A^*}), \tag{8}$$

where

$$S_{A^*} = S_A \cup \{a^*\}, \tag{9}$$

$$\mathcal{X}_{A^*} = \mathcal{X}_A \cup \{\{a^*, x\} \mid x \in \mathcal{X}_A(0)\} \cup \{a^*\}, \tag{10}$$

$$\mathrm{rk}_{A^*}(x) = \begin{cases} \mathrm{rk}_A(x), & \text{if } x \in \mathcal{X}_A, \\ 0, & \text{if } x = a^*, \\ 1, & \text{if } x = \{a^*, x'\}, x' \in \mathcal{X}_A(0). \end{cases} \tag{11}$$

$$\mathrm{c}_{A^*}(x) = \begin{cases} \mathrm{c}_A(x), & \text{if } x \in \mathcal{X}_A, \\ \text{a fresh color}, & \text{if } x = a^*, \\ \text{a fresh color}, & \text{if } x = \{a^*, x'\}, x' \in \mathcal{X}_A(0). \end{cases} \tag{12}$$

Similarly, for $B^*$:

$$B^* = (S_{B^*}, \mathcal{X}_{B^*}, \mathrm{rk}_{B^*}, \mathrm{c}_{B^*}), \tag{13}$$

where

$$S_{B^*} = S_B \cup \{b^*\}, \tag{14}$$

$$\mathcal{X}_{B^*} = \mathcal{X}_B \cup \{\{b^*, x\} \mid x \in \mathcal{X}_B(0)\} \cup \{b^*\}, \tag{15}$$

$$\mathrm{rk}_{B^*}(x) = \begin{cases} \mathrm{rk}_B(x), & \text{if } x \in \mathcal{X}_B, \\ 0, & \text{if } x = b^*, \\ 1, & \text{if } x = \{b^*, x'\}, x' \in \mathcal{X}_B(0). \end{cases} \tag{16}$$

$$\mathrm{c}_{B^*}(x) = \begin{cases} \mathrm{c}_B(x), & \text{if } x \in \mathcal{X}_B, \\ \text{a fresh color}, & \text{if } x = b^*, \\ \text{a fresh color}, & \text{if } x = \{b^*, x'\}, x' \in \mathcal{X}_B(0). \end{cases} \tag{17}$$

Next, we compare $\chi_{A^*,k}^{(\infty)}$ and $\chi_{B^*,k}^{(\infty)}$ to determine whether $k$-CCWL can distinguish $A$ and $B$. Note that the fresh 0-ranked and 1-ranked cells are initially colored with a new fresh color, so they can be distinguished from the original cells in $A$ and $B$. We need also to mention when during CC isomorphism test, the only way to define a mapping between $A$ and $B$ is to map the fresh 0-ranked cell in $A^*$ to the fresh 0-ranked cell in $B^*$, and similarly for the fresh 1-ranked cells. Therefore, we have the following result:

*Remark* B.2. There is a CC-isomorphism between $A^*$ and $B^*$ if and only if there is a CC-isomorphism between $A$ and $B$; in particular, adding broadcast-anchor cells does not affect whether $A$ and $B$ are CC-isomorphic.

For simplicity and to avoid redundancy, we use $A$ and $B$ to refer to these new uniform ACCs, previously denoted by $A^*$ and $B^*$, respectively.

Intuitively, these fresh cells act as broadcast agents to help distinguish $A$ and $B$. We show that if there is a difference in the global stable coloring of these two uniform ACCs, the fresh cells will propagate this difference to all other cells in $A$ and $B$.

**Base path.** In a uniform ACC $\Gamma = (S, \mathcal{X}, \mathrm{rk}, \mathrm{c})$, the *base distance* is defined as follows: the base path of a cell $x \in \mathcal{X}$, is a sequence of cells $x = x_0, x_1, \ldots, x_t$, such that

- $x_t$ is a non-fresh 0-ranked cell,
- for every $0 \le i < t$, $x_i \in \mathcal{N}_i(x_{i+1})$ for some $\mathcal{N}_i \in \{\mathcal{N}_B, \mathcal{N}_C, \mathcal{N}_\downarrow, \mathcal{N}_\uparrow\}$, and
- the path has minimal possible length and we define $d_{\mathrm{base}}(x) = t - 1$.

We denote a base path for $x$ by $\pi(x)$.

The goal of the following lemma is to prove that if 1-CCWL test is performed on $A$ and $B$, and the stable coloring of the fresh cells in two uniform ACCs is different, then the set of stable colors of all cells in $A$ and $B$ are disjoint. In other words, if the stable coloring of the fresh cells in $A$ and $B$ differs, this difference will be broadcast to all other cells in $A$ and $B$.

**Lemma B.3.** *If $A = (S_A, \mathcal{X}_A, \mathrm{rk}_A, \mathrm{c}_A)$ and $B = (S_B, \mathcal{X}_B, \mathrm{rk}_B, \mathrm{c}_B)$ are two uniform attributed combinatorial cell complexes, and 1-CCWL is performed on these two uniform ACCs. Let $a^*$ and $b^*$ be the fresh 0-ranked cells in $\mathcal{X}_{A^*} \setminus \mathcal{X}_A$ and $\mathcal{X}_{B^*} \setminus \mathcal{X}_B$, respectively. If $\chi_{1,A}^{(\infty)}(a^*) \neq \chi_{1,B}^{(\infty)}(b^*)$, then*

$$\left\{ \chi_{1,A}^{(\infty)}(x_A) \,\middle|\, x_A \in \mathcal{X}_A \right\} \cap \left\{ \chi_{1,B}^{(\infty)}(x_B) \,\middle|\, x_B \in \mathcal{X}_B \right\} = \emptyset.$$

*Proof of Lemma B.3.* Let $\Gamma = (S, \mathcal{X}, \mathrm{rk}, \mathrm{c})$ be a uniform ACC. In the 1-CCWL algorithm, each cell $x \in \mathcal{X}$ is initially colored with the pair of $(\mathrm{c}(x), \mathrm{rk}(x))$. Therefore, in the stable 1-CCWL coloring state, the cells of the same color have the same rank. To extend this, in the stable coloring state, if a cell $x_A \in \mathcal{X}_A$ has $\chi_{1,A}^{(\infty)}(x_A) = C$, then only cells $x_B \in \mathcal{X}_B$ with $d_{\mathrm{base}}^A(x_A) = d_{\mathrm{base}}^B(x_B)$, can have color $C$ as their stable color. This is because if the base distances of two cells $x_A$ and $x_B$ are different, then one cell receives messages from a 0-ranked cell earlier than the other cell during the 1-CCWL iterations, which leads to different stable colors for these two cells. Let $D_{\mathrm{base}}$ be the maximum base distance over both ACCs, which is defined as

$$D_{\mathrm{base}} := \max \left\{ \max_{x_A \in \mathcal{X}_A} d_{\mathrm{base}}^A(x_A), \ \max_{x_B \in \mathcal{X}_B} d_{\mathrm{base}}^B(x_B) \right\}. \tag{18}$$

Hence, the following statements are equivalent:

- The multisets of colors of cells with base distance $t$ are disjoint in $A$ and $B$, for all $t \leq D_{\mathrm{base}}$.

- The multisets of colors of all cells are disjoint in $A$ and $B$.

For $t \in [D_{\mathrm{base}}]_0$, define the multisets of stable colors at base distance $t$ by

$$\mathcal{C}_A(t) := \left\{\!\!\left\{ \chi_{1,A}^{(\infty)}(x) \,\middle|\, x_A \in \mathcal{X}_A, \ d_{\mathrm{base}}^A(x_A) = t \right\}\!\!\right\}, \tag{19}$$

$$\mathcal{C}_B(t) := \left\{\!\!\left\{ \chi_{1,B}^{(\infty)}(x) \,\middle|\, x_B \in \mathcal{X}_B, \ d_{\mathrm{base}}^B(x_B) = t \right\}\!\!\right\}. \tag{20}$$

Then the first statement can be formalized as

$$\mathcal{C}_A(t) \cap \mathcal{C}_B(t) = \emptyset \quad \text{for all } t \in [D_{\mathrm{base}}]_0. \tag{21}$$

Furthermore, in the stable coloring state, the color of a cell remains fixed and is still dependent on the colors of its neighbors. Since the stable coloring has been reached in both uniform attributed combinatorial complexes $A$ and $B$, if for two cells $x_A \in \mathcal{X}_A$ and $x_B \in \mathcal{X}_B$, there exists a cell in the boundary neighborhood of cell $x_A$ with a completely different color from any cell in the boundary neighborhood of cell $x_B$, it implies that the stable coloring of $x_A$ and $x_B$ are different. Therefore, for any two cells $x_A \in \mathcal{X}_A$, and $x_B \in \mathcal{X}_B$, if there exist a cell $y_A \in \mathcal{N}_B(A, x_A)$ such that for all $y_B \in \mathcal{N}_B(B, x_B)$, $\chi_{1,A}^{(\infty)}(y_A) \neq \chi_{1,B}^{(\infty)}(y_B)$, it follows that $\chi_{1,A}^{(\infty)}(x_A) \neq \chi_{1,B}^{(\infty)}(x_B)$. The same argument also applies to the coboundary, lower, and upper neighborhoods. To prove the Lemma B.3, we show Equation (21) using induction on $t$, which will imply the lemma.

1. Initial Step: We prove the equality in Equation (21) holds for $t = 0$. The cells with base distance 0, are exactly the 0-ranked cells in the uniform ACCs $A$ and $B$. The fresh cells $a^*$ and $b^*$ are connected to all 0-ranked cells in their respective uniform ACCs. By assumption, $\chi_{1,A}^{(\infty)}(a^*) \neq \chi_{1,B}^{(\infty)}(b^*)$, which implies that their adjacent 0-ranked cells do not have any common color. Thus,

$$\mathcal{C}_A(0) \cap \mathcal{C}_B(0) = \emptyset.$$

2. Inductive Step: Assume that the equality in Equation (21) holds for $t = i - 1$, then

$$\mathcal{C}_A(i-1) \cap \mathcal{C}_B(i-1) = \emptyset. \tag{22}$$

We prove that the equality Equation (21) holds for $t = i$. Consider the multisets of colors of cells with base distance $i$ in $A$ and $B$. Since $A$ and $B$ are uniform, each $i$-ranked cell has at least one adjacent cell with base distance $i - 1$. By the inductive hypothesis, the multisets of colors of cells with base distance $i - 1$ are disjoint in $A$ and $B$. Consequently, the multisets of colors of cells with base distance $i$ must also be disjoint in $A$ and $B$. Hence, the equality in Equation (21) holds for $t = i$, and

$$\mathcal{C}_A(i) \cap \mathcal{C}_B(i) = \emptyset. \tag{23}$$

By induction, Equation (21) holds for all $t \leq D_{\text{base}}$. Therefore, the lemma is proved. $\qquad\square$

We now show that if there is a difference in the global stable coloring of other cells, the fresh cells will capture these differences. Formally, we state the following lemma:

**Lemma B.4.** *Let $A = (S_A, \mathcal{X}_A, \text{rk}_A, c_A)$ and $B = (S_B, \mathcal{X}_B, \text{rk}_B, c_B)$ be two uniform attributed combinatorial cell complexes, and $1$-CCWL is performed on these two uniform ACCs. If there exists a color $C$ such that*

$$\left| \left\{ x_A \in \mathcal{X}_A \,\middle|\, \chi_{1,A}^{(\infty)}(x_A) = C \right\} \right| \neq \left| \left\{ x_B \in \mathcal{X}_B \,\middle|\, \chi_{1,B}^{(\infty)}(x_B) = C \right\} \right|,$$

*then $\chi_{1,A}^{(\infty)}(a^*) \neq \chi_{1,B}^{(\infty)}(b^*)$.*

*Proof of Lemma B.4.* We proceed with the same idea as in the proof of Lemma B.3, which is that cells of the same color have the same rank, because each cell is initially colored with the pair of its attribute and its rank. Therefore, it is sufficient to prove that if there is a different number of cells of color $C$ with base distance $t$ in $A$ and $B$, then the stable coloring of the fresh cells $a^*$ and $b^*$ in $A$ and $B$, respectively, are also different. Let $D_{\text{base}}$ be the maximum base distance over both ACCs, which is defined as

$$D_{\text{base}} := \max \left\{ \max_{x_A \in \mathcal{X}_A} d_{\text{base}}^A(x_A), \max_{x_B \in \mathcal{X}_B} d_{\text{base}}^B(x_B) \right\}. \tag{24}$$

Hence, we can conclude that the following two statements are equivalent:

1. There exists a color $C$ such that the number of cells with stable color $C$ and base distance $t$ is different in $A$ and $B$, for some $t \in [D_{\text{base}}]_0$.

2. There exists a color $C$ such that the number of cells with stable color $C$ is different in $A$ and $B$.

For $t \in [D_{\text{base}}]_0$, define the multisets of stable colors at base distance $t$ by

$$\mathcal{C}_A(C, t) := \left\{\!\!\left\{ x_A \in \mathcal{X}_A \,\middle|\, \chi_{1,A}^{(\infty)}(x_A) = C, \; d_{\text{base}}^A(x_A) = t \right\}\!\!\right\}, \tag{25}$$

$$\mathcal{C}_B(C, t) := \left\{\!\!\left\{ x_B \in \mathcal{X}_B \,\middle|\, \chi_{1,B}^{(\infty)}(x_B) = C, \; d_{\text{base}}^B(x_B) = t \right\}\!\!\right\}. \tag{26}$$

The first statement can be formalized as follows: There exists a color $C$ such that

$$|\mathcal{C}_A(C, t)| \neq |\mathcal{C}_B(C, t)|, \quad \text{for some } t \in [D_{\text{base}}]_0. \tag{27}$$

Instead of directly proving Lemma B.4, we now prove the following lemma:

**Lemma B.5.** *Let $A = (S_A, \mathcal{X}_A, \text{rk}_A, c_A)$ and $B = (S_B, \mathcal{X}_B, \text{rk}_B, c_B)$ be two uniform attributed combinatorial cell complexes, and $1$-CCWL is performed on these two uniform ACCs. If there exists a color $C$ such that*

$$|\mathcal{C}_A(C, t)| \neq |\mathcal{C}_B(C, t)|, \quad \text{for some } t \in [D_{\text{base}}]_0,$$

*then $\chi_{1,A}^{(\infty)}(a^*) \neq \chi_{1,B}^{(\infty)}(b^*)$.*

Next, we prove Lemma B.5, by using induction on $t$.

1. Initial Step: We prove Lemma B.5 holds for $t = 0$. The fresh cells $a^*$ and $b^*$ are connected to all 0-ranked cells in their respective uniform ACCs. Therefore, if there is a different number of cells of color $C$ with base distance $t = 0$ in $A$ and $B$, then $a^*$ and $b^*$ will have different colors. Formally, if for any color $C$, it holds that

$$|\mathcal{C}_A(C, 0)| \neq |\mathcal{C}_B(C, 0)|,$$

   then it implies that

$$\chi_{1,A}^{(\infty)}(a^*) \neq \chi_{1,B}^{(\infty)}(b^*).$$

2. Inductive Step: Assume Lemma B.5 holds for $t = i$. We must prove that if there exists a color $C$ such that

$$|\mathcal{C}_A(C, i)| \neq |\mathcal{C}_B(C, i)|, \tag{28}$$

   then $\chi_{1,A}^{(\infty)}(a^*) \neq \chi_{1,B}^{(\infty)}(b^*)$. By the induction hypothesis, for any color $C$, the number of cells with stable color $C$ and base distance $i - 1$ in $A$ and $B$ must be the same. If this is not the case, the lemma has already been proved. Thus, it is known that for all $C$, it holds that

$$|\mathcal{C}_A(C, i - 1)| = |\mathcal{C}_B(C, i - 1)|.$$

Since the number of cells with base distance $i - 1$ is equal, but the number of $i$-ranked cells with stable color $C$ differs, there is an inconsistency in the adjacency relations between cells of base distance $i$ with color $C$ and cells of base distance $i$. Therefore, the lemma is proved for $t = i$. Formally, without loss of generality, assume there is a color $C$ such that:

$$|\mathcal{C}_A(C, i)| > |\mathcal{C}_B(C, i)|.$$

In a uniform ACC, each cell of base distance $i$ must be adjacent to at least one cell of base distance $i - 1$. Assume that a cell of base distance $i$ with color $C$ is connected to a cell of base distance $i - 1$ with color $C'$. Since the stable coloring has been reached in both uniform ACCs, the number of adjacency relations between cells of base distance $i$ with color $C$ and cells of base distance $i - 1$ with color $C$ should be the same in $A$ and $B$. Formally, if we define

$$W_A(i) := \left\{ (x_A, y_A) \,\middle|\, \begin{array}{l} y_A \in \mathcal{N}(A, x_A), \quad \text{for some } \mathcal{N} \in \{\mathcal{N}_B, \mathcal{N}_C, \mathcal{N}_\downarrow, \mathcal{N}_\uparrow\}, \\ \left(\chi_{1,A}^{(\infty)}(x_A), \chi_{1,A}^{(\infty)}(y_A)\right) = (C, C'), \\ d_{\text{base}}^A(x_A) = d_{\text{base}}^A(y_A) + 1 = i \end{array} \right\},$$

$$W_B(i) := \left\{ (x_B, y_B) \,\middle|\, \begin{array}{l} y_B \in N_B^{\mathcal{N}_B}(x_B), \quad \text{for some } \mathcal{N} \in \{\mathcal{N}_B, \mathcal{N}_C, \mathcal{N}_\downarrow, \mathcal{N}_\uparrow\}, \\ \left(\chi_{1,B}^{(\infty)}(x_B), \chi_{1,B}^{(\infty)}(y_B)\right) = (C, C'), \\ d_{\text{base}}^B(x_B) = d_{\text{base}}^B(y_B) + 1 = i \end{array} \right\}.$$

it means that we must have $|W_A(i)| = |W_B(i)|$. In the stable state, this implies that each cell of base distance $i$ with color $C$ must have the same number of adjacent cells of base distance $i - 1$ with color $C'$. Since there are more cells of base distance $i$ with color $C$ in $A$, the pigeonhole principle ensures that at least a cell of base distance $i$ with color $C$ in $A$ has fewer adjacent cells of base distance $i - 1$ with color $C'$ than required. This contradicts the assumption that stable coloring has been reached. Thus, $|\mathcal{C}_A(C, i)| \neq |\mathcal{C}_B(C, i)|$. Hence, the lemma is proved for $t = i$.

Since Lemma B.5 holds for all $t \in [D_{\text{base}}]_0$, we conclude that Lemma B.4 is proved. $\qquad\square$

**Lemma B.6.** *Let $A = (S_A, \mathcal{X}_A, \text{rk}_A, c_A)$ and $B = (S_B, \mathcal{X}_B, \text{rk}_B, c_B)$ be two attributed combinatorial complexes, and let $k$-CCWL ($k \geq 2$) be performed on these two ACCs. Let $\chi_{k,A}^{(\infty)}(\mathbf{x}_A) = \chi_{k,B}^{(\infty)}(\mathbf{x}_B)$ for some $\mathbf{x}_A \in \mathcal{X}_A^k$ and $\mathbf{x}_B \in \mathcal{X}_B^k$. Then,*

$$\left\{\!\!\left\{ \left( \text{atp}_{k+2}(\mathbf{x}_A \alpha_A \alpha_A), \Delta_{k,A}^{(\infty)}(\mathbf{x}_A; \alpha_A, \alpha_A) \right) \,\middle|\, \alpha_A \in \mathcal{X}_A \right\}\!\!\right\}$$
$$= \left\{\!\!\left\{ \left( \text{atp}_{k+2}(\mathbf{x}_B \alpha_B \alpha_B), \Delta_{k,B}^{(\infty)}(\mathbf{x}_B; \alpha_B, \alpha_B) \right) \,\middle|\, \alpha_B \in \mathcal{X}_B \right\}\!\!\right\}.$$

*Proof of Lemma B.6.* For any $\alpha_A, \beta_A \in \mathcal{X}_A$ and $\alpha_B \in \mathcal{X}_B$, if

$$\left(\text{atp}_{k+2}(\mathbf{x}_A \alpha_A \beta_A),\ \Delta_{k,A}^{(\infty)}(\mathbf{x}_A;\ \alpha_A, \beta_A)\right)$$
$$= \left(\text{atp}_{k+2}(\mathbf{x}_B \alpha_B \alpha_B),\ \Delta_{k,B}^{(\infty)}(\mathbf{x}_B;\ \alpha_B, \alpha_B)\right),$$

then $\alpha_A = \beta_A$. This follows from the fact that the atomic type of a $k$-tuple encodes the equality of the cells in the tuple. Furthermore, for any $\alpha_B, \beta_B \in \mathcal{X}_B$ and $\alpha_A \in \mathcal{X}_A$, if

$$\left(\text{atp}_{k+2}(\mathbf{x}_A \alpha_A \alpha_A),\ \Delta_{k,A}^{(\infty)}(\mathbf{x}_A;\ \alpha_A, \alpha_A)\right)$$
$$= \left(\text{atp}_{k+2}(\mathbf{x}_B \alpha_B \beta_B),\ \Delta_{k,B}^{(\infty)}(\mathbf{x}_B;\ \alpha_B, \beta_B)\right),$$

then $\alpha_B = \beta_B$. Since $\chi_{k,A}^{(\infty)}(\mathbf{x}_A) = \chi_{k,B}^{(\infty)}(\mathbf{x}_B)$, it follows that $\mathcal{D}_{k,A}^{(\infty)}(\mathbf{x}_A) = \mathcal{D}_{k,B}^{(\infty)}(\mathbf{x}_B)$. Therefore,

$$\left\{\!\!\left\{ \left(\text{atp}_{k+2}(\mathbf{x}_A \alpha_A \beta_A),\ \Delta_{k,A}^{(\infty)}(\mathbf{x}_A;\ \alpha_A, \beta_A)\right) \middle| \alpha_A, \beta_A \in \mathcal{X}_A \right\}\!\!\right\}$$
$$= \left\{\!\!\left\{ \left(\text{atp}_{k+2}(\mathbf{x}_B \alpha_B \beta_B),\ \Delta_{k,B}^{(\infty)}(\mathbf{x}_B;\ \alpha_B, \beta_B)\right) \middle| \alpha_B, \beta_B \in \mathcal{X}_B \right\}\!\!\right\}.$$

By the above observations, if we restrict the left-hand side to $\alpha_A = \beta_A$, and the right-hand side to $\alpha_B = \beta_B$, the equality still holds. This completes the proof. □

Combining Lemmas B.3 and B.4, we conclude that when the CC isomorphism test is performed on $A$ and $B$ via $k$-CCWL, the stable global labels of these two UACCs are either entirely disjoint, or identical. This result is formalized in the following theorem:

**Proposition B.7** (Identical-vs-disjoint)**.** *Let* $A = (S_A, \mathcal{X}_A, \text{rk}_A, \text{c}_A)$ *and* $B = (S_B, \mathcal{X}_B, \text{rk}_B, \text{c}_B)$ *be two attributed combinatorial cell complexes, and $k$-CCWL is performed on these two uniform ACCs. Then, only one of the following two cases can occur:*

- $\left\{\!\!\left\{ \chi_{k,A}^{(\infty)}(\mathbf{x}_A) \middle| \mathbf{x}_A \in \mathcal{X}_A^k \right\}\!\!\right\} = \left\{\!\!\left\{ \chi_{k,B}^{(\infty)}(\mathbf{x}_B) \middle| \mathbf{x}_B \in \mathcal{X}_B^k \right\}\!\!\right\}$,

- $\left\{\!\!\left\{ \chi_{k,A}^{(\infty)}(\mathbf{x}_A) \middle| \mathbf{x}_A \in \mathcal{X}_A^k \right\}\!\!\right\} \cap \left\{\!\!\left\{ \chi_{k,B}^{(\infty)}(\mathbf{x}_B) \middle| \mathbf{x}_B \in \mathcal{X}_B^k \right\}\!\!\right\} = \emptyset$.

*Proof of Proposition B.7.* We prove if there exist two $k$-tuples $u_A \in \mathcal{X}_A^k$ and $u_B \in \mathcal{X}_B^k$ such that $\chi_{k,A}^{(\infty)}(u_A) = \chi_{k,B}^{(\infty)}(u_B)$, then

$$\left| \left\{ \mathbf{x}_A \in \mathcal{X}_A^k \middle| \chi_{k,A}^{(\infty)}(\mathbf{x}_A) = C \right\} \right|$$
$$= \left| \left\{ \mathbf{x}_B \in \mathcal{X}_B^k \middle| \chi_{k,B}^{(\infty)}(\mathbf{x}_B) = C \right\} \right|, \quad \text{for all color } C. \tag{29}$$

Note that Equation (29) holds if and only if the multisets of stable colors of $k$-tuples in $A$ and $B$ are the same, which is the first case in Proposition B.7.

If $k = 1$ and there exists a color $C$ such that

$$\left| \left\{ x_A \in \mathcal{X}_A \middle| \chi_{1,A}^{(\infty)}(x_A) = C \right\} \right| \neq \left| \left\{ x_B \in \mathcal{X}_B \middle| \chi_{1,B}^{(\infty)}(x_B) = C \right\} \right|,$$

then by using Lemma B.4, it follows that the stable coloring of the fresh cells $a^*$ and $b^*$ in $A$ and $B$ are different. Furthermore, by Lemma B.3, It can be concluded that

$$\left\{\!\!\left\{ \chi_{1,A}^{(\infty)}(x_A) \middle| x_A \in \mathcal{X}_A \right\}\!\!\right\} \cap \left\{\!\!\left\{ \chi_{1,B}^{(\infty)}(x_B) \middle| x_B \in \mathcal{X}_B \right\}\!\!\right\} = \emptyset.$$

However, this contradicts the assumption that $\chi_{1,A}^{(\infty)}(u_A) = \chi_{1,B}^{(\infty)}(u_B)$. Therefore, Equation (29) holds for $k = 1$.

If $k > 1$, then for $\Gamma \in \{A, B\}$ and any color $C$, define

$$F_\Gamma^C(\mathbf{x}, P) = \left\{ y \in \mathcal{X}_\Gamma^k \middle| \chi_{k,\Gamma}^{(\infty)}(y) = C,\ \text{diff}(\mathbf{x}, y) = P \right\}, \quad \text{for any } \mathbf{x} \in \mathcal{X}_\Gamma^k \text{ and } P \subseteq [k].$$

This is the set of all $k$-tuples in $\mathcal{X}_\Gamma^k$ that have color $C$ and differ from $\mathbf{x}$ precisely in the positions indexed by $P$. The proof proceeds by induction on $|P|$ to show that for any color $C$, it holds that

$$|F_A^C(u_A, P)| = |F_B^C(u_B, P)|, \quad \text{for all } P \subseteq [k]. \tag{30}$$

1. Initial step: We prove the equality in Equation (30) holds for $P$ with cardinality at most one. It is known that $\chi_{k,A}^{(\infty)}(u_A) = \chi_{k,B}^{(\infty)}(u_B)$, implying the equality in Equation (30). Let $P = \{i\}$ for some $i \in [k]$. Since $\chi_{k,A}^{(\infty)}(u_A) = \chi_{k,B}^{(\infty)}(u_B)$, it follows that $\mathcal{D}_{k,A}^{(\infty)}(u_A) = \mathcal{D}_{k,B}^{(\infty)}(u_B)$, and by Lemma B.6:

$$\left\{\!\!\left\{ \left( \mathrm{atp}_{A,k+2}(u_A x_A x_A), \Delta_{k,A}^{(\infty)}(u_A; x_A, x_A) \right) \,\middle|\, x_A \in \mathcal{X}_A \right\}\!\!\right\}$$
$$= \left\{\!\!\left\{ \left( \mathrm{atp}_{B,k+2}(u_B x_B x_B), \Delta_k^{(\infty)}(u_B; x_B, x_B) \right) \,\middle|\, x_B \in \mathcal{X}_B \right\}\!\!\right\}.$$

Thus, $F_A^C(u_A, \{i\}) = F_B^C(u_B, \{i\})$.

2. Inductive step: Assume that the equality in Equation (30) holds for $P$ with cardinality at most $n - 1$. We must show that it also holds for $P$ with cardinality $n$. By the induction hypothesis, for any color $C'$:

$$|F_A^{C'}(u_A, P')| = |F_B^{C'}(u_B, P')|,$$

where $P' = P \setminus \{m\}$ for some $m \in P$. Then, for any color $C$:

$$|F_A^C(u_A, P)| = \sum_{C'} \sum_{w_A \in F_A^{C'}(u_A, \{m\})} |F_A^C(w_A, P')|$$
$$= \sum_{C'} \sum_{w_B \in F_B^{C'}(u_B, \{m\})} |F_B^C(w_B, P')|$$
$$= |F_B^C(u_B, P)|.$$

Thus, the equality in Equation (30) holds for $P$ with cardinality $n$, which completes the proof.

$\square$

**Algorithmic benefits of Proposition 3.1.** The Proposition 3.1 states that when performing $k$-CCWL on two uniform ACCs, the stable colorings of these two ACCs are either identical or disjoint. This property can help to improve the performance of comparing global stable color of two ACCs. Formally, without this property, to compare the global stable color of these two uniform ACCs, we need to compare each color in the first ACC with each color in the second ACC. This requires $O(|\mathcal{X}_A|^k |\mathcal{X}_B|^k)$ comparisons. However, by using the identical-vs-disjoint property, we can reduce the number of comparisons to $O(\min(|\mathcal{X}_A^k|, |\mathcal{X}_B^k|))$, which is significantly more efficient.

**Uniformity requirement in Proposition 3.1.** As it can be observed from the proofs of Lemma B.3 and Lemma B.4, the uniformity is sufficient to maintain a flow of information from fresh cells to other cells, and vice versa. As a result, the identical-vs-disjoint property holds. Now, by giving the following example, we show that the uniformity is also necessary for the identical-vs-disjoint property to hold.

*Example* B.8 (Non-uniform ACCs breaking identical-vs-disjoint). Let $A = (S_A, \mathcal{X}_A, \mathrm{rk}_A, \mathrm{c}_A)$ and $B = (S_B, \mathcal{X}_B, \mathrm{rk}_B, \mathrm{c}_B)$ be two attributed combinatorial cell complexes. The ACC $A$ is defined as follows:

- $S_A = \{a_1, a_2, a_3, a_4, a_5, a_6\}$

- $\mathcal{X}_A(0) = \left\{ \{a_1\}, \{a_2\}, \{a_3\}, \{a_4\}, \{a_5\}, \{a_6\} \right\},$

- $\mathcal{X}_A(1) = \left\{ \{a_1, a_2\}, \{a_2, a_3\}, \{a_1, a_3\} \right\},$

- $\mathcal{X}_A(2) = \left\{ \{a_4, a_5, a_6\} \right\},$

- $\mathcal{X}_A = \mathcal{X}_A(0) \cup \mathcal{X}_A(1) \cup \mathcal{X}_A(2)$, and

- the attribute function $\mathrm{c}_A$ assigns the same attribute to all cells in $\mathcal{X}_A$.

Then, the ACC $B$ is defined as follows:

- $S_B = \{b_1, b_2, b_3, b_4, b_5, b_6\}$

- $\mathcal{X}_B(0) = \Big\{\{b_1\}, \{b_2\}, \{b_3\}, \{b_4\}, \{b_5\}, \{b_6\}\Big\}$,

- $\mathcal{X}_B(1) = \Big\{\{b_1, b_2\}, \{b_2, b_3\}\Big\}$,

- $\mathcal{X}_B(2) = \Big\{\{b_4, b_5, b_6\}\Big\}$,

- $\mathcal{X}_B = \mathcal{X}_B(0) \cup \mathcal{X}_B(1) \cup \mathcal{X}_B(2)$, and

- the attribute function $c_B$ assigns the same attribute to all cells in $\mathcal{X}_B$.

It can be seen that both ACCs are non-uniform, since in both of ACCs there are 2-ranked cells that are not adjacent to any 1-ranked cells or any other 2-ranked cells. Therefore, these 2-ranked cells cannot receive any information from other cells in their respective ACCs, and they are already reached to their stable state. The same argument applies to the cells $\{a_4\}$, $\{a_5\}$, and $\{a_6\}$ in ACC $A$ and the cells $\{b_4\}$, $\{b_5\}$, $\{b_6\}$ in ACC $B$. However, in $A$, the 1-ranked cells form a triangle, while in $B$, they form a path, and thus, the stable colors of 1-ranked cells in $A$ and $B$ are different. Therefore, the stable coloring of these two ACCs are neither identical nor disjoint, which shows that the uniformity is necessary for Proposition 3.1 to hold.

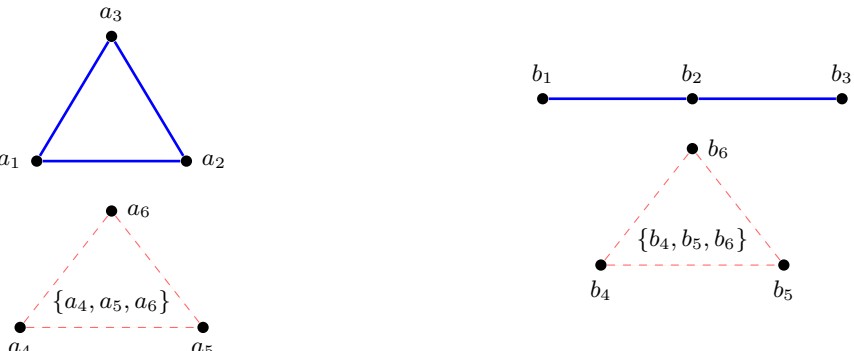

Figure 3: Non-uniform ACCs from Example B.8. Blue edges denote 1-cells; red shaded triangles denote 2-cells whose 1-faces are *not* in the complex (dashed edges), so the 2-cells have empty boundary.

## C  LOGIC

In this section, we first familiarize the reader with the basic concepts of logic, including syntax, semantics, and satisfiability. We also give the concepts of Topological Counting Logic $\mathsf{TC}_k$, which are extensions of First-Order Logic.

**Definition C.1** (Language). Let $\Sigma$ be a set of symbols called the *alphabet*. The set $L$ is called a language if $L \subseteq \Sigma^*$. The elements of $L$ are called *strings*.

*Example* C.2. Let $\Sigma = \{0, 1\}$. The set $L = \Sigma^*$ is a language of all binary strings. To familiarize the reader further, consider the English language. The alphabet $\Sigma'$ consists of all the letters in the English language, both uppercase and lowercase, as well as punctuation marks and spaces. An example of a language $L$ with alphabet $\Sigma'$ is the set of all valid English sentences. This language includes every possible combination of words and punctuation that form valid English sentences.

### C.1  FIRST-ORDER LOGIC

A first-order logical language $L$ is defined by the following symbols:

- Variables: $\text{Var}_L = \{x_1, x_2, x_3, \dots\}$
- Predicates and functions: The predicate symbols $P_1, P_2, P_3, \dots, P_i$, the binary equality symbol $=$, and the function symbols $f_1, f_2, f_3, \dots, f_j$.
- Logical connectives: $\wedge, \neg$
- Quantifier: $\exists$ (there exists)
- Parentheses and commas: $(, )$

Note that the 0-ary function symbols are called constants. It is possible for a logical language to not contain any function symbols or any predicate symbols other than equality.

### Syntax

The terms of the language are defined as follows:

- Each variable is a term.
- If $t_1, t_2, \dots, t_n$ are terms and $f$ is an $n$-ary function symbol, then $f(t_1, t_2, \dots, t_n)$ is a term.

The formulas of the logical language are defined as follows:

- If $t_1, t_2, \dots, t_n$ are terms and $P$ is an $n$-ary predicate symbol, then $P(t_1, t_2, \dots, t_n)$ is a formula.
- If $t_1$ and $t_2$ are terms, then $(t_1 = t_2)$ is a formula.
- If $\phi$ and $\psi$ are formulas, then $\phi \wedge \psi$ is a formula.
- If $\phi$ is a formula, then $\neg\phi$ is a formula.
- If $\phi$ is a formula and $x$ is a variable, then $\exists x\phi$ is a formula.

The formulas built from the first two rules are called **atomic formulas**.

**Definition C.3.** Let $\phi$ be a formula. The set of all variables in $\phi$ is denoted by $\text{Var}(\phi)$, which includes all variables that appear in $\phi$. The sets of **bound variables**, denoted by $\text{Bound}(\phi)$, and **free variables**, denoted by $\text{Free}(\phi)$, are defined recursively as follows:

1. If $\phi$ is an atomic formula:
$$\text{Bound}(\phi) = \emptyset, \quad \text{Free}(\phi) = \text{Var}(\phi).$$

2. If $\phi = \neg\psi$:
$$\text{Bound}(\phi) = \text{Bound}(\psi), \quad \text{Free}(\phi) = \text{Free}(\psi).$$

3. If $\phi = \psi_1 \wedge \psi_2$:
$$\text{Bound}(\phi) = \text{Bound}(\psi_1) \cup \text{Bound}(\psi_2), \quad \text{Free}(\phi) = \text{Free}(\psi_1) \cup \text{Free}(\psi_2).$$

4. If $\phi = \text{Q}x\psi$ where Q is a quantifier:
$$\text{Bound}(\phi) = \text{Bound}(\psi) \cup \{x\}, \quad \text{Free}(\phi) = \text{Free}(\psi) \setminus \{x\}.$$

Let $V \subseteq \text{Var}_L$. The sub-language $L(V)$ is the language that syntactically restricts the variables to $V$, using only variables from $V$.

### Semantics

A mathematical structure $S$ is $(D, P_1^S, \dots, P_i^S, f_1^S, \dots, f_j^S)$ where:

- $D$ is a non-empty set called the universe of $S$.
- $P_r^S \subseteq D^n$ for $r \in [i]$ are interpretations of $P_r$ in $S$, where $n$ is the arity of $P_r$.
- $f_r^S : D^n \to D$ for $r \in [j]$ are interpretations of $f_r$ in $S$, where $n$ is the arity of $f_r$.

A mapping $\mu : \text{Var}_L \rightharpoonup D$ is called a partial valuation. The term-mapping $\overline{\mu}$ assigns each term $t \in L(\text{dom}_\mu)$ to an element in $D$ as follows:

- If $t$ is a variable, then $\overline{\mu}(t) = \mu(t)$.
- If $t = f_r(t_1, t_2, \dots, t_n)$, then $\overline{\mu}(t) = f_r^S(\overline{\mu}(t_1), \overline{\mu}(t_2), \dots, \overline{\mu}(t_n))$ for $r \in [j]$.

Let $S$ be a structure and $\mu : \mathrm{Var}_L \rightharpoonup D$ a partial valuation for $L$. For any formula $\phi \in L$ with $\mathrm{Free}(\phi) \subseteq \mathrm{dom}_\mu$, the satisfaction relation $S, \mu \models \phi$ (or $S, \mu$ satisfies $\phi$) is defined as follows:

- If $\phi = P_r(t_1, t_2, \ldots, t_n)$, then $S, \mu \models \phi$ iff $(\overline{\mu}(t_1), \overline{\mu}(t_2), \ldots, \overline{\mu}(t_n)) \in P_r^S$.
- If $\phi = (t_1 = t_2)$, then $S, \mu \models \phi$ iff $\overline{\mu}(t_1) = \overline{\mu}(t_2)$.
- If $\phi = \psi_1 \wedge \psi_2$, then $S, \mu \models \phi$ iff $S, \mu \models \psi_1$ and $S, \mu \models \psi_2$.
- If $\phi = \neg\psi$, then $S, \mu \models \phi$ iff $S, \mu \not\models \psi$.[2]
- If $\phi = \exists x_i \psi$, then $S, \mu \models \phi$ iff there exists $c \in D$ such that $S, \mu[x_i \to c] \models \psi$.

Here, $\mu[x_i \to c]$ denotes the partial valuation that maps $x_i$ to $c$ and agrees with $\mu$ on all other variables. The set of all first-order logical languages is denoted by $\mathsf{FOL}$. We use the notation $\mathsf{FOL}_k$ denotes the set of languages containing at most $k$ variables.

## C.2 TOPOLOGICAL COUNTING LOGIC

Intuitively, we introduce a new variant of first-order logic capable of counting the number of pairs of elements satisfying a given property. To do this, we define a new counting quantifier that counts over pairs of variables. This new quantifier is called the *pairwise counting quantifier* and it is written as $\exists^N$ for any $N \in \mathbb{N}_0$. Although this symbol is the same as the standard counting quantifier, its syntax and semantics differ. This pairwise counting quantifier $\exists^N(x_i, x_j)$ counts pairs of elements in the universal set. Formally, if $\phi = \exists^N(x_i, x_j)\,\psi$, then $S, \mu \models \phi$ if and only if there exist at least $N$ pairs of elements $(c_1, c_2) \in D^2$ such that

$$S, \mu[x_i \to c_1, x_j \to c_2] \models \psi.$$

Here, $\mu[x_i \to c_1, x_j \to c_2]$ is the partial valuation that maps $x_i$ to $c_1$ and $x_j$ to $c_2$, and agrees with $\mu$ on all other variables. The pairwise counting quantifier can be recursively defined in terms of standard quantifiers as follows:

1. $\exists^0(x_i, x_j)\psi$ is equivalent to $\top$.
2. $\exists^{N+1}(x_i, x_j)\psi$ is equivalent to

$$\exists x_i \exists x_j \left( \psi \wedge \exists^N(y, z) \left( \neg(x_i = y) \wedge \neg(x_j = z) \wedge \psi[y/x_i, z/x_j] \right) \right)$$

Here, $\psi[y/x_i, z/x_j]$ is the formula obtained from $\psi$ by replacing all occurrences of $x_i$ with $y$ and all occurrences of $x_j$ with $z$. Although it is possible to recursively define the pairwise counting quantifier, it needs extra variables and is inefficient. Hence, the new quantifier is introduced to increase the expressive power of the logic.

The language $\mathsf{TC}_k$, finite topological counting logic with $k$ variables, is defined by the following symbols:

- Variables: $\mathrm{Var}_{\mathsf{TC}_k} = \{x_1, x_2, x_3, \ldots, x_k\}$
- Unary predicate symbols: $R_1, R_2, R_3, \ldots, R_\rho$ and $P_1, P_2, P_3, \ldots, P_\ell$
- Binary predicate symbols: $E^{\mathcal{N}_B}, E^{\mathcal{N}_C}, E^{\mathcal{N}_\downarrow}, E^{\mathcal{N}_\uparrow}$, and $=$
- Logical connectives: $\wedge, \neg$
- Counting quantifier: $\exists^N$ for $N \in \mathbb{N}_0$
- Parentheses and commas: $(,)$

**Syntax**

Since the only terms in the language $\mathsf{TC}_k$ are variables, the formulas of the language are defined as follows:

- If $x_i$ is a variable, then $R_r(x_i)$ is an atomic formula for $r \in [\rho]_0$.
- If $x_i$ is a variable, then $P_s(x_i)$ is an atomic formula for $s \in [\ell]$.
- If $x_i$ and $x_j$ are variables, then $E^{\mathcal{N}_B}(x_i, x_j)$, $E^{\mathcal{N}_C}(x_i, x_j)$, $E^{\mathcal{N}_\downarrow}(x_i, x_j)$, and $E^{\mathcal{N}_\uparrow}(x_i, x_j)$ are atomic formulas.

---

[2]The notation $S, \mu \not\models \psi$ indicates that $S, \mu$ do not satisfy the formula $\psi$.

- If $x_i$ and $x_j$ are variables, then $(x_i = x_j)$ is an atomic formula.
- If $\phi_1$ and $\phi_2$ are formulas, then $\phi_1 \wedge \phi_2$ is a formula.
- If $\phi$ is a formula, then $\neg\phi$ is a formula.
- If $\phi$ is a formula and $x_i$ and $x_j$ are distinct variables, then $\exists^N(x_i, x_j)\,\phi$ is a formula.

In terms of free and bound variables, the rules are similar to those of first-order logic, except that if $\phi = \exists^N(x_i, x_j)\,\psi$, then:

$$\text{Bound}(\phi) = \text{Bound}(\psi) \cup \{x_i, x_j\} \quad \text{Free}(\phi) = \text{Free}(\psi) \setminus \{x_i, x_j\}.$$

### Semantics

Let $\Gamma = (S, \mathcal{X}, \text{rk}, \text{c})$ be an attributed combinatorial complex. Consider the following structure:

$$(\mathcal{X}, R_0, \ldots, R_\rho, C_1, \ldots, C_\ell, E_\Gamma^{\mathcal{N}_B}, E_\Gamma^{\mathcal{N}_C}, E_\Gamma^{\mathcal{N}_\downarrow}, E_\Gamma^{\mathcal{N}_\uparrow}), \tag{31}$$

where $R_r = \mathcal{X}(r)$ for any $r \in [\rho]_0$ and

$$C_s = \{x \in \mathcal{X} \,|\, (\text{c}(x))_s = 1\} \quad \text{for all } s \in [\ell].$$

For any valuation $\mu : \{x_1, \ldots, x_k\} \to \mathcal{X}$ with $\text{Free}(\phi) \subseteq \text{dom}_\mu$, the satisfaction relation $\Gamma, \mu \models \phi$ is defined as follows:

- If $\phi = R_r(x_i)$, then $\Gamma, \mu \models \phi$ iff $\mu(x_i) \in R_r$.
- If $\phi = P_s(x_i)$, then $\Gamma, \mu \models \phi$ iff $\mu(x_i) \in C_s$.
- If $\phi = E^{\mathcal{N}_B}(x_i, x_j)$, then $\Gamma, \mu \models \phi$ iff $(\mu(x_i), \mu(x_j)) \in E_\Gamma^{\mathcal{N}_B}$.
- If $\phi = E^{\mathcal{N}_C}(x_i, x_j)$, then $\Gamma, \mu \models \phi$ iff $(\mu(x_i), \mu(x_j)) \in E_\Gamma^{\mathcal{N}_C}$.
- If $\phi = E^{\mathcal{N}_\uparrow}(x_i, x_j)$, then $\Gamma, \mu \models \phi$ iff $(\mu(x_i), \mu(x_j)) \in E_\Gamma^{\mathcal{N}_\uparrow}$.
- If $\phi = E^{\mathcal{N}_\downarrow}(x_i, x_j)$, then $\Gamma, \mu \models \phi$ iff $(\mu(x_i), \mu(x_j)) \in E_\Gamma^{\mathcal{N}_\downarrow}$.
- If $\phi = (x_i = x_j)$, then $\Gamma, \mu \models \phi$ iff $\mu(x_i) = \mu(x_j)$.
- If $\phi = (\psi_1 \wedge \psi_2)$, then $\Gamma, \mu \models \phi$ iff $\Gamma, \mu \models \psi_1$ and $\Gamma, \mu \models \psi_2$.
- If $\phi = \neg\psi$, then $\Gamma, \mu \models \phi$ iff $\Gamma, \mu \not\models \psi$.
- If $\phi = \exists^N(x_i, x_j)\,\psi$, then $\Gamma, \mu \models \phi$ iff there exist at least $N$ pairs of elements $(c_1, c_2) \in \mathcal{X}^2$ such that $\Gamma, \mu[x_i \to c_1, x_j \to c_2] \models \psi$.

Therefore, $\Gamma = (S, \mathcal{X}, \text{rk}, \text{c})$ can represent a structure in the language $\mathsf{TC}_k$. For simplicity, instead of using the structure 31, we use $\Gamma$ itself as a structure.

**Guarded logic $\mathsf{GTC}_3$.** The guarded logic $\mathsf{GTC}_3$ is a fragment of $\mathsf{TC}_3$, which only allows to have formulas of the form $x_i = x_j$, rank and attribute predicates, and formula $\phi$ of the following form:

$$\begin{aligned}
\phi &= \neg\psi, \\
\phi &= \psi_1(x) \wedge \psi_2(x), \\
\phi &= \exists^N(y, z)\big(y = z \wedge E^{\mathcal{N}}(y, x) \wedge \psi(y)\big), \\
\phi &= \exists^N(y, z)\big(E^{\mathcal{N}_1}(y, x) \wedge E^{\mathcal{N}_2}(z, x) \wedge E^{\mathcal{N}_2}(z, y) \wedge \psi_1(y) \wedge \psi_2(z)\big)
\end{aligned}$$

where $\mathcal{N} \in \{\mathcal{N}_B, \mathcal{N}_C\}$, $(\mathcal{N}_1, \mathcal{N}_2) \in \{(\mathcal{N}_\downarrow, \mathcal{N}_B), (\mathcal{N}_\uparrow, \mathcal{N}_C)\}$, and $\psi, \psi_1, \psi_2$ are again $\mathsf{GTC}_3$ formulas. The semantics are inherited from $\mathsf{TC}_3$.

### C.3 GENERAL NOTATION FOR LOGICAL LANGUAGES

Let $L$ be a language with at least $k$ variables, $S$ be a structure with universe $D$, and $\mu$ be a partial valuation for $L$. For any formula $\phi \in L$, when it is said that $S, \mu \models \phi$ or $S, \mu \not\models \phi$, it necessarily means that $\text{Free}(\phi) \subseteq \text{dom}_\mu$. Otherwise, the satisfaction relation is not defined. In cases where $\text{dom}_\mu = \emptyset$, instead of writing $S, \mu \models \phi$ or $S, \mu \not\models \phi$, it is said that $S \models \phi$ or $S \not\models \phi$, respectively. Each $\mu = (\mu_1, \ldots, \mu_k) \in D^k$ can represent a partial valuation such that $\mu(x_i) = \mu_i$ for all $i \in [k]$. Additionally, when we have that $S_1, \mu_1$ and $S_2, \mu_2$ agree on a formula $\phi \in L$, it means that both $S_1, \mu_1$ and $S_2, \mu_2$ either satisfy $\phi$ or do not satisfy $\phi$.

**Definition C.4** (*L*-Equivalence). Let $S_1$ and $S_2$ be two structures for a language $L$, and let $\mu_1$ and $\mu_2$ be partial valuations corresponding to $S_1$ and $S_2$, respectively. The following statements are equivalent, and we say $S_1, \mu_1$ and $S_2, \mu_2$ are $L$-equivalent:

1. $S_1, \mu_1$ and $S_2, \mu_2$ agree on all formulas $\phi \in L$.

2. $S_1, \mu_1 \underset{L}{\equiv} S_2, \mu_2$.

3. $S_1, \mu_1$ and $S_2, \mu_2$ are not $L$-distinguishable.

These statements are equivalent to the following conditions:

- $\text{dom}_{\mu_1} = \text{dom}_{\mu_2}$

- For all $\phi \in L$, it holds that $S_1, \mu_1 \models \phi$ if and only if $S_2, \mu_2 \models \phi$.

In terms of $\mathsf{TC}_k$, defining the length and depth of logical formulas can bring an inductive approach to solving the problems. For a formula $\phi$, $\text{len}(\phi)$ is defined recursively as follows:

1. If $\phi$ is an atomic formula, then $\text{len}(\phi) = 1$.
2. If $\phi$ is of the form $\neg\psi$, then $\text{len}(\phi) = 1 + \text{len}(\psi)$.
3. If $\phi$ is of the form $\psi_1 \wedge \psi_2$, then $\text{len}(\phi) = 1 + \text{len}(\psi_1) + \text{len}(\psi_2)$.
4. If $\phi$ is of the form $\exists^N x\psi$ or $\exists^N(x,y)\psi$, then $\text{len}(\phi) = 1 + \text{len}(\psi)$.

The *quantifier depth* refers to the maximum nesting level of quantifiers within a formula, indicating how deeply quantifiers are embedded. For a formula $\phi$, $\text{depth}(\phi)$ is defined recursively as follows:

1. If $\phi$ is an atomic formula, then $\text{depth}(\phi) = 0$.
2. If $\phi$ is of the form $\neg\psi$, then $\text{depth}(\phi) = \text{depth}(\psi)$.
3. If $\phi$ is of the form $\psi_1 \wedge \psi_2$, then $\text{depth}(\phi) = \max(\text{depth}(\psi_1), \text{depth}(\psi_2))$.
4. If $\phi$ is of the form $\exists^N x\psi$ or $\exists^N(x,y)\psi$, then $\text{depth}(\phi) = 1 + \text{depth}(\psi)$.

The notations $\mathsf{TC}_{k,t}$ denote the sets of formulas in $\mathsf{TC}_k$ with quantifier depth at most $t$, respectively.

## C.4 RELATION OF LOGIC $\mathsf{TC}_k$ AND $k$-CCWL

In this section, we explore the expressive power of $k$-CCWL in terms of logic. Intuitively, the following lemma shows that there exists a logical formula capable of characterizing $k$-tuples with the same atomic type value.

**Lemma C.5.** *Let* $\Gamma = (S, \mathcal{X}, \text{rk}, \text{c})$ *be a combinatorial cell complex and let* $a$ *be an atomic type value. Assume the representation of* $a$ *is as follows:*

$$\mathbf{a} = a_{x_1,x_2}^= \cdots a_{x_1,x_2}^{\mathcal{N}_B} \cdots a_{x_1,x_2}^{\mathcal{N}_C} \cdots a_{x_1,x_2}^{\mathcal{N}_\downarrow} \cdots a_{x_1,x_2}^{\mathcal{N}_\uparrow} \cdots a_{x_1,1}^c \ldots a_{x_1,\ell}^c \cdots a_{x_1,1}^{\text{rk}} \ldots a_{x_1,\rho}^{\text{rk}} \cdots$$

*where for all* $x_i, x_j \in \mathcal{X}$,

- $a_{x_i,x_j}^=$ *encodes the equality information between* $x_i$ *and* $x_j$,

- $a_{x_i,x_j}^{\mathcal{N}_B}$, $a_{x_i,x_j}^{\mathcal{N}_C}$, $a_{x_i,x_j}^{\mathcal{N}_\downarrow}$, $a_{x_i,x_j}^{\mathcal{N}_\uparrow}$ *encode the boundary, coboundary, lower, and upper adjacency relations between* $x_i$ *and* $x_j$, *respectively,*

- $a_{x_i,l}^c$ *encodes the information about* $c_\ell(x_i)$, *and*

- $a_{x_i,r}^{\text{rk}}$ *encodes the rank information of* $x_i$.

*Then, there exists a formula in* $\mathsf{TC}_k$ *that characterizes* $k$-*tuples having the atomic type value* $a$. *Specifically, there exists a formula* $\phi_a$ *in* $\mathsf{TC}_k$ *such that for all* $\mathbf{x} \in \mathcal{X}^k$, *it holds that:*

$$\Gamma, \mathbf{x} \models \phi_a \text{ if and only if } \text{atp}_{\Gamma,k}(\mathbf{x}) = a.$$

*Proof of Lemma C.5.* To prove this lemma, we construct the formula $\phi_a$ in $\mathsf{TC}_k$ iteratively as follows:

1. Initialization: Let $\phi_a = ()$.

2. Equality: For each $1 \leq i < j \leq k$, if $a^{=}_{x_i, x_j} = 1$, then $\phi_a = \phi_a \wedge (x_i = x_j)$; otherwise, $\phi_a = \phi_a \wedge \neg(x_i = x_j)$.

3. Adjacencies: For each $\mathcal{N} \in \{\mathcal{N}_B, \mathcal{N}_C, \mathcal{N}_\downarrow, \mathcal{N}_\uparrow\}$, if $i < j$ and $a^{\mathcal{N}}_{x_i, x_j} = 1$, then $\phi_a = \phi_a \wedge E^{\mathcal{N}}(x_i, x_j)$; otherwise, $\phi_a = \phi_a \wedge \neg E^{\mathcal{N}}(x_i, x_j)$.

4. Color: For each $s \in [\ell]$, if $a^c_{x_i, s} = 1$, then $\phi_a = \phi_a \wedge P_s(x_i)$; otherwise, $\phi_a = \phi_a \wedge \neg P_s(x_i)$.

5. Rank: For each $r \in [\rho]_0$, if $a^{rk}_{x_i, r} = 1$, then $\phi_a = \phi_a \wedge R_r(x_i)$; otherwise, $\phi_a = \phi_a \wedge \neg R_r(x_i)$.

Now, we prove that the following holds:

$$\Gamma, x \models \phi_a \iff \text{atp}_{\Gamma, k}(x) = a, \quad \text{for all } x \in \mathcal{X}^k$$

We prove this in two directions:

1. If $\text{atp}_{\Gamma, k}(x) = a$, then $\Gamma, x \models \phi_a$: By construction, $\phi_a$ is built to match the atomic structure $a$. Thus, if $\text{atp}_{\Gamma, k}(x) = a$, it directly follows that $\Gamma, x \models \phi_a$.

2. If $\Gamma, x \models \phi_a$, then $\text{atp}_{\Gamma, k}(x) = a$: Assume, for contradiction, that $\Gamma, x \models \phi_a$ and $\text{atp}_{\Gamma, k}(x) = b \neq a$. This implies that $\Gamma, x \models \phi_b$. Therefore, $\Gamma, x \models \phi_a \wedge \phi_b$, which leads to $\Gamma, x \models \bot$. This contradicts the fact that $\Gamma$ is a non-trivial and valid combinatorial complex.

$\square$

The next lemma shows that if the initial coloring of two ACCs $A$ and $B$ differs during the execution of $k$-CCWL, then there exists a quantifier-free logical formula in $\mathsf{TC}_k$ that can distinguish these two ACCs.

**Lemma C.6.** *Let $A = (S_A, \mathcal{X}_A, \text{rk}_A, c_A)$ and $B = (S_B, \mathcal{X}_B, \text{rk}_B, c_B)$ be two attributed combinatorial complexes. Let $u_A \in \mathcal{X}_A^k$ and $u_B \in \mathcal{X}_B^k$. If $A, u_A$ and $B, u_B$ agree on all quantifier-free formulas in $\mathsf{TC}_k$, then it implies:*

$$\chi^{(0)}_{k, A}(u_A) = \chi^{(0)}_{k, B}(u_B)$$

*Proof of Lemma C.6.* Since $A, u_A$ and $B, u_B$ agree on all quantifier-free formulas, $A, u_A$ and $B, u_B$ agree on $\{R_r(x_i), \neg R_r(x_i)\}$, for all $i \in [k]$ and $r \in [\rho]_0$. Consequently, we conclude

$$\text{rk}_A(u_A(x_i)) = \text{rk}_B(u_B(x_i)), \quad \text{for all } i \in [k].$$

Similarly, $A, u_A$ and $B, u_B$ agree on $\{P_s(x_i), \neg P_s(x_i)\}$ for all $i \in [k]$ and $s \in [l]$. Therefore, we conclude

$$c_A(u_A(x_i)) = c_B(u_B(x_i)), \quad \text{for all } i \in [k].$$

Moreover, $A, u_A$ and $B, u_B$ agree on $\{x_i = x_j, \neg(x_i = x_j)\}$ for all $i, j \in [k]$, which implies:

$$u_A(x_i) = u_A(x_j) \iff u_B(x_i) = u_B(x_j), \quad \text{for all } i, j \in [k]$$

Furthermore, $A, u_A$ and $B, u_B$ agree on $\{E^{\mathcal{N}}(x_i, x_j), \neg E^{\mathcal{N}}(x_i, x_j)\}$ for all $i, j \in [k]$ and $\mathcal{N} \in \{\mathcal{N}_B, \mathcal{N}_C, \mathcal{N}_\downarrow, \mathcal{N}_\uparrow\}$. Therefore, we conclude for all $i, j \in [k]$, and $\mathcal{N} \in \{\mathcal{N}_B, \mathcal{N}_C, \mathcal{N}_\downarrow, \mathcal{N}_\uparrow\}$:

$$u_A(x_i) \in \mathcal{N}(A, u_A(x_j)) \Leftrightarrow u_B(x_i) \in \mathcal{N}(B, u_B(x_j)).$$

Thus, we conclude that $A, u_A \sim_k B, u_B$, and consequently, $\text{atp}_{A, k}(u_A) = \text{atp}_{B, k}(u_B)$. Therefore, $\chi^{(0)}_{k, A}(u_A) = \chi^{(0)}_{k, B}(u_B)$. For $k = 1$, having $c_A(u_A(x)) = c_B(u_B(x))$ and $\text{rk}_A(u_A(x)) = \text{rk}_B(u_B(x))$ is sufficient to obtain the desired result. $\square$

The following two lemmas show that if two ACCs $A$ and $B$ are distinguished by $k$-CCWL, then they can also be distinguished by $\mathsf{TC}_{k+2}$. Furthermore, there exists a logical formula in $\mathsf{TC}_{k+2}$ on which these two ACCs do not agree on.

**Lemma C.7.** *Let $A = (S_A, \mathcal{X}_A, \mathrm{rk}_A, c_A)$ and $B = (S_B, \mathcal{X}_B, \mathrm{rk}_B, c_B)$ be two attributed combinatorial complexes. Let $u_A \in \mathcal{X}_A$ and $u_B \in \mathcal{X}_B$. If $A, u_A$ and $B, u_B$ are not $\mathsf{GTC}_3$-distinguishable, then*

$$\chi_{1,A}^{(\infty)}(u_A) = \chi_{1,B}^{(\infty)}(u_B).$$

*Proof of Lemma C.7.* We prove

$$A, u_A \underset{\mathsf{GTC}_{3,t}}{\equiv} B, u_B \implies \chi_{1,A}^{(t)}(u_A) = \chi_{1,B}^{(t)}(u_B), \quad \text{for all } t \in \mathbb{N}_0, \tag{32}$$

by induction on $t$.

1. Initial step: We prove that Equation (32) holds for $t = 0$. According to Lemma C.6, if $A, u_A$ and $B, u_B$ agree on all quantifier-free formulas, then $\chi_{1,A}^{(0)}(u_A) = \chi_{1,B}^{(0)}(u_B)$. Thus, the result holds for $t = 0$.

2. Inductive step: Assume that Equation (32) holds for all $t \in [n-1]_0$. We must show that it also holds for $t = n$. For contradiction, assume $\chi_{1,A}^{(n)}(u_A) \neq \chi_{1,B}^{(n)}(u_B)$. Then, there are two cases to consider:

   - If $\chi_{1,A}^{(n-1)}(u_A) \neq \chi_{1,B}^{(n-1)}(u_B)$: By the induction hypothesis, there exists a formula $\phi \in \mathsf{GTC}_{3,n-1} \subset \mathsf{GTC}_{3,n}$ such that $A, u_A$ and $B, u_B$ do not agree. This contradicts the assumption.

   - If $\mathcal{M}_A^{(n)}(u_A) \neq \mathcal{M}_B^{(n)}(u_B)$: This case only needs to be analyzed if the aggregated values of either the boundary neighborhood or the lower adjacency neighborhood. The reasoning for other types of neighborhoods is similar. Suppose $c^{B,(n)}(A, u_A) \neq c^{B,(n)}(B, u_B)$. Then, there exists a color $\tilde{C}$ such that:

$$\left| \left\{ \alpha_A \in \mathcal{N}_B(A, u_A) \, \middle| \, \chi_{1,A}^{(n-1)}(\alpha_A) = \tilde{C} \right\} \right| \neq \left| \left\{ \alpha_B \in \mathcal{N}_B(B, u_B) \, \middle| \, \chi_{1,B}^{(n-1)}(\alpha_B) = \tilde{C} \right\} \right|.$$

Let $N$ be the cardinality of the larger set. Since two cells have the same color in iteration $n-1$ if they agree on all formulas in $\mathsf{GTC}_{3,n-1}$. Let $\psi_C$ be the conjunction of the formulas in $\mathsf{GTC}_{3,n-1}$, which characterizes color $C$ on iteration $n-1$. Formally, for any color $C$:

$$\Gamma, x_\Gamma \models \psi_C \Leftrightarrow \chi_{1,\Gamma}^{(n-1)}(x_\Gamma) = C, \quad \text{for all } \Gamma \in \{A, B\}, \text{and } x_\Gamma \in \mathcal{X}.$$

Now consider the formula:

$$\xi = \exists^N(x_2, x_3) \left( \psi_{\tilde{C}}(x_2) \wedge E^{\mathcal{N}_B}(x_2, x_1) \wedge (x_2 = x_3) \right).$$

It follows that $A, u_A$ and $B, u_B$ will not agree on the formula $\xi \in \mathsf{GTC}_{3,n}$, and it contradicts with the fact that $A, u_A$ and $B, u_B$ agree on all formulas in $\mathsf{GTC}_{3,n}$. Moreover, if $c^{\downarrow,(n)}(A, u_A) \neq c^{\downarrow,(n)}(B, u_B)$, then there exist colors $\tilde{C}$ and $\hat{C}$ such that

$$\left| \left\{ (\alpha_A, \beta_A) \, \middle| \, \begin{array}{l} \alpha_A \in \mathcal{N}^\downarrow(A, u_A), \\ \beta_A \in \mathcal{N}^B(A, u_A \cap \alpha_A), \\ \chi_{1,A}^{(n-1)}(\alpha_A) = \tilde{C}, \\ \chi_{1,A}^{(n-1)}(\beta_A) = \hat{C} \end{array} \right\} \right| \neq \left| \left\{ (\alpha_B, \beta_B) \, \middle| \, \begin{array}{l} \alpha_B \in \mathcal{N}^\downarrow(B, u_B), \\ \beta_B \in \mathcal{N}^B(B, u_B \cap \alpha_B), \\ \chi_{1,B}^{(n-1)}(\alpha_B) = \tilde{C}, \\ \chi_{1,B}^{(n-1)}(\beta_B) = \hat{C} \end{array} \right\} \right|$$

Let $N$ be the cardinality of the larger set. Consider the formula:

$$\xi = \exists^N(x_2, x_3) \left( \psi_{\tilde{C}}(x_2) \wedge \psi_{\hat{C}}(x_3) \wedge E^{\mathcal{N}_\downarrow}(x_2, x_1) \wedge E^{\mathcal{N}_B}(x_3, x_1) \wedge E^{\mathcal{N}_B}(x_3, x_2) \right).$$

Again, $A, u_A$ and $B, u_B$ do not agree on the formula $\xi \in \mathsf{GTC}_{3,n}$, which contradicts the assumption. Therefore, $\chi_{1,A}^{(n)}(u_A) = \chi_{1,B}^{(n)}(u_B)$.

Hence, the lemma holds for all $t \in \mathbb{N}_0$, and we conclude

$$A, u_A \underset{\mathsf{GTC}_{3,t}}{\equiv} B, u_B \implies \chi_{1,A}^{(\infty)}(u_A) = \chi_{1,B}^{(\infty)}(u_B).$$

$\square$

**Theorem C.8.** *Let $A = (S_A, \mathcal{X}_A, \mathrm{rk}_A, c_A)$ and $B = (S_B, \mathcal{X}_B, \mathrm{rk}_B, c_B)$ be two attributed combinatorial complexes and let $u_A \in \mathcal{X}_A^k$ and $u_B \in \mathcal{X}_B^k$. Then if $A, u_A$ and $B, u_B$ are not $\mathsf{TC}_{k+2}$-distinguishable, then:*

$$\chi_{k,A}^{(\infty)}(u_A) = \chi_{k,B}^{(\infty)}(u_B)$$

*Proof of Theorem C.8.* We prove

$$A, u_A \underset{\mathsf{TC}_{k+2,t}}{\equiv} B, u_B \implies \chi_{k,A}^{(t)}(u_A) = \chi_{k,B}^{(t)}(u_B), \quad \text{for all } t \in \mathbb{N}_0, \tag{33}$$

by induction on $t$.

1. Initial step: We prove that Equation (33) holds for $t = 0$. According to Lemma C.6, if $A, u_A$ and $B, u_B$ agree on all quantifier-free formulas, then $\chi_{k,A}^{(0)}(u_A) = \chi_{k,B}^{(0)}(u_B)$. Thus, the result holds for $t = 0$.

2. Inductive step: Assume that Equation (33) holds for all $t \in [n-1]_0$. We must show that it also holds for $t = n$. For contradiction, assume $\chi_{k,A}^{(n)}(u_A) \neq \chi_{k,B}^{(n)}(u_B)$. Then, there are two cases to consider:

   - If $\chi_{k,A}^{(n-1)}(u_A) \neq \chi_{k,B}^{(n-1)}(u_B)$: By the induction hypothesis, there exists a formula $\phi \in \mathsf{TC}_{k+2,n-1} \subset \mathsf{TC}_{k+2,n}$ such that $A, u_A$ and $B, u_B$ do not agree. This contradicts the assumption.

   - If $\mathcal{D}_A^{(n)}(u_A) \neq \mathcal{D}_B^{(n)}(u_B)$: there exists a $k$-tuple of pairs of colors $\bar{C} = \left( \begin{pmatrix} \tilde{C}_1 \\ \hat{C}_1 \end{pmatrix}, \ldots, \begin{pmatrix} \tilde{C}_k \\ \hat{C}_k \end{pmatrix} \right)$ and an atomic type value $a$ of size $k + 2$, such that:

$$\left| \left\{ (\alpha_A, \beta_A) \in \mathcal{X}_A^2 \,\middle|\, \left( \mathrm{atp}_{k+2}(u_A \alpha_A \beta_A),\ \Delta_{k,A}^{(n-1)}(u_A; \alpha_A, \beta_A) \right) = (a, \bar{C}) \right\} \right|$$
$$\neq \left| \left\{ (\alpha_B, \beta_B) \in \mathcal{X}_B^2 \,\middle|\, \left( \mathrm{atp}_{k+2}(u_B \alpha_B \beta_B),\ \Delta_{k,B}^{(n-1)}(u_B; \alpha_B, \beta_B) \right) = (a, \bar{C}) \right\} \right|.$$

   Let $N$ be the cardinality of the larger set. Cells have the same color in iteration $n - 1$ if they agree on all formulas in $\mathsf{TC}_{k+2,n-1}$. Let $\psi_C$ be the conjunction of the formulas $\mathsf{TC}_{k+2,n-1}$, which characterize color $C$ in iteration $n - 1$. Formally, for any color $C$:

$$\Gamma, x_\Gamma \models \psi_C \iff \chi_{k,\Gamma}^{(n-1)}(x_\Gamma) = C, \quad \text{for all } \Gamma \in \{A, B\}, \text{ and } x_\Gamma \in \mathcal{X}.$$

   According to Lemma C.5, let $\phi_a \in \mathsf{TC}_{k+2}$ be the logical formula that characterizes the atomic type value $a$. Formally:

$$\Gamma, x_\Gamma \models \phi_a \iff \mathrm{atp}_{\Gamma, k+2}(x_\Gamma) = a, \quad \text{for all } \Gamma \in \{A, B\}, \text{ and } x_\Gamma \in \mathcal{X}.$$

   Now, consider the following formula:

$$\xi = \exists^N (x_{k+1}, x_{k+2}) \left( \phi_a(x_1, \ldots, x_{k+2}) \wedge \left( \bigwedge_{i=1}^k \left( \psi_{\tilde{C}_i}[x_i/x_{k+1}] \wedge \psi_{\hat{C}_i}[x_i/x_{k+2}] \right) \right) \right).$$

   It follows that $A, u_A$ and $B, u_B$ do not agree on the formula $\xi \in \mathsf{TC}_{k+2,n}$, which contradicts the assumption that $A, u_A$ and $B, u_B$ agree on all formulas in $\mathsf{TC}_{k+2,n}$. Therefore, $\chi_{k,A}^{(n)}(u_A) = \chi_{k,B}^{(n)}(u_B)$.

Hence, Equation (33) holds for all $t \in \mathbb{N}_0$, and we conclude:

$$A, u_A \underset{\mathsf{TC}_{k+2}}{\equiv} B, u_B \implies \chi_{k,A}^{(\infty)}(u_A) = \chi_{k,B}^{(\infty)}(u_B).$$

$\square$

C.5 RELATION BETWEEN GAME, LOGIC, AND $k$-CCWL

In this section, we show the correspondence between $\mathsf{TC}_{k+2}$, $k$-CCWL and the topological $(k+2)$-pebble game. Recall that 1-CCWL refines the color of a single cell by aggregating over its boundary, coboundary, lower, and upper neighborhood, and in Lemma C.7, we show that $\mathsf{GTC}_3$ refines 1-CCWL. Intuitively, For the base case $k = 1$, we slightly restrict $\mathsf{TC}_3$ to a guarded fragment $\mathsf{GTC}_3$ so that counting quantifiers only range over the same neighborhoods that 1-CCWL sees. In order to maintain the precise correspondence, we introduce the $\mathsf{GTC}_3$ game, as follows:

- If no pebble is currently placed on either side, Player I may choose any set $P_A \subseteq X_A \times X_A$ of pairs in $A$, as in the standard game.

- Otherwise, suppose for example that the first pebble $x_1$ is placed on a cell $a$ in $A$ (i.e. $u_A(x_1) = a$). Then Player I must choose $P_A$ consisting only of pairs $(b, c)$ such that one of the following holds:

  - $b = c$
  - $b \in \mathcal{N}_\downarrow(a)$ and $c \in \mathcal{N}_B(a \cap b)$,
  - $b \in \mathcal{N}_\uparrow(a)$ and $c \in \mathcal{N}_C(a \cap b)$.

  (The same condition holds symmetrically on $P_B$ using the current position of the corresponding pebble in $B$.)

**Lemma C.9.** *Let $A$ and $B$ be two attributed combinatorial complexes, and let $u_A : \{x_1, \ldots, x_k\} \to \mathcal{X}_A$ and $u_B : \{x_1, \ldots, x_k\} \to \mathcal{X}_B$ be two partial functions. If Player II has a winning strategy for the 0-move $\mathsf{TC}_k$ game on $A$ and $B$ with initial configuration $(u_A, u_B)$, then $A, u_A$ and $B, u_B$ agree on all quantifier-free formulas in $\mathsf{TC}_k$.*

*Proof of Lemma C.9.* We prove this lemma by induction on the length of the quantifier-free formula $\lambda \in \mathsf{TC}_k 0$, as follows:

1. Initial step: We prove that if Player II has a winning strategy for the 0-move $\mathsf{TC}_k$ game, then $A, u_A$ and $B, u_B$ agree on all atomic formulas $\lambda \in \mathsf{TC}_k$. We analyze each possible case for the atomic formula $\lambda$ separately:

   - If $\lambda = (x_i = x_j)$ for all $i, j \in [k]$: Since Player II has a winning strategy in the 0-move game, it means that $u_A(x_i) = u_A(x_j)$ implies $u_B(x_i) = u_B(x_j)$, and vice versa. Therefore, $A, u_A$ and $B, u_B$ agree on the formula $\lambda$.
   - If $\lambda = R_r(x_i)$ for some $r \in [\rho]_0$ and for all $i \in [k]$: Since Player II has a winning strategy in the 0-move game, it means $\mathrm{rk}_A(u_A(x_i)) = \mathrm{rk}_B(u_B(x_i))$ for all $i \in [k]$. Therefore, $A, u_A$ and $B, u_B$ agree on the formula $\lambda$.
   - If $\lambda = P_s(x_i)$ for some $s \in [\ell]$ and for all $i \in [k]$: Since Player II has a winning strategy in the 0-move game, it means $\mathrm{c}_A(u_A(x_i)) = \mathrm{c}_B(u_B(x_i))$ for all $i \in [k]$. Therefore, $A, u_A$ and $B, u_B$ agree on the formula $\lambda$.
   - If $\lambda = E^\mathcal{N}(x_i, x_j)$ for some $\mathcal{N} \in \{\mathcal{N}_B, \mathcal{N}_C, \mathcal{N}_\downarrow, \mathcal{N}_\uparrow\}$ and for all $i, j \in [k]$: Since Player II has a winning strategy in the 0-move game, it means that if $u_A(x_i) \in \mathcal{N}(A, u_A(x_j))$, then $u_B(x_i) \in \mathcal{N}(B, u_B(x_j))$, and vice versa. Therefore, $A, u_A$ and $B, u_B$ agree on the formula $\lambda$.

   Therefore, we conclude that $A, u_A$ and $B, u_B$ agree on the quantifier-free formula $\lambda$ with $\mathrm{len}(\lambda) = 1$.

2. Inductive step: Assume that if Player II has a winning strategy for the 0-move $\mathsf{TC}_k$ game, then $A, u_A$ and $B, u_B$ agree on all formulas $\lambda \in \mathsf{TC}_k$ with $\mathrm{len}(\lambda) < m$. We must show that it also holds for all formulas $\lambda \in \mathsf{TC}_k$ with $\mathrm{len}(\lambda) = m$. For contradiction, assume that $A, u_A \models \lambda$ but $B, u_B \not\models \lambda$. Each case for $\lambda$ is analyzed separately:

   - If $\lambda$ is of the form $\neg\omega$:

$$A, u_A \models \phi \implies A, u_A \not\models \omega,$$
$$B, u_B \not\models \phi \implies B, u_B \models \omega.$$

   Thus, $A, u_A$ and $B, u_B$ do not agree on $\omega$, which has $\mathrm{len}(\omega) < m$. This contradicts the induction hypothesis. Hence, $A, u_A$ and $B, u_B$ agree on $\lambda$.

- If $\phi$ is of the form $\omega_1 \wedge \omega_2$: Without loss of generality, we can assume $A, u \models \omega_1$ but $B, v \not\models \omega_1$. In this case, $A, u_A$ and $B, u_B$ do not agree on $\omega_1$, where $\text{len}(\omega_1) < m$. This again contradicts the induction hypothesis. Hence, $A, u_A$ and $B, u_B$ agree on $\lambda$.

Therefore, we have proved that if Player II has a winning strategy for the $0$-move $\mathsf{TC}_k$ game, then $A, u_A$ and $B, u_B$ agree on all quantifier-free formulas $\lambda \in \mathsf{TC}_k$. $\qquad\square$

**Theorem C.10.** *Let $A = (S_A, \mathcal{X}_A, \text{rk}_A, c_A)$ and $B = (S_B, \mathcal{X}_B, \text{rk}_B, c_B)$ be two attributed combinatorial complexes and let $u_A : \{x_1, \ldots, x_k\} \rightharpoonup \mathcal{X}_A$ and $u_B : \{x_1, \ldots, x_k\} \rightharpoonup \mathcal{X}_B$ be the partial functions. If Player II has a winning strategy for $\mathsf{TC}_k$ game with initial configuration $(u_A, u_B)$, then:*

$$A, u_A \underset{\mathsf{TC}_k}{\equiv} B, u_B$$

*Proof of Theorem C.10.* We prove this theorem by induction on the quantifier depth of the formula $\lambda \in \mathsf{TC}_k$. Specifically, we show that if Player II has a winning strategy for the $t$-move $\mathsf{TC}_k$ game, then $A, u_A$ and $B, u_B$ agree on all formulas $\lambda \in \mathsf{TC}_{k,t}$. We prove this by induction on $t$.

1. Initial step: We prove that if Player II has a winning strategy for the $0$-move $\mathsf{TC}_k$ game, then $A, u_A$ and $B, u_B$ agree on all quantifier-free formulas $\lambda \in \mathsf{TC}_k$. We prove this in Lemma C.9.

2. Inductive step: Assume that if Player II has a winning strategy for the $t$-move $\mathsf{TC}_k$ game for all $t \in [n-1]_0$, then $A, u_A$ and $B, u_B$ agree on all formulas $\lambda \in \mathsf{TC}_{k,t}$. We must show this also holds for $t = n$. For contradiction, assume that $A, u_A \models \lambda$ but $B, u_B \not\models \lambda$. The only case to consider is when $\lambda$ is of the form $\exists^N (x_i, x_j)\omega$ for some $i, j \in [k]$. We define the following set:

$$P_A = \left\{ (\alpha_A, \beta_A) \in \mathcal{X}_A^2 \,\middle|\, A, u_A[x_i \to \alpha_A, x_j \to \beta_A] \models \omega \right\}.$$

By assumption:

$$|P_A| \geq N > \left| \left\{ (\alpha_B, \beta_B) \in \mathcal{X}_B^2 \,\middle|\, B, u_B[x_i \to \alpha_B, x_j \to \beta_B] \models \omega \right\} \right|.$$

If Player I selects $P_A$, then Player II must respond with a set $P_B$. Regardless of which set $P_B$ Player II selects, there exists at least one pair of cells $(\alpha_B, \beta_B) \in \mathcal{X}_B^2$ such that

$$B, u_B[x_i \to \alpha_B, x_j \to \beta_B] \not\models \omega.$$

On the other hand, for all $(\alpha_A, \beta_A) \in P_A$, it holds that

$$A, u_A[x_i \to \alpha_A, x_j \to \beta_A] \models \omega.$$

If Player I modifies the partial function $u_B = u_B[x_i \to \alpha_B, x_j \to \beta_B]$, then Player II cannot have a winning strategy for the remainder of the game. By the induction assumption, $A, u_A[x_k \to \alpha_A, x_{k-1} \to \beta_A]$ and $B, u_B[x_k \to \alpha_B, x_{k-1} \to \beta_B]$ must agree on $\omega$, where $\text{len}(\omega) < n$. This contradicts the assumption that Player II has a winning strategy. Therefore, $A, u$ and $B, v$ agree on the formula $\lambda$ with $\text{len}(\lambda) = n$.

Thus, if Player II has a winning strategy for the $\mathsf{TC}_k$ game, then $A, u_A$ and $B, u_B$ agree on all formulas $\lambda \in \mathsf{TC}_k$. $\qquad\square$

*Remark C.11.* By the same approach used in the proofs of Lemma C.9 and Theorem C.10, one can show that if Player II has a winning strategy in the $\mathsf{GTC}_3$ game with initial configuration $(u_A, u_B)$, then we have

$$A, u_A \underset{\mathsf{GTC}_3}{\equiv} B, u_B$$

**Theorem C.12.** *Let $A = (S_A, \mathcal{X}_A, \text{rk}_A, c_A)$ and $B = (S_B, \mathcal{X}_B, \text{rk}_B, c_B)$ be two attributed combinatorial complexes. Let $u_A \in \mathcal{X}_A^k$ and $u_B \in \mathcal{X}_B^k$. If $\chi_{k,A}^{(\infty)}(u_A) = \chi_{k,B}^{(\infty)}(u_B)$, then Player II has a winning strategy for $\mathsf{TC}_{k+2}$ game with initial configuration $(u_A, u_B)$.*

*Proof of Theorem C.12.* We prove that for all $t \geq 0$, if $\chi_{k,A}^{(t)}(u_A) = \chi_{k,B}^{(t)}(u_B)$, then Player II has a winning strategy for the $t$-move $\mathsf{TC}_{k+2}$ game. We prove this by induction on $t$.

1. Initial step: We prove that if $\chi_{k,A}^{(t)}(u_A) = \chi_{k,B}^{(t)}(u_B)$, then Player II has a winning strategy for the 0-move $\mathsf{TC}_{k+2}$ game. Since $\chi_{k,A}^{(t)}(u_A) = \chi_{k,B}^{(t)}(u_B)$, it follows that $\mathrm{atp}_{A,k}(u_A) = \mathrm{atp}_{B,k}(u_B)$. This implies the following conditions hold:

   - $u_A(x_i) = u_A(x_j) \iff u_B(x_i) = u_B(x_j)$, for all $i, j \in [k]$,
   - $\mathrm{rk}_A(u_A(x_i)) = \mathrm{rk}_B(u_B(x_i))$, for all $i \in [k]$,
   - $c_A(u_A(x_i)) = c_B(u_B(x_i))$ for all $i \in [k]$, and
   - $u_A(x_i) \in \mathcal{N}(A, u_A(x_j)) \Leftrightarrow u_B(x_i) \in \mathcal{N}(B, u_B(x_j))$, for all $\mathcal{N} \in \{\mathcal{N}_B, \mathcal{N}_C, \mathcal{N}_\downarrow, \mathcal{N}_\uparrow\}$ and $i, j \in [k]$.

   Therefore, Player II has a winning strategy for the 0-move game.

2. Inductive step: Assume that if $\chi_{k,A}^{(t)}(u_A) = \chi_{k,B}^{(t)}(u_B)$, then Player II has a winning strategy for the $t$-move $\mathsf{TC}_{k+2}$ game for all $t < n$. We must show that this also holds for $t = n$. Note that Player I's strongest move is to modify an unmodified pair of variables. Without loss of generality, assume Player I modifies the pair $(x_{k+2}, x_{k+1})$. The game proceeds as follows:

   - Player I selects $N$ pairs of cells, denoted as $P_A$.
   - For each atomic type value $a$ and $k$-tuple of pairs of colors $\bar{C}$, let $N_{a,\bar{C}}$ be the number of pairs $(\alpha_A, \beta_A) \in P_A$ such that

   $$\left(\mathrm{atp}_{A,k+2}(u_A\alpha_A\beta_A), \Delta_{k,A}^{(n-1)}(u_A; \alpha_A, \beta_A)\right) = (a, \bar{C})$$

   Since $\chi_{k,A}^{(t)}(u_A) = \chi_{k,B}^{(t)}(u_B)$, it means that there are $N_{a,\bar{C}}$ pairs of cells $(\alpha_B, \beta_B) \in \mathcal{X}_B^2$, for all atomic type $a$ and $k$-tuple of pairs of colors $\bar{C}$, such that

   $$\left(\mathrm{atp}_{B,k+2}(u_B\alpha_B\beta_B), \Delta_{k,B}^{(n-1)}(u_B; \alpha_B, \beta_B)\right) = (a, \bar{C})$$

   Player II includes all these $N_{a,\bar{C}}$ pairs in $P_B$, for all atomic types $a$ and $k$-tuples of pairs of colors $\bar{C}$.
   - Player I modifies $u_B[x_{k+2} \to \alpha_B, x_{k+1} \to \beta_B]$ for some pair $(\alpha_B, \beta_B) \in P_B$.
   - Player II modifies $u_A[x_{k+2} \to \alpha_A, x_{k+1} \to \beta_A]$ for $(\alpha_A, \beta_A) \in P_A$ such that

   $$\left(\mathrm{atp}_{k+2}(u_A\alpha_A\beta_A), \Delta_{k,A}^{(n-1)}(u_A; \alpha_A, \beta_A)\right) = \left(\mathrm{atp}_{k+2}(u_B\alpha_B\beta_B), \Delta_{k,B}^{(n-1)}(u_B; \alpha_B, \beta_B)\right).$$

   - The remaining valuations are $u_A[x_{k+2} \to \alpha_A, x_{k+1} \to \beta_A]$ and $u_B[x_{k+2} \to \alpha_B, x_{k+1} \to \beta_B]$.

   Player II has not lost the game. In the next move, Player I modifies a valuation for a pair $(x_i, x_j)$. By the induction assumption, Player II has a winning strategy for the remainder of the game.

Thus, we have shown that for all $t \geq 0$, if $\chi_{k,A}^{(t)}(u_A) = \chi_{k,B}^{(t)}(u_B)$, then Player II has a winning strategy for the $t$-move $\mathsf{TC}_{k+2}$ game. $\qquad\square$

*Remark* C.13. One can use the similar approach as in the proof of Theorem C.12, and show that if $\chi_{1,A}^{(\infty)}(u_A) = \chi_{1,B}^{(\infty)}(u_B)$, then Player II has a winning strategy for $\mathsf{GTC}_3$ game with initial configuration $(u_A, u_B)$.

**Theorem C.14.** *The $k$-CCWL procedure and $\mathsf{TC}_{k+2}$ are equivalent in expressive power for distinguishing non-isomorphic uniform ACCs.*

*Proof.* Let $A = (S_A, \mathcal{X}_A, \mathrm{rk}_A, c_A)$ and $B = (S_B, \mathcal{X}_B, \mathrm{rk}_B, c_B)$ be two uniform ACCs. Recall that the $k$-CCWL *procedure* on $A$ and $B$ is defined by: augmenting each complex with a identical-vs-disjoint (a fresh 0-cell connected to all 0-cells via freshly colored 1-cells), running $k$-CCWL, and then comparing the stable global signatures $\chi_{k,A}^{(\infty)}$ and $\chi_{k,B}^{(\infty)}$. The procedure distinguishes $A$ and $B$ iff $\chi_{k,A}^{(\infty)} \neq \chi_{k,B}^{(\infty)}$.

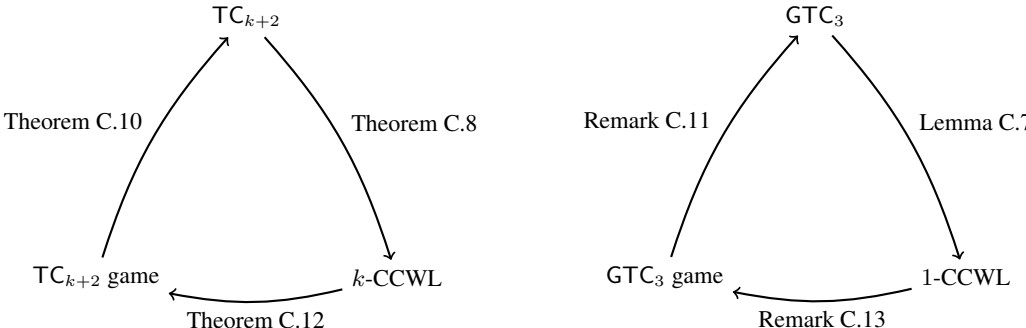

Figure 4: Left: correspondence between $k$-CCWL, $\mathsf{TC}_{k+2}$, and the $\mathsf{TC}_{k+2}$game. Right: the case $k = 1$ with guarded logic $\mathsf{GTC}_3$ and the $\mathsf{GTC}_3$ game.

By Proposition 3.1, applied to uniform ACCs with the identical-vs-disjoint, exactly one of the following holds:

$$\chi_{k,A}^{(\infty)} = \chi_{k,B}^{(\infty)} \quad \text{or} \quad \chi_{k,A}^{(\infty)} \cap \chi_{k,B}^{(\infty)} = \emptyset.$$

We show that these two cases are equivalent to agreement vs. disagreement on all sentences of $\mathsf{TC}_{k+2}$.

*Case 1:* $\chi_{k,A}^{(\infty)} = \chi_{k,B}^{(\infty)}$. Then the multisets of stable $k$-CCWL colors on $\mathcal{X}_A^k$ and $\mathcal{X}_B^k$ coincide. In particular, for every color $C$ the number of $k$-tuples of color $C$ is the same in $A$ and $B$. Hence we can fix a color-preserving bijection

$$f : \mathcal{X}_A^k \to \mathcal{X}_B^k \quad \text{with} \quad \chi_{k,A}^{(\infty)}(u_A) = \chi_{k,B}^{(\infty)}(f(u_A)) \text{ for all } u_A \in X_A^k.$$

By Corollary 5.3, equality of stable $k$-CCWL colors for tuples is equivalent to indistinguishability in $\mathsf{TC}_{k+2}$. Concretely, for every $u_A \in \mathcal{X}_A^k$, $u_B \in \mathcal{X}_B^k$,

$$\chi_{k,A}^{(\infty)}(u_A) = \chi_{k,B}^{(\infty)}(u_B) \iff \forall \varphi \in \mathsf{TC}_{k+2} : A, u_A \models \varphi \iff B, u_B \models \varphi.$$

Applying this to each pair $(u_A, f(u_A))$ we obtain

$$\forall u_A \in \mathcal{X}_A^k, \, \forall \varphi \in \mathsf{TC}_{k+2} : \quad A, u_A \models \varphi \iff B, f(u_A) \models \varphi.$$

Now let $\psi$ be any *sentence* of $\mathsf{TC}_{k+2}$ (no free variables). Since the truth of $\psi$ does not depend on the chosen valuation, for any $u_A$ we have $A \models \psi \iff A, u_A \models \psi$, and similarly for $B$. Taking any $u_A \in X_A^k$ and $u_B = f(u_A)$, we get

$$A \models \psi \iff A, u_A \models \psi \iff B, u_B \models \psi \iff B \models \psi.$$

Thus $A$ and $B$ satisfy exactly the same $\mathsf{TC}_{k+2}$ sentences, so $\mathsf{TC}_{k+2}$ cannot distinguish them whenever the $k$-CCWL procedure cannot.

*Case 2:* $\chi_{k,A}^{(\infty)} \neq \chi_{k,B}^{(\infty)}$. By Proposition 3.1, this implies $\chi_{k,A}^{(\infty)} \cap \chi_{k,B}^{(\infty)} = \emptyset$, i.e., no stable color occurs in both $A$ and $B$. Hence there exists a stable color $C$ and a $k$-tuple $u_A \in \mathcal{X}_A^k$ such that

$$\chi_{k,A}^{(\infty)}(u_A) = C \quad \text{and} \quad \chi_{k,B}^{(\infty)}(u_B) \neq C \text{ for all } u_B \in X_B^k.$$

Let $\equiv_{\mathsf{TC}_{k+2}}$ be the equivalence relation on $k$-tuples taken from $A$ and $B$ defined by

$$u_A \equiv_{\mathsf{TC}_{k+2}} u_B \iff \forall \varphi \in \mathsf{TC}_{k+2} : A, u_A \models \varphi \iff B, u_B \models \varphi,$$

for $u_A \in \mathcal{X}_A^k$ and $u_B \in \mathcal{X}_B^k$. By Corollary 5.3, this equivalence coincides exactly with equality of stable $k$-CCWL colors: for all $u_A \in \mathcal{X}_A^k$, $u_B \in \mathcal{X}_B^k$,

$$\chi_{k,A}^{(\infty)}(u_A) = \chi_{k,B}^{(\infty)}(u_B) \iff u_A \equiv_{\mathsf{TC}_{k+2}} u_B.$$

Fix a stable color $C$ that occurs in $A$ but not in $B$, and choose $u_A \in \mathcal{X}_A^k$ with $\chi_{k,A}^{(\infty)}(u_A) = C$. Consider the set

$$[u_A] := \{\, v_A \in \mathcal{X}_A^k \mid \chi_{k,A}^{(\infty)}(v_A) = C \,\} \cup \{\, v_B \in \mathcal{X}_B^k \mid \chi_{k,B}^{(\infty)}(v_B) = C \,\}.$$

By assumption, no tuple in $B$ has color $C$, so the second set is empty and

$$[u_A] = \{\, v_A \in \mathcal{X}_A^k \mid \chi_{k,A}^{(\infty)}(v_A) = C \,\}.$$

By the previous paragraph, all tuples in $[u_A]$ are mutually $\equiv_{\mathsf{TC}_{k+2}}$-equivalent, and no tuple in $B$ is equivalent to them. Thus $[u_A]$ is a single $\equiv_{\mathsf{TC}_{k+2}}$-equivalence class among the tuples of $A$ and $B$.

By standard finite-model-theoretic arguments, each $\equiv_{\mathsf{TC}_{k+2}}$-equivalence class is definable by some $\mathsf{TC}_{k+2}$-formula with $k$ free variables. Hence, there exists a formula $\varphi_C(x_1, \ldots, x_k) \in \mathsf{TC}_{k+2}$ such that, for every $D \in \{A, B\}$ and every $v \in X_D^k$,

$$D, v \models \varphi_C \quad \Longleftrightarrow \quad v \in [u_A] \quad \Longleftrightarrow \quad \chi_{k,D}^{(\infty)}(v) = C.$$

In particular, there exists $u_A \in \mathcal{X}_A^k$ with $A, u_A \models \varphi_C$, while for all $u_B \in \mathcal{X}_B^k$ we have $B, u_B \not\models \varphi_C$ because no tuple in $B$ has color $C$.

Consider now the $\mathsf{TC}_{k+2}$-sentence

$$\exists x_1 \ldots \exists x_k \, \varphi_C(x_1, \ldots, x_k).$$

By construction,

$$A \models \exists x_1 \ldots \exists x_k \, \varphi_C(x_1, \ldots, x_k) \quad \text{but} \quad B \not\models \exists x_1 \ldots \exists x_k \, \varphi_C(x_1, \ldots, x_k),$$

so $\mathsf{TC}_{k+2}$ distinguishes $A$ and $B$ whenever the $k$-CCWL procedure does. $\qquad \square$

$k$**-CCWL** $\preceq (k-1)$**-CCWL.** Now, we know that $k$-CCWL has the same expressive power as $\mathsf{TC}_{k+2}$, and $(k-1)$-CCWL has the same expressive power as $\mathsf{TC}_{k+1}$. It is also known that $\mathsf{TC}_{k+1} \subset \mathsf{TC}_{k+2}$, therefore, $k$-CCWL is at least as expressive as $(k-1)$-CCWL. Now, in the next section, we want to show that $k$-CCWL is strictly more expressive than $(k-1)$-CCWL in terms of distinguishing non-isomorphic ACCs.

# D    CCWL ON GRAPHS AND ITS RELATION TO WEISFEILER–LEMAN

In the following, we want to show that 2-CCWL has strictly more expressive power than 1-CCWL. In the proof of Lemma D.1, instead of running 2-CCWL on the pair of ACC, we use the result from Theorem C.14, and its corresponding logic to make the proof simpler, which provides a great example of a useful tool in terms of distinguishing ACCs.

**Lemma D.1** (1-CCWL < 2-CCWL). *There is a pair of ACCs such that* 1*-CCWL cannot distinguish them, but* 2*-CCWL can distinguish them.*

*Proof of Lemma D.1.* We use the standard example from the graph setting: the 6-cycle $C_6$ and the disjoint union of two triangles $C_3 \sqcup C_3$, which 1-WL fails to distinguish, see e.g. Grohe (2021).

We now view these graphs as 1-dimensional ACCs:

- On the left, the ACC $A$ obtained from the 6-cycle $C_6$, with 6 cells of rank 0 (the vertices) and 6 cells of rank 1 (the edges);

- On the right, the ACC $B$ obtained from $C_3 \sqcup C_3$, again with 6 cells of rank 0 and 6 cells of rank 1.

In both cases, there are no cells of rank $\geq 2$.

Intuitively, on these ACCs, 1-CCWL behaves exactly like 1-WL on the underlying graphs. In both ACCs, each 0-cell $v$, aggregates colors exactly from two 0-rank cells and two 1-rank cells. Since initially all 0-cells share the same color and all 1-cells share the same color, every 0-cell sees the

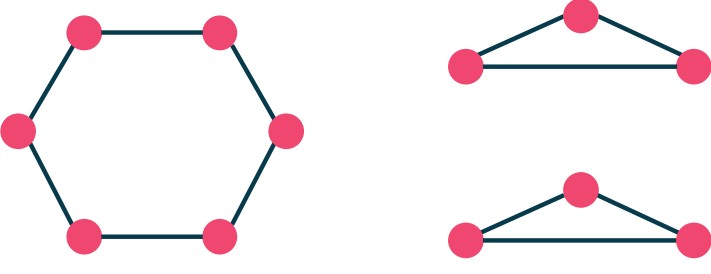

Figure 5: Left: the ACC associated with the 6-cycle $C_6$. Right: the ACC associated with the disjoint union $C_3 \sqcup C_3$. Both have six 0-cells, but only the right-hand side contains a 3-cycle (triangle).

same multiset of colors. Dually, each 1-cell $e$ has two 0-rank cells in its boundary and two 1-rank cells in its lower neighborhood, so all 1-cells also see the same multiset. Hence one refinement step already yields a stable coloring that is identical on the two ACCs, and 1-CCWL cannot distinguish them.

As a result, after one refinement step the coloring on both ACCs is already stable and identical, and 1-CCWL cannot distinguish them. Let us also note that adding the broadcast-anchor cell used in our TNN constructions does not affect this argument. If we attach the same anchor cell to $A$ and $B$ in the same way, then all cells in $A$ and $B$ gain the same additional neighbor of the same color. Hence the symmetry of the refinement process is preserved, and 1-CCWL still cannot distinguish the two ACCs.

However, consider the following formula $\phi \in \mathsf{TC}_4$:

$$\phi \;=\; \exists^1(x_1, x_2)\Big(R_1(x_1) \wedge R_1(x_2) \wedge E^{\mathcal{N}_\downarrow}(x_1, x_2) \wedge$$
$$\exists^1(x_3, x_4)\big(R_1(x_3) \wedge R_1(x_4) \wedge E^{\mathcal{N}_\downarrow}(x_3, x_4) \wedge E^{\mathcal{N}_\downarrow}(x_1, x_3) \wedge x_2 = x_4\big)\Big).$$

Informally, this formula expresses the existence of three rank-1 cells (edges) that are pairwise adjacent via their lower neighborhoods, i.e., the presence of a triangle. In the ACC $B$ (two triangles), this property holds. In the ACC $A$ (6-cycle), there is no triangle at all, so no such quadruple $(x_1, x_2, x_3, x_4)$ exists and $A \not\models \phi$. Moreover, we can make an adjustment to $\phi$ and create a new formula $\phi'$ that expresses the existence of at least two such triangles:

$$\phi' \;=\; \exists^{12}(x_1, x_2)\Big(R_1(x_1) \wedge R_1(x_2) \wedge E^{\mathcal{N}_\downarrow}(x_1, x_2) \wedge$$
$$\exists^1(x_3, x_4)\big(R_1(x_3) \wedge R_1(x_4) \wedge E^{\mathcal{N}_\downarrow}(x_3, x_4) \wedge E^{\mathcal{N}_\downarrow}(x_1, x_3) \wedge x_2 = x_4\big)\Big).$$

One might worry that adding the identical-vs-disjoint cell to $A$ could create triangles. However, as discussed before, the anchor cell and all edges involving it are added with a fresh color, so they are easily recognizable. In particular, we can use the predicate $P_s(\cdot)$ to restrict attention to non-fresh edges and thus ignore any triangles that use the anchor. Therefore, $\phi$ still separates $A$ and $B$ even in the presence of the identical-vs-disjoint cell.

Thus, $A$ and $B$ disagree on a $\mathsf{TC}_4$-sentence. Therefore, 2-CCWL is able to distinguish $A$ and $B$. Hence, 1-CCWL cannot distinguish them, this proves that 1-CCWL is strictly less expressive than 2-CCWL. $\qquad\square$

Given the above lemma, it seems that $k$-WL and $k$-CCWL have similar behavior in terms of distinguishability of pairs of graphs, if we only consider nodes as 0-rank cells and edges as 1-rank cells. To formalize this assumption, Let $G = (V_G, E_G, c_G)$ be a vertex-colored simple graph. We associate to $G$ a 1-dimensional uniform ACC

$$A = G^{\mathrm{CC}} := (S_A, \mathcal{X}_A, \mathrm{rk}_A, c_A)$$

as follows:

- $S_A := V_G$;
- $\mathcal{X}_A(0) := \{\{v\} : v \in V_G\}$,
- $\mathcal{X}_A(1) := \{\{u, v\} : \{u, v\} \in E_G\}$,
- $\mathcal{X}_A := X_A(0) \cup X_A(1)$, and $\mathrm{rk}_A(\{v\}) = 0$, $\mathrm{rk}_A(\{u, v\}) = 1$,
- $c_A(\{v\}) := c_G(v)$, and all 1-cells receive the same fresh color $c_{\mathrm{edge}}$.

**Theorem D.2.** *Let $G, H$ be vertex-colored graphs, and let $A := G^{\mathrm{CC}}$ and $B := H^{\mathrm{CC}}$ be their associated 1-dimensional ACCs. Then, for a fix $k \geq 1$, we have*

$$\left\{\!\!\left\{ \mathrm{wl}_{k,G}^{(t)}(v_G) : v_G \in V_G^k \right\}\!\!\right\} = \left\{\!\!\left\{ \mathrm{wl}_{k,H}^{(t)}(v_H) : v_H \in V_H^k \right\}\!\!\right\} \tag{34}$$

*if and only if*

$$\left\{\!\!\left\{ \chi_{k,A}^{(t)}(x_A) : x_A \in \mathcal{X}_A^k \right\}\!\!\right\} = \left\{\!\!\left\{ \chi_{k,B}^{(t)}(x_B) : x_B \in \mathcal{X}_B^k \right\}\!\!\right\}. \tag{35}$$

*Proof of Theorem D.2.* We follow the following steps to prove the theorem.

- Step 1: Let $A = (S_A, \mathcal{X}_A, \mathrm{rk}_A, c_A)$ be the associated 1-dimensional ACC of the graph $G$. We define the incidence graph $A^{\mathrm{inc}}$ of $A$ as follows:

  - $V_{A^{\mathrm{inc}}} := V_G \,\dot\cup\, E_G$. We consider a new vertex type $v^e$ for each edge $e = \{u, v\} \in E_G$.
  - $E_{A^{\mathrm{inc}}} := E_G \cup \big\{\{u, e\} : e = \{u, v\} \in E_G, \ u \in e\big\}$.
  - Vertex colors: each $v \in V_G$ keeps its original color but is tagged as "vertex"; each $e \in E_G$ receives a fresh color tagged as "edge", so vertex- and edge-nodes are distinguishable.

  We define $B^{\mathrm{inc}}$ in the same way.

- Step 2: We claim that for every $t \geq 0$, we have

$$\left\{\!\!\left\{ \mathrm{wl}_{k,A^{\mathrm{inc}}}^{(t)}(v) : v \in V_{A^{\mathrm{inc}}}^k \right\}\!\!\right\} = \left\{\!\!\left\{ \mathrm{wl}_{k,B^{\mathrm{inc}}}^{(t)}(u) : u \in V_{B^{\mathrm{inc}}}^k \right\}\!\!\right\}$$

$$\Updownarrow$$

$$\left\{\!\!\left\{ \chi_{k,A}^{(t)}(x) : x \in \mathcal{X}_A^k \right\}\!\!\right\} = \left\{\!\!\left\{ \chi_{k,B}^{(t)}(y) : y \in \mathcal{X}_B^k \right\}\!\!\right\}.$$

  We prove this by induction on $t \geq 0$.

  1. Initial step: We prove that the theorem holds for $t = 0$. The $k$-WL atomic type $\mathrm{atp}_{k,A^{\mathrm{inc}}}(v)$ records:
     - the equality pattern among the $v_i$'s,
     - which pairs $(v_i, v_j)$ are adjacent in $A^{\mathrm{inc}}$, and
     - the initial vertex colors $c_{A^{\mathrm{inc}}}(v_i)$.

     The $k$-CCWL atomic type $\mathrm{atp}_{k,A}(x)$ records:
     - the equality pattern among the $x_i$'s,
     - the truth values of $\mathcal{N}_B, \mathcal{N}_C, \mathcal{N}_\downarrow, \mathcal{N}_\uparrow$ between $x_i, x_j$, and
     - the initial cell colors $c_A(x_i)$.

     Moreover, we know that $x_i \in \mathcal{N}_\uparrow(x_j)$ if and only if $\{v_i, v_j\} \in E_G$. Therefore, we can define a bijective function $F : \mathcal{X}_A^k \cup \mathcal{X}_B^k \to V_{A^{\mathrm{inc}}}^k \cup V_{B^{\mathrm{inc}}}^k$ such that for all $x_A \in \mathcal{X}_A^k$, we have $F(x_A) = v = (v_1, \ldots, v_k)$ where $v_i = v^e$ if $x_i = e = \{p, q\} \in E_G$ and $v_i = p$

if $x_i = \{p\} \in V_G$. Same also holds for all $x_B \in \mathcal{X}_B^k$. Therefore, for all $x_A \in \mathcal{X}_A^k$ and $x_B \in \mathcal{X}_B^k$, let $v = F(x_A)$ and $u = F(x_B)$, we have

$$
\begin{aligned}
\mathrm{wl}_{k,A^{\mathrm{inc}}}^{(0)}(v) = \mathrm{wl}_{k,B^{\mathrm{inc}}}^{(0)}(u) &\iff \mathrm{atp}_{k,A^{\mathrm{inc}}}(v) = \mathrm{atp}_{k,B^{\mathrm{inc}}}(u) \\
&\iff \mathrm{atp}_{k,A}(x_A) = \mathrm{atp}_{k,B}(x_B) \\
&\iff \chi_{k,A}^{(0)}(x_A) = \chi_{k,B}^{(0)}(x_B).
\end{aligned}
$$

Therefore, we have

$$
\left\{\!\!\left\{ \chi_{k,A}^{(0)}(x) : x \in \mathcal{X}_A^k \right\}\!\!\right\} = \left\{\!\!\left\{ \chi_{k,B}^{(0)}(y) : y \in \mathcal{X}_B^k \right\}\!\!\right\}
$$

if and only if

$$
\left\{\!\!\left\{ \mathrm{wl}_{k,A^{\mathrm{inc}}}^{(0)}(v) : v \in V_{A^{\mathrm{inc}}}^k \right\}\!\!\right\} = \left\{\!\!\left\{ \mathrm{wl}_{k,B^{\mathrm{inc}}}^{(0)}(u) : u \in V_{B^{\mathrm{inc}}}^k \right\}\!\!\right\}.
$$

2. Inductive step: Assume the statement holds at round $t-1$. We compare the refinement multisets used by $k$-WL and $k$-CCWL on incidence graphs and 1-ACCs, [3].

   For $k$-WL, the refinement of $v$ at round $t$ is a hash of $\mathrm{wl}_{k,A^{\mathrm{inc}}}^{(t-1)}(v)$ together with the multiset

$$
M_{k,A^{\mathrm{inc}}}^{(t)}(v) := \left\{\!\!\left\{ (\mathrm{atp}_{k+1,A^{\mathrm{inc}}}(\bar{v}[v_i \leftarrow u]), \langle \mathrm{wl}_{k,A^{\mathrm{inc}}}^{(t-1)}(\bar{v}[v_i \leftarrow u])\rangle_{i=1}^k) : u \in V_{A^{\mathrm{inc}}} \right\}\!\!\right\},
$$

   For $k$-CCWL on $A$, the refinement of $x$ at round $t$ is a hash of $\chi_{k,A}^{(t-1)}(x)$ and the multiset

$$
D_{k,A}^{(t)}(x) := \left\{\!\!\left\{ (\mathrm{atp}_{k+2,A}(\bar{x}_{\alpha\beta}), \langle(\chi_{k,A}^{(t-1)}(\bar{x}[\alpha/i]), \chi_{k,A}^{(t-1)}(\bar{x}[\beta/i]))\rangle_{i=1}^k) : \alpha, \beta \in \mathcal{X}_A \right\}\!\!\right\},
$$

   Same as the base of induction, for all $x_A \in \mathcal{X}_A^k$ and $x_B \in \mathcal{X}_B^k$, let $v = F(x_A)$ and $u = F(x_B)$, we have

$$
\begin{aligned}
M_{k,A^{\mathrm{inc}}}^{(t)}(v) = M_{k,B^{\mathrm{inc}}}^{(t)}(u) &\iff D_{k,A}^{(t)}(x_A) = D_{k,B}^{(t)}(x_B) \\
&\iff \chi_{k,A}^{(t)}(x_A) = \chi_{k,B}^{(t)}(x_B).
\end{aligned}
$$

   Therefore, by the induction hypothesis, we have

$$
\left\{\!\!\left\{ \chi_{k,A}^{(t)}(x) : x \in \mathcal{X}_A^k \right\}\!\!\right\} = \left\{\!\!\left\{ \chi_{k,B}^{(t)}(y) : y \in \mathcal{X}_B^k \right\}\!\!\right\}
$$

   if and only if

$$
\left\{\!\!\left\{ \mathrm{wl}_{k,A^{\mathrm{inc}}}^{(t)}(v) : v \in V_{A^{\mathrm{inc}}}^k \right\}\!\!\right\} = \left\{\!\!\left\{ \mathrm{wl}_{k,B^{\mathrm{inc}}}^{(t)}(u) : u \in V_{B^{\mathrm{inc}}}^k \right\}\!\!\right\}.
$$

- Step 3: We now show that $k$-WL on $G$ and on $A^{\mathrm{inc}}$ have the same distinguishing power. Intuitively, $A^{\mathrm{inc}}$ is obtained from $G$ by *adding* a vertex per edge, connecting to its endpoint and coloring the new edge-vertices by a fresh color. Conversely, $G$ can be recovered uniquely from $A^{\mathrm{inc}}$ by restricting to the "vertex-type" nodes.

- Step 4: Combining Step 2 and Step 3, we obtain for all $t \geq 0$:

$$
\begin{aligned}
&\left\{\!\!\left\{ \mathrm{wl}_{k,G}^{(t)}(v_G) : v_G \in V_G^k \right\}\!\!\right\} = \left\{\!\!\left\{ \mathrm{wl}_{k,H}^{(t)}(v_H) : v_H \in V_H^k \right\}\!\!\right\} \\
\iff &\left\{\!\!\left\{ \mathrm{wl}_{k,A^{\mathrm{inc}}}^{(t)}(z) : z \in V_{A^{\mathrm{inc}}}^k \right\}\!\!\right\} = \left\{\!\!\left\{ \mathrm{wl}_{k,B^{\mathrm{inc}}}^{(t)}(z') : z' \in V_{B^{\mathrm{inc}}}^k \right\}\!\!\right\} \\
\iff &\left\{\!\!\left\{ \chi_{k,A}^{(t)}(x_A) : x_A \in \mathcal{X}_A^k \right\}\!\!\right\} = \left\{\!\!\left\{ \chi_{k,B}^{(t)}(x_B) : x_B \in \mathcal{X}_B^k \right\}\!\!\right\},
\end{aligned}
$$

which is exactly the statement of Theorem D.2.

$\square$

---

[3]The case $k = 1$ can be handled similarly; therefore, we only consider $k \geq 2$.

In the following, we want to generalize the result in Lemma D.1 and prove

**Theorem D.3** $((k-1)$-CCWL $< k$-CCWL$)$**.** *For any $k \geq 1$, there is a pair of ACCs such that $(k-1)$-CCWL cannot distinguish them, but $k$-CCWL can distinguish them.*

*Proof of Theorem D.3.* We use the result that is found in Cai et al. (1992).

**Theorem D.4** (Cai et al. (1992))**.** *For any $k \geq 1$, there is a pair of graphs $G_k$ and $H_k$ such that $k$-WL can distinguish them, but $(k-1)$-WL cannot distinguish them.*

Now, as a direct result of Theorem D.4 and Theorem D.2, we can conclude that for any $k \geq 2$, there is a pair of ACCs $G_k^{\text{CC}}$ and $H_k^{\text{CC}}$ such that $k$-CCWL can distinguish them, but $(k-1)$-CCWL cannot distinguish them. □

