# OpenReview forum: "The Logical Expressiveness of Topological Neural Networks"
_ICLR.cc/2026/Conference — ICLR 2026 Poster_

### Official Review · Reviewer_6rEA · 2025-10-26

**Soundness:** 3
**Presentation:** 3
**Contribution:** 2
**Rating:** 4
**Confidence:** 4

**Summary:**

This paper develops the first formal theory describing how expressive Topological Neural Networks (TNNs) are. It introduces the Combinatorial Complex Weisfeiler–Leman (k-CCWL) test, the Topological Counting Logic ($TC_k$) with a new pairwise counting quantifier, and a Topological Pebble Game. The main result proves that $(k-)$CCWL $\equiv$ TC$_{k+2}$​ $\equiv$ Topological $(k+2)$-Pebble Game, creating a unified logic–game–algorithm framework that precisely characterizes the expressive power of TNNs and shows how it extends beyond classical graph neural networks.

**Strengths:**

1. The introduction of $TC_k$ ​ with a pairwise counting quantifier  $\exists^{\geq N}(x_i, x_j)$ tailored to the upper/lower adjacency flows in TNNs is novel and well-motivated by message passing that aggregates over intermediary face/co-face pairs. This is a clean, logic-first abstraction of a topological operation.
2. Establishing expressiveness bounds for TNNs on combinatorial complexes is timely and important: it clarifies when and why TNNs can surpass GNNs and gives a yardstick for future TNN variants (i.e., with persistence, sheaves, or equivariance). The "pairwise-counting view" should inform model design and benchmarking going forward.

**Weaknesses:**

1. **Tightness of $TC_{k+2}$:** The $+2$ variable overhead is plausible but not yet proved tight. Add a lower-bound separation ($TC_{k+1}$ ​ vs. $TC_{k+2}$) or, minimally, a conjecture plus partial evidence (i.e., a candidate pair indistinguishable by $TC_{k+1}$).
2. The broadcast-anchor argument and "identical-vs-disjoint" global signatures rely on uniform ACCs. Please either: (1) show the theorem under weaker conditions, or (2) give a counterexample and make the limitation prominent (incl. a discussion of how non-uniform ACCs appear in practice).
3. **Algorithmic complexity \& stabilization bounds:** Provide explicit bounds for: (1) Per-round cost of $k-CCWL$ with the double-shift ($O(|X|^{k+2}$)?); (2) Number of rounds to stabilization in terms of $|X|, \rho, k$.
4. Can you please provide 1 or 2 toy ACC pairs and explicit $TC_{k+2}$ ​ formulas that $k$-CCWL detects but $k$-WL on graphs cannot. This would concretely demonstrate the step from vertex-adjacency to topological relations.

**Questions:**

1. **Why the "+2 variables" gap is tight:** You prove $k$-CCWL $\equiv$ $TC_{k+2}$ $\equiv$ topological $(k+2)$-pebble game. Please provide a separation example showing $TC_{k+1}$ ​ is insufficient - i.e., a pair of ACCs indistinguishable by $TC_{k+1}$ ​but separated by $k-CCWL/TC_{k+2}$. This would make the +2 blow-up not just sufficient but necessary (currently this is motivated by the "double shift" and pairwise counting intuition.)
2. **Scope: which TNN families are covered exactly?** Equivalences are stated for "general message-passing TNNs" with injective aggregators and the four adjacencies $N_B ​ ,N_C ​ ,N_{\uparrow} ​, N_{\downarrow}$. Please clarify which concrete architectures (i.e., CWN, TopoTune variants, sheaf-style updates, etc.) are exactly simulated by $k$-CCWL, and which require extra assumptions (i.e., uniformity, binary labels, injectivity). A small coverage table would help practitioners map models to the theory.
3. **Uniform ACC assumption in Prop./Thm. 3.1:** The "broadcast anchor" gadget yields the identical-vs-disjoint global signature property for uniform ACCs. Is uniformity essential? Please give a counterexample without uniformity or relax the condition (i.e., "every non-0 cell has $\ge$ 1 facet") and state the minimal requirement.
4. **Complexity and convergence of $k$-CCWL:** (1) What is the per-iteration complexity in $|X|, k$, and the maximum rank $\rho$, given the "double shift" considers all $(\alpha, \beta) ⁣ \in ⁣ X^2$?; (2) Do you prove a polynomial stabilization bound (as for $k$-WL on graphs) for ACCs? If so, please state it; if not, clarify whether only finite convergence is established.
5. **Game rules vs. pairwise quantification:** In the topological pebble game, Spoiler picks sets of pairs $P \subseteq X^2$ . Please add a short completeness sketch that shows every $TC_k$ ​formula with a pairwise quantifier translates to a finite-round Spoiler strategy with $k$ pebbles (and conversely), making the alignment fully explicit (beyond the proof sketch).
6. **Expressiveness vs. standard $k$-WL baselines:** Since $C_{k+1} ​\equiv $k$-WL$ on graphs, the $+2$ variable shift for ACCs is intuitive. Please add a diagram/table contrasting: $k-WL \leftrightarrow C_{k+1}$​ (graphs) vs. $k-CCWL \leftrightarrow TC_{k+2}$ ​(ACCs), plus a simple ACC "witness" where $k-CCWL$ strictly dominates $k-WL$.
7. **Worked micro-examples:** The running example in Fig. 1 (2-CCWL on a triangle with ranks/colors) is helpful - please add 1-2 non-trivial pairs of ACCs separated by 2-CCWL but not by 1-CCWL, with the corresponding $TC_3$ ​formulas and two-pebble strategies spelled out.

---

> ### Author Response · Authors · 2025-11-23
> **Part 1/3**
>
> We thank the reviewer for the positive assessment of our main ideas, and the many constructive suggestions. We have revised the manuscript accordingly and, below, we address each comment in detail.
>
>
> > Question 1. Why is the $+2$ variables gap tight? You prove $k$-CCWL $\equiv TC\_{k+2} \equiv$ topological $(k{+}2)$-pebble game. Please provide a separation example showing that $TC\_{k+1}$ is insufficient, i.e., a pair of ACCs indistinguishable by $TC\_{k+1}$ but separated by $k$-CCWL / $TC\_{k+2}$. This would make the $+2$ blow-up not just sufficient but also necessary (beyond the ``double shift'' and pairwise counting intuition).[Related to Weakness 1]
>
> _Why is the "+2 variables" gap tight?_ Thanks for your question. In the revised version, we now show that the "+2 variable" gap is not only sufficient but also necessary. In Section D, we first define 1-dimensional ACC of a graph, denoted as $A = G^{\mathrm{CC}}$. Then, we prove that $k$-CCWL on $G^{\mathrm{CC}}$ has the same expressivity power as $k$-WL on $G$. Basically, we show:
>
> $$
> \text{$k$-WL distinguishes $G$ and $H$} \iff \text{$k$-CCWL distinguishes $G^{\mathrm{CC}}$ and $H^{\mathrm{CC}}$}.
> $$
>
> We can now use a result by Cai et al. (1992), which shows that for any $k \geq 1$, there is a pair of graphs $G_k$ and $H_k$ such that $k$-WL can distinguish, but $(k-1)$-WL cannot. Therefore, we can use $G_k^{\mathrm{CC}}$ and $H_k^{\mathrm{CC}}$  to obtain the separation we are looking for, i.e., that **$k$-CCWL \ $TC_{k+2}$ is strictly more powerful than $(k-1)$-CCWL** \ $TC_{k+1}$. This is now established in Theorem D.3 of the revised manuscript.
>
> > Question 2. Scope: Which TNN families are covered exactly? The equivalences are stated
> for ``general message-passing TNNs'' with injective aggregators and the four adjacencies
> $N_B,N_C,N_\uparrow,N_\downarrow$. Please clarify which concrete architectures
> (e.g., CWN, TopoTune variants, sheaf-style updates, etc.) are exactly simulated by $k$-CCWL, and which require extra assumptions (uniformity, binary labels, injectivity). A small coverage table would help practitioners map models to the theory.
>
> Thanks for your question. TNNs on cell complexes were originally introduced as higher-order generalizations of message-passing networks. Consequently, our construction (1-CCWL) subsumes the expressive power of most TNN families proposed to date. This includes CW Networks and their Cell Isomorphism Network variant (CIN) [Bodnar et al., 2021(a)], message-passing simplicial networks [Bodnar et al., 2021(B)], and higher-order GNNs such as UniGCN and UniGIN [Huang et al., 2021], among many others, summarized in [Papillon et al., 2024].
>
> Importantly, all these architectures operate on topological domains that can be modeled as uniform combinatorial complexes. However, only a subset of them employs injective aggregation and update functions. For those architectures, 1-CCWL provides an upper bound on expressivity. In particular, for models with injective aggregators, such as the Cell Isomorphism Network (CIN), 1-CCWL matches their power.
> We emphasize this in our revised manuscript.

---

> ### Author Response · Authors · 2025-11-23
> **Part 2/3**
>
> > Question 3. Uniform ACC assumption in Prop./Thm.~3.1: The ``broadcast anchor'' gadget
> yields the identical-vs-disjoint global signature property for uniform ACCs. Is uniformity essential? Please give a counterexample without uniformity or a relaxation of the
> condition (e.g., every non-0 cell has at least one facet) together with the minimal
> requirement. [Related to Weakness 2]
>
> We thank the reviewer for raising this point.
> **Regarding a weaker condition.** Indeed, we can relax the uniformity condition. Uniformity was used to ensure a “flow of information’’ between every cell and the fresh rank-$0$ anchor cell. Technically, uniformity ensures that, for every cell $c$, there exists a finite sequence
>
> $$
> c = x_0, x_1, x_2, \dots, x_m
> $$
>
> such that each $x_i$ lies in one of the CCWL neighborhoods of $x_{i+1}$ (boundary, coboundary, lower or upper), and the sequence ends at a $0$-rank cell. Along such a path, CCWL refinement can propagate information from $c$ to the broadcast-anchor and back.
>
> **The uniformity assumption in Proposition 3.1 is not just a technical convenience:** as can be seen from the proofs of Lemmas B.3 and B.4, we only obtain the identical-vs-disjoint global signature property because information can flow from the fresh “broadcast anchor’’ cells to all other cells (and back) along adjacency chains. Uniformity guarantees exactly this kind of incidence connectivity. In the revised version we make this explicit and, in Appendix B (Example B.8 and Fig. 3), we now give a concrete counterexample showing that dropping uniformity can break the result: we construct two non-uniform ACCs whose 2-cells have empty boundary (none of their 1-faces are present in the complex), so these 2-cells never interact with the rest of the structure. Therefore, the 1-skeletons stabilize to different colors (triangle vs path), while the 2-cells remain indistinguishable, and the final colorings are neither identical nor disjoint. This shows that some structural assumption beyond “arbitrary ACC’’ is necessary.
>
> Based on these, in our revised manuscript, we provide an alternative definition of uniformity in an ACC (see Background in the updated PDF) such that for each cell, there should be a sequence of adjacent cells that ends to a $0$-rank cell. Importantly, this aligns with the sufficient and necessary conditions we need to prove our results.
>
> >Question 4. Complexity and convergence of $k$-CCWL: (1) What is the per-iteration
> complexity in $|X|$, $k$, and the maximum rank $\rho$, given that the double shift considers all
> $(\alpha,\beta)\in X^2$? (2) Do you prove a polynomial stabilization bound (as for $k$-WL on
> graphs) for ACCs? If so, please state it; if not, clarify whether you only establish finite
> convergence. [Related to Weakness 3]
>
> We thank the reviewer for this suggestion. In the revised version (Section B), we now state these bounds explicitly along with a proof.
>
> In our setting, $k$-CCWL is always guaranteed to converge, for the same reason as the classical $k$-WL algorithm on finite graphs. In all algorithmic parts of the paper (CCWL and $k$-CCWL), we consider finite ACCs, i.e., the underlying cell set $X$ is finite. We will make this finiteness assumption explicit in the revised version. For clarity, we have added Lemma B.1 in the Appendix, which shows that on any finite ACC the $k$-CCWL refinement sequence stabilizes and $\chi_k^{(\infty)}$ is well-defined.
>
> Let $X$ be the set of cells, $|X|=n$, and let $k$ be fixed.
>
> **Per-round cost.** By definition, in each refinement round $t \to t{+}1$ we iterate over all $k$-tuples $x \in X^k$ and, for each $x$, range over all $(\alpha,\beta) \in X^2$ in the double-shift. A straightforward implementation, therefore, has worst-case cost
> $$
> O\bigl(|X|^k \cdot |X|^2\bigr) = O\bigl(n^{k+2}\bigr)
> $$
> per round, which we now make explicit in the paper.
>
> **Rounds to stabilization.** As in standard $k$-WL, each non-trivial round strictly increases the number of distinct colors on $X^k$, and there are at most $|X|^k$ tuples. Hence, the number of rounds to stabilization satisfies
> $$
> t_{stab} < |X|^k.
> $$
> Putting this together gives a worst-case bound
> $$
> \mathrm{TIME}(k\text{-CCWL})
> \leq O\bigl(T \cdot n^{k+2}\bigr)
> \subseteq O\bigl(n^{2k+2}\bigr),
> $$
> with space complexity $O(n^k)$ to store tuple colors. In addition, we need the space for the ACC itself (incidence, ranks, attributes), which is at most quadratic in $n$ for fixed rank. Thus, the overall space complexity is $O(n^k)$ for fixed $k$.

---

> ### Author Response · Authors · 2025-11-23
> **Part 3/3**
>
> > Question 5. Game rules vs. pairwise quantification: In the topological pebble game, Spoiler picks sets of pairs $P \subseteq X^2$. Please add a short completeness sketch showing that every $TC\_{k}$ formula with a pairwise quantifier translates to a finite-round Spoiler strategy with $k$ pebbles (and conversely), making the alignment fully explicit beyond the current proof sketch.
>
> We thank the reviewer for this helpful suggestion. In the current version, the completeness of the topological pebble game with respect to $TC\_{k+2}$ is already proved in Appendix C, but we agree that the link to pairwise quantification can be made more explicit.
> In fact, Theorem C.10 shows that if Player II has a winning strategy in the topological $(k{+}2)$-pebble game, then the two ACCs satisfy exactly the same $TC\_{k+2}$ formulas. Equivalently, by contraposition, if there exists a formula $\phi \in TC\_{k+2}$ that separates two ACCs, then Player I has a finite-round winning strategy in the game with $k{+}2$ pebbles, which corresponds to $\phi$.
>
> Moreover, in Theorems C.8 and C.12, we can start from a winning strategy for Player I in the topological $(k{+}2)$-pebble game, and construct a separating formula $\phi\in TC\_{k+2}$. Taken together, these results give exactly the two-way correspondence the reviewer asks for: every $TC\_{k+2}$ formula (with pairwise counting) induces a finite-round Spoiler strategy with $k{+}2$ pebbles, and every such strategy gives a separating $TC\_{k+2}$ formula.
>
> In the revised version, we will add a short paragraph in Appendix C summarizing these implications explicitly in terms of “formula $\leftrightarrow$ strategy”. In addition, we now include an toy example (Example 5.5 / Fig. 2), which is a pair of ACCs that are $2$-CCWL distinguishable, and we show that they are separated by $2$-CCWL by (i) giving an explicit $\phi \in TC\_{4}$ and (ii) spelling out a winning strategy for Player I in the topological $4$-pebble game that directly corresponds to $\phi$.
>
> > Question 6. Expressiveness vs. standard $k$-WL baselines: Since $C\_{k+1} \equiv k$-WL on graphs, the $+2$ variable shift for ACCs is intuitive. Please add a diagram/table contrasting $k$-WL $\leftrightarrow C\_{k+1}$ (graphs) vs. $k$-CCWL $\leftrightarrow TC\_{k+2}$ (ACCs), plus a simple ACC +witness_ where $k$-CCWL strictly dominates $k$-WL. [Related to Weakness 4]
>
> We thank the reviewer for this helpful suggestion. In the revised version, we have added a small diagram in the introduction that explicitly contrasts the two correspondences  $k$-WL $\leftrightarrow C\_{k+1}$ (for graphs) and $k$-CCWL $\leftrightarrow TC\_{k+2}$ (for ACCs), and make the $+2$ variables'' shift visually clear. In addition, we now include Figure 2 to show a toy pair of graphs that $1$-WL cannot distinguish. Their associated $1$-dimensional ACCs are also indistinguishable by $1$-CCWL. However, their $2$-dimensional ACCs (adding minimal cycles as $2$-cells) can be distinguished $1$-CCWL / $TC_{3}$: in one complex, the lift creates two $5$-cycles, while in the other it creates two $4$-cycles. This example concretely shows how moving from pure vertex adjacency to topological relations on ACCs strictly increases expressiveness beyond standard graph $k$-WL.
>
> > Question 7. Worked micro-examples: The running example in Fig. 1 (2-CCWL on a triangle with ranks/colors) is helpful - please add one or two non-trivial pairs of ACCs separated by 2-CCWL but not by 1-CCWL, together with the corresponding $TC\_{3}$ formulas and two-pebble strategies spelled out.
>
> We thank the reviewer for this suggestion. As suggested, in the revised version, we have added Example 5.5, which presents a pair of ACCs that are indistinguishable by 1-WL, 1-CCWL, and Simplicial WL but can be separated by 2-CCWL. These examples rely on the fact that 1-WL and 1-CCWL have identical expressive power on 1-dimensional ACCs (Theorem D.2). We demonstrate separability in two ways: i) by providing a formula $\phi \in TC_4$ that distinguishes the pair; and ii) by showing a winning strategy for Player I in the topological 4-pebble game corresponding to $\phi$.
>
>
> References:
>
> [Bodnar et al., 2021(a)] Weisfeiler and Lehman Go Cellular: CW Networks. NeurIPS, 2021.
>
> [Bodnar et al., 2021(b)] Weisfeiler and Lehman Go Topological, ICML, 2021
>
> [Huang et al., 2021] UniGNN: a Unified Framework for Graph and Hypergraph Neural Networks. International Joint Conference on Artificial Intelligence, 2021
>
> [Papillon et al., 2024] Architectures of Topological Deep Learning: A Survey on Topological Neural Networks. ArXiv, 2024
>
> —-------------------------------------
>
> Thank you once more for the careful review and helpful suggestions. We believe the revisions directly address the raised concerns, and we would sincerely appreciate your strengthened support. We are, of course, happy to clarify any remaining points.

---

> > ### Comment · Reviewer_6rEA · 2025-11-24
> > **Author Response Acknowledgement**
> >
> > Thank you to the authors for the detailed and thoughtful rebuttal, as well as the substantial additions made to the revised manuscript.
> >
> > I have a few follow-up questions that would further strengthen the final version:
> >
> > 1. **Separation Tightness Beyond 1-Dimensional Lifts:** The new Theorem D.3 establishes the necessity of the $+2$ variable gap via 1-dimensional ACCs derived from graphs. Do you believe there exist intrinsically higher-dimensional ACC pairs that also witness this gap?
> >
> > 2. **Complexity in Practice:** The worst-case per-round cost is now explicit, but do you have empirical measurements of refinement depth (number of iterations until stabilization) on real ACCs from TNN benchmarks? Even a qualitative statement (e.g., typically <10 iterations) would help contextualize the theoretical bounds.
> >
> > 3. **Coverage of Emerging TNN Variants:** You note that 1-CCWL upper-bounds the expressiveness of injective-aggregator models like CIN. For newer architectures involving sheaf Laplacians, attention-based updates, or persistence-guided lifting, how far does your simulation argument extend?

---

> ### Author Response · Authors · 2025-12-01
> **Responses to Follow-up Questions: Part 1/2**
>
> Thank you for acknowledging our rebuttal. We are glad you found our response insightful and that the additional content strengthened our contribution. Below, we fully address your follow-up questions.
>
> > #### Question 1: **Separation Tightness Beyond 1-Dimensional Lifts:** The new Theorem D.3 establishes the necessity of the $+2$ variable gap via $1$-dimensional ACCs derived from graphs. Do you believe there exist intrinsically higher-dimensional ACC pairs that also witness this gap?
>
> Thanks for bringing up this interesting point. The current Theorem D.3 shows that the $(k{+}2)$-variable gap already appears on $1$-dimensional ACCs created from graphs, but this property is not only for dimension $1$.
>
> In fact, our separation construction can be “thickened’’ to higher-dimensional ACCs. Concretely, we can start from the $1$-dimensional witness pair $C_6$​ vs. $C_3\dot\cup C_3$​ (which separates $1$-CCWL from $2$-CCWL), replace each vertex in both graphs by a $C_6$​, and then add a $2$-cell that fills each of these cycles. The result is a pair of $2$-dimensional ACCs whose only 2-cells are these newly attached disks. Because we attach exactly the same gadget to every vertex in both complexes, $1$-CCWL sees only a “decoration’’ of the original structures, and therefore still cannot distinguish the thickened pair. On the other hand, $2$-CCWL continues to detect the difference in how these gadgets are interconnected, so the separation persists.
>
> More generally, any $1$-dimensional witness pair $(A,B)$ for the $k$-vs.-$(k{-}1)$ CCWL gap can be lifted to higher dimensions by gluing a fixed higher-dimensional ACC gadget to each $1$-cell of $A$ and $B$ in a uniform way. This preserves indistinguishability at level $(k{-}1)$ while keeping the level-$k$ separation.
>
>
> A full characterization of minimal higher-dimensional witnesses is an interesting open problem, and we can now mention this as a direction for future work.
>
>
> > #### Question 2: **Complexity in Practice:** The worst-case per-round cost is now explicit, but do you have empirical measurements of refinement depth (number of iterations until stabilization) on real ACCs from TNN benchmarks? Even a qualitative statement (e.g., typically <10 iterations) would help contextualize the theoretical bounds.
>
>  We first emphasize that when CCWL is used as an expressivity proxy for $L$-layer TNNs (as in our work), it is not necessary to run the refinement until full stabilization: one can stop after $L$ rounds or as soon as no further color splits occur. To provide some empirical evidence on the refinement depth, we implemented 1-CCWL on ACCs obtained from the Regular and Basic subsets of BREC dataset. In all of  that $1$-CCWL-distinguishable pairs, the refinement distinguishes a pair in fewer than 5 iterations.

---

> ### Author Response · Authors · 2025-12-01
> **Responses to Follow-up Questions: Part 2/2**
>
> > #### Question 3: **Coverage of Emerging TNN Variants:** You note that 1-CCWL upper-bounds the expressiveness of injective-aggregator models like CIN. For newer architectures involving sheaf Laplacians, attention-based updates, or persistence-guided lifting, how far does your simulation argument extend?
>
> Thank you for asking us to clarify this point. Our current simulation argument is phrased for message-passing TNNs with local, isomorphism-invariant update rules, but it is possible to extend to several of the newer architectures the reviewer mentions.
>
>
> - **Sheaf Laplacians / learnt sheaves.**
> In sheaf neural networks on graphs, one works with a cellular sheaf $(\mathcal{F}, \rho)$ over a graph $G = (V,E)$, where each vertex $v \in V$ carries a vector space $\mathcal{F}(v) \cong \mathbb{R}^{d_v}$, each edge $e \in E$ carries $\mathcal{F}(e) \cong \mathbb{R}^{d_e}$, and every incidence $v \in e$ has a linear restriction map
> $$
> \rho_{e \to v} : \mathcal{F}(e) \to \mathcal{F}(v).
> $$
> According to [Bodnar et al., 2023], these restriction maps are learned using locally available information. Thus, they each $\rho_{e \to v}(x_v, x_u)$ is a learned via a parametric matrix-valued function $\Phi$, such that:
> $$
> \rho_{e \to v}(x_v, x_u) = \Phi\bigl(x_v, x_u \bigr),
> $$
> for an edge $e = (u,v)$. Then
> The sheaf Laplacian is built from the incidence structure and the learned restriction maps, but each layer still behaves like standard message passing: a node updates its state from its own features and transformed features of its neighbors. We can extend this from graphs to ACCs, and assign each cell a vector space and for every adjacency, we use a learned restriction map. As long as the parameterization of the $\rho$'s is shared across isomorphic incidences (so the model is ACC-isomorphism invariant), these updates remain local and are based only on the ACC neighborhood structure.
>
> - **Persistence-guided lifting.** Persistence-guided liftings start from a graph, compute persistent homology (or related invariants), and then “lift” the graph to a higher-dimensional complex whose cells encode topological information, before running a higher-order message-passing network. From our viewpoint, this just changes the underlying ACC and its initial attributes: once the lifted ACC is fixed, any subsequent local, isomorphism-invariant message passing on it is upper-bounded by 1-CCWL (or $k$-CCWL, for higher-order variants). In particular, our results apply directly to persistence-guided lifting pipelines where persistence is computed once on the input and then used to define the ACC on which the TNN operates.
>
> - **Attention-based updates.** Attention layers that compute weights from pairs of neighboring features and then aggregate via permutation-invariant operations are still multiset functions over local neighborhoods. From the expressiveness point of view, this does not go beyond message passing in the WL sense: on graphs, such models are known to be at most as powerful as $1$-WL, and the same reasoning carries over to ACCs. Thus, architectures like topological transformers that use local attention along the incidence structure of a complex are captured by $1$-CCWL (again, or $k$-CCWL if they are defined on $k$-tuples). Our framework abstracts _what_ structural patterns can be distinguished, independently of whether the local update is implemented via simple sums, gated updates, or attention.
>
> -------
>
> References
>
> [Bodnar et al., 2023] Neural Sheaf Diffusion: A Topological Perspective on Heterophily and Oversmoothing in GNNs, NeurIPS, 2022
>
> ------
> #### Many thanks for your constructive feedback and follow-up questions that have helped us reinforce the many strengths of this work. We are particularly excited by the avenues this work opens up for design of novel TNN architectures, including, based on the ideas that we included above in our response.

---

### Official Review · Reviewer_bQiY · 2025-10-29

**Soundness:** 3
**Presentation:** 2
**Contribution:** 3
**Rating:** 6
**Confidence:** 3

**Summary:**

The paper presents new expressiveness characterisations in the realm of topological deep learning, a recent field dealing with the design of  neural network architectures for higher-order relational data. The main result is an equivalence triad which mirrors results in the expressive power of machine learning on graphs. This results connects isomorphism testing, a fragment of FOL, and pebble games.

In particular, the authors introduce a novel: (i) variant of the k-WL test adapted to combinatorial complexes (CCs); (ii) "topological" fragment of FOL with k-variables and counting quantifiers; (iii) "topological pebble game".

(i) The authors introduce the k-CCWL test hierarchy to distinguish non-isomorphic combinatorial complexes. The main novelty wrt to the standard k-WL test, consists in simultaneously tracking two cell substitution per tuple position. The authors formalise this aspect trying to link it to the main feature of typical upper- and lower- neighbourhoods in topological neural networks, which introduce the feature of the boundary or co-boundary cells shared between two adjacent ones.

(ii) The authors introduce a "topological counting logic" which resembles the one studied in graphs, but with the main difference being that counting quantifiers are pairwise. The authors show an equivalence between the k-CCWL  and this fragment with k+2 variables.

(iii) The authors introduce a variant of the pebble game over combinatorial complexes, where the main difference w.r.t. the "standard" pebble games is that players select and mark couples of cells (instead of nodes). The authors show the equivalence between the k-variable topological counting logic and the topological k-pebble game in terms of the duplicator winning strategies.

Finally the authors show the equivalence between the k-CCWL test and the (k+2)-pebble game.

**Strengths:**

[S1] The work is relevant and timely. It contributes a set of insights and tools which, going forward, will support better and more informed quantifications of expressive power for topological neural network architectures.

[S2] The paper is presented in a precise way, the authors give sufficient background to grasp the relevance of the contribution and the existing, base results in the realm of graphs.

[S3] The way the fundamental expressiveness characterisations are extended is intriguing and interesting, that is, the "shift" to the pairwise paradigm in counting quantifiers, pebbles, tuple variable substitutions.

**Weaknesses:**

[W1] The main results are relevant, but remain rather abstract.
- [W1.1] The authors does not provide evidence of how they can, for example, help characterise the expressiveness of existing architectures. Do these tools already support some, even preliminary, architectural stratification?
- [W1.2] Does the new counting logic allow to grasp some intuition on what kind of topological structural properties methods can or cannot capture?
- [W1.3] Other than completing the "triad", what kind of interesting intuition can we draw from the newly introduced pebble game?

[W2] Related to W1.1 – The paper does not connect to the learning aspect at all. Generally speaking this is not an easy endeavour, and even in the graph literature, this connection is rather unexplored. A precise characterisation is clearly out of scope, but considering the current venue scope ... _experimentally_, are expressiveness insights minimally reflected in practice, even in some synthetic benchmark?

[W3] A recent paper explores expressiveness limits of topological neural networks: Eitan et al, 2025 (https://arxiv.org/pdf/2408.05486). It would be extremely interesting, other than expected, to discuss at least some links between the two works. The authors does not seem to do that.

[W4] Generally speaking, the paper could appear not to be accessible to all readers, being rather theory heavy. I believe something that could significantly improve the quality of the paper is to provide more intuitions and illustrations.
- [W4.1] This "shift" to the pairwise paradigm that marks a difference w.r.t. standard expressiveness characterisation is only intuitively justified in lines 63 through 67, but I believe this is one of the most important aspects behind the contribution. The authors should give it more emphasis and illustrate it better to ground the following contributions.
- [W4.2] Lines 251 -- 257 – This paragraph is rather unclear; what is the intuition behind the broadcast anchors?

[W5] Minor – some references are not up to date and still report preprints instead of publication venues.

Overall, with the caveat that I have not checked the math in the proofs reported in appendix, I lean towards recommending acceptance, but addressing the above weaknesses would positively strengthen my evaluation.

**Questions:**

[Q1] Lines 251 -- 257: Can the authors please explain the idea of the broadcast anchor? this paragraph appears rather detached from the rest of the paper and, to the best of my knowledge, this is also something different w.r.t. the standard k-WL test.

[Q2] Can the authors explain why the atps are injected in multisets in every refinement step, even in standard k-WL? This is not something standard, to the best of my knowledge (see, e.g., Eq. 4).

Also see other weaknesses

---

> ### Author Response · Authors · 2025-11-23
> **Part 1/4**
>
> Thank you very much for your helpful feedback. Below, we provide point-by-point responses to your comments.
>
> > Regarding Weakness 1: The main results are relevant, but remain rather abstract.
>
> > [W1.1] The authors does not provide evidence of how they can, for example, help characterise the expressiveness of existing architectures. Do these tools already support some, even preliminary, architectural stratification?
>
> _Our results give a hierarchy of ACC-based architectures:_ Intuitively, any model with invariant update functions under $k$-CCWL cannot distinguish ACCs that $k$-CCWL (or equivalently $\mathrm{TC}_{k+2}$) cannot separate. In case $k = 1$, we use the standard message-passing architectures on ACCs by updating over boundary, coboundary, lower, and upper adjacencies, same as $1$-CCWL. In $k>1$, the architecture works on $k$-tuples and updates the labels based on the double-shift operation, same as $k$-CCWL. Therefore, the practical impact can be a provable improvement in expressivity power by increasing $k$, and using higher-order topological architectures.
>
> > [W1.2] Does the new counting logic allow to grasp some intuition on what kind of topological structural properties methods can or cannot capture?
>
> _What structures can be captured:_ The new counting quantifier lets us have formulas such as $\exists^{N}(x,y) E^{\mathcal{N}\_{B}}(x,y)$, with quantifier depth $1$ and two variables $x$ and $y$. This formula expresses that there are at least $N$  pairs $(x,y)$ in an ACC such that $x$ is in the boundary of $y$. Considering formulas in $\mathrm{C}_{2}$ (counting logic with simple quantifier) with quantifier depth $1$, it is impossible to have a formula that counts pairs of cells satisfying a specific property, such as this boundary relation. Therefore, with the same number of variables and even smaller quantifier depth (which correspond, respectively, to the index $k$ and the number of iterations of the coloring algorithm), $k$-CCWL can distinguish earlier than $k$-WL. At the same time, $\mathrm{TC}\_{k+2}$ still has clear limitations. For fixed $k$, it cannot express majority-style properties (e.g., “more than half of the cells have property $P$”), nor properties that depend on connectivity at unbounded distance (e.g., whether two far-away components are connected beyond any fixed radius). In the revised version, we added a short “Limitations” paragraph in the main text to make these blind spots explicit.
>
> > [W1.3] Other than completing the "triad", what kind of interesting intuition can we draw from the newly introduced pebble game?
>
> _Interesting intuition of new pebble game:_ Although we use the topological pebble game to complete the triad, we can mention the following useful insights for the game:
>
> - The game provides _an interactive view of distinguishability on ACCs_, while Player I tries to expose the differences and Player II tries to hide them.
> -  The way we proved the triad, in fact, we give a way to translate a winning strategy for Player I to a TC formula that separates the ACCs. In other words, it provides us _a constructive method for finding the separating properties_.
> - The number of pebbles that are needed shows the query complexity of detecting a property. In particular, if a property requires $k$ pebbles, and we have a pair of ACCs that can only be separated using this property, then neural architectures of dimensionality $(k - 2)$ are insufficient to distinguish these two ACCs. Therefore, it can be used as a method to find a _lower bound on architectural complexity_.
>
> > Weakness 3. A recent paper explores expressiveness limits of topological neural networks: Eitan et al, 2025 (https://arxiv.org/pdf/2408.05486). It would be extremely interesting, other than expected, to discuss at least some links between the two works. The authors does not seem to do that.
>
> We thank the reviewer for pointing us to this very relevant work. In the revised version, we now explicitly discuss Eitan et al. (2025) in the main text.
>
> In fact, Eitan et al. study the expressiveness of higher-order message passing (HOMP) architectures on CCs using a topological notion of indistinguishability based on coverings. They show that if two complexes share a common cover, then HOMP cannot tell them apart, which leads to concrete "blinspots" for these models, e.g., they cannot distinguish complexes that differ in diameter, orientability, planarity or homology. Our work is complementary: we develop a logical / WL-style hierarchy on ACCs (CCWL $\leftrightarrow$ counting logic $\leftrightarrow$ pebble games) that is architecture-agnostic, and it gives a general upper bounds for any ACC architecture whose update rules are invariant under $k$-CCWL (including many HOMP-style, message-passing, and incidence-based models). From our point of view, many of the “blindspots” found by Eitan et al. correspond exactly to properties that need unbounded quantification and therefore fall outside $TC\_{k+2}$ for any fixed $k$.

---

> > ### Comment · Reviewer_bQiY · 2025-11-26
> >
> > > What structures can be captured [...]
> >
> > Can the authors help me understand the following:
> > - Why comparing to $C_2$ in the formula example? Apologies if I missed something, but: how is the standard k-WL defined when operating on ACCs? It would better help for example to report an example of a concrete structural property captured by (k+1)-CCWL but not k-CCWL.
> > - What is the intuitive reasons why majority-style properties or properties that depend on connectivity at unbounded distance are not within reach for $TC_{k+2}$ and any fixed $k$? Does this mean that these cannot be captured for any $k$? I guess I am not well understanding the role of $k$ in these.
> >
> > In relation to Eitan et al., can the authors better explain why:
> > >  many of the “blindspots” found by Eitan et al. correspond exactly to properties that need unbounded quantification and therefore fall outside of [...]
> >
> > ?

---

> ### Author Response · Authors · 2025-11-23
> **Part 2/4**
>
> >Weakness 2.  The paper does not connect to the learning aspect at all. [...] considering the current venue scope experimentally, are expressiveness insights minimally reflected in practice, even in some synthetic benchmark? [Related to Weakness 1.1]
>
> Thanks for your question. We believe that expressiveness plays an important role in practice. It is well known that the excess risk of a model class decomposes into approximation, optimization, and statistical errors, and expressivity directly governs the approximation term. A model that fails to represent the relevant functions on the domain will necessarily incur a non-negligible approximation error, regardless of training procedure or data availability.
>
> In Graph ML, expressiveness has long become a central theme. Many influential architectures were explicitly developed to overcome the well-documented limitations of vanilla message-passing GNNs. Examples include positional encodings, feature injection schemes, subgraph-based architectures, and topological or structural descriptor extraction. Reflecting this importance, the community has even introduced dedicated benchmarks for measuring expressivity on graphs, such as the BREC benchmark [Wang \& Zhang, 2024].
>
> To illustrate this point, we ran experiments on two BREC datasets: Basic (60 graph pairs) and Regular (50 graph pairs). While 1-WL—and therefore any standard message-passing GNN—fails to distinguish any of these pairs, we find that 1-CCWL on rank-2 complexes and 1-CCWL on rank-3 complexes distinguish 91.7\% / 98.3\% of the Basic pairs, and 86\% / 96\% of the Regular pairs, respectively.
>
> In these experiments, we construct the underlying combinatorial complexes using clique lifting combined with coface-to-cell lifting. We will add these results to the revised manuscript.
>
> Overall, we believe that uncovering the expressive power of ML models is essential for understanding their strengths and limitations, and for guiding the design of more principled, nuanced architectures that are both theoretically grounded and practically effective.
>
>
> > Weakness 4. Generally speaking, the paper could appear not to be accessible to all readers, being rather theory heavy. I believe something that could significantly improve the quality of the paper is to provide more intuitions and illustrations. [W4.1] This "shift" to the pairwise paradigm that marks a difference w.r.t. standard expressiveness characterisation is only intuitively justified in lines 63 through 67, but I believe this is one of the most important aspects behind the contribution. The authors should give it more emphasis and illustrate it better to ground the following contributions.
>
>
>
> We thank the reviewer for this valuable comment, and we agree that the shift to the pairwise viewpoint is one of the central ideas of the paper and should be explained more clearly.
>
> In the revised version, we added more Figures 2, 3, 4, and 5 to improve the quality. We add a short informal paragraph in the introduction (tentatively titled ``Why pairwise counting on ACCs?'') that explains this shift with concrete intuition. In particular, we emphasize:
>
> - _Topological relations are mediated by third cells:_ In graphs, relationships are directly between two vertices: either there is an edge $(u,v)$ or not. In ACCs, many relations between cells are _mediated_ by other cells. For example, two edges $e_1,e_2$ are lower-adjacent if they share a vertex $v$ (i.e. $e_1,v,e_2$), and upper-adjacent if they lie in a common higher cell $t$ (i.e. $e_1,t,e_2$).
>
> - _Boundary–coboundary interactions are inherently pairwise:_ Each cell has a boundary (lower-dimensional faces) and a coboundary (higher-dimensional cofaces). Many interesting topological patterns depend on how boundary and coboundary neighborhoods interact: e.g., “which faces of $x$ are also shared with cells that co-occur with $x$ in some higher cell?”. Capturing such interactions needs tracking pairs $(\alpha,\beta)$ of boundary and coboundary cells.
>
> - _Adjacency via shared cells and incidence matrices:_ Lower and upper adjacencies can be defined via shared boundary or coboundary cells, and algebraically they appear as products of incidence matrices (e.g. $B_k^\top B_k$ and $B_{k+1} B_{k+1}^\top$). These products sum over indices that mediate connections between two cells, again reflecting an underlying pairwise
>
> > Weakness5. Minor – some references are not up to date and still report preprints instead of publication venues.
>
> We thank the reviewer for pointing this out. In the revised version, we have updated all references that now have publication venues and replaced preprint citations with the final published versions.

---

> > ### Author Response · Authors · 2025-11-23
> > **Part 3/4**
> >
> > > Question 1. Lines 251 -- 257: Can the authors please explain the idea of the broadcast anchor? this paragraph appears rather detached from the rest of the paper and, to the best of my knowledge, this is also something different w.r.t. the standard k-WL test. [Related to W4.2]
> >
> > _Does the isomorphism test work without it?_ Yes! We note that broadcast-anchors allows us to define fresh colors to 0 and 1-rank cells so they can be distinguished from the original ones in two input CCs A and B. To define a CC-isomorphism test between A* and B* (i.e., A and B with anchors), we must map the anchor in A* to the anchor in B*. Therefore, there exists a CC-isomorphism between A* and B* iff there is a CC-isomorphism between A and B. Thus, **adding broadcast-anchor cells does not affect whether A and B are CC-isomorphic**.
> >
> > _Why do we need a broadcast-anchor?_  The intuition is that the broadcast-anchor allows for capturing color differences between CC elements and propagating them to other cells. The broadcast-anchor is introduced to prove the _identical-vs-disjoint_ property for uniform ACCs (Proposition 3.1). This property is important for two reasons:
> >
> > - It is important for the proof of the equivalence between $k$-CCWL and TCLogic{k+2} on uniform ACCs (Theorem 5.4). In particular, it lets us connect equality of global stable colors of two ACCs with agreement vs. disagreement on all sentences of TCLogic{k+2} (i.e., formulas without free variables). Intuitively, this gives a clean structure-level perspective on the expressive power of $k$-CCWL and TCLogic{k+2} by eliminating the need to reason about valuations.
> >
> > - It also leads to an efficiency improvement for the $k$-CCWL isomorphism test. With the anchor and the identical-vs-disjoint property in place, we can run $k$-CCWL on each ACC once and then, after stabilization, inspect the color of a single global tuple in one ACC and only check whether this color occurs in the other ACC: if it does, then the two sets of global stable colors coincide, hence $k$-CCWL cannot distinguish the complexes; if it does not, then the sets are disjoint and $k$-CCWL does distinguish them. Without the anchor, even if one color appears in both complexes, we still have to search for all other colors (and their multiplicities). To see this, consider two graphs $A=K_2 \cup K_{1,3}$ and $B=K_2 \cup P_4$  --- here, $K_{n,m}$ is a bipartite graph with partitions of size $n$ and $m$ and $P_4$ is a path graph. Here, we have that their multisets of stable colors overlap but are different.
> >
> > _Did we need the broadcast-anchor in the standard $k$-WL test?_ As it is discussed in Chapter 3 of Kiefer (2020), for the similar reason, the broadcast-anchor is also introduced in the standard $k$-WL test on graphs, to make sure the identical-vs-disjoint property for graphs, and as a result of that, they prove the equivalence between $k$-WL and $\mathrm{C}_{k+1}$ counting logic on graphs as a structure-level isomorphism test.
> >
> > >Question 2.  Can the authors explain why the atps are injected in multisets in every refinement step, even in standard k-WL? This is not something standard, to the best of my knowledge (see, e.g., Eq. 4).
> >
> >
> > In standard $k$-WL on graphs, the initial coloring of $k$-tuples is defined by the isomorphism type, which is another interpretation of atomic type. Therefore, in [Cai et al. (1992); Grohe(2021); Huang & Villar (2021); Morris et al. (2019b)], the atomic type is used as the initial coloring of tuples, and then carried along in the later iterations as part of the previous color.
> >
> > In our setting, as well as in [Geerts & Reutter (2022); Kiefer (2020)], the atomic type is not only used as the initial coloring, but also is injected in every iteration. Furthermore, for $k$-CCWL, the atomic type $\operatorname{atp}_{k{+}2}(\mathbf{x}\alpha \beta)$, encodes all the local information: the initial colors, and ranks of $\alpha$, and $\beta$, their adjacency, and equality relations to the enteries of $\mathbf{x}$. Therefore, we can directly use the atomic type, rather than referring to the information that is carried by the colors of the previous iteration.
> >
> > From a complexity point of view, as it is noted, the atomic type of a tuple can be computed in $O(1)$ time per $k$-tuple, in each round, so this formulation does not change the asymptotic running time. From a theoretical point of view, injecting atomic type in every iteration does not increase the expressive power compared to the standard definition of $k$-WL, and it simplifies several of our arguments.

---

> > ### Comment · Reviewer_bQiY · 2025-11-26
> >
> > > we find that 1-CCWL on rank-2 complexes and 1-CCWL on rank-3 complexes distinguish 91.7% / 98.3% of the Basic pairs, and 86% / 96% of the Regular pairs, respectively.
> >
> > In regards to the experiment on BREC: how is this supporting claims in the paper? I was expecting variations in $k$ for example, not in the max rank of the lifting?

---

> ### Author Response · Authors · 2025-11-23
> **Part 4/4**
>
> References:
>
> [Huang & Villar, 2021] A short tutorial on the Weisfeiler-Lehman test and its variants. International Conference on Acoustics, Speech and Signal Processing (ICASSP), 2021
>
> [Morris et al., (2019b)] Weisfeiler and Leman Go Neural: Higher-order Graph Neural Networks. AAAI Conference on Artificial Intelligence, 2019
>
> [Grohe, 2021] The logic of graph neural networks, 36th Annual ACM/IEEE Symposium on Logic in Computer Science. 2021
>
> [Cai et al., 1992] An Optimal Lower Bound on the Number of Variables for Graph Identification. Combinatorica, 1992.
>
> [Geerts & Reutter, 2022] Expressiveness and approximation properties of graph neural networks, ICLR, 2022
>
> [Kiefer, 2020] Power and limits of the Weisfeiler-Leman algorithm, Dissertation, RWTH Aachen University, 2020
>
> —-------------------------------------
>
> Thank you once more for the careful review and helpful suggestions. We believe the revisions directly address the raised concerns, and we would sincerely appreciate your strengthened support. We are, of course, happy to clarify any remaining points.

---

> ### Comment · Reviewer_bQiY · 2025-11-26
> **Response to rebuttal**
>
> I really appreciate the insightful responses to my review, which I believe, integrated in the manuscript, make the contribution stronger.
>
> I have some outstanding questions related to these answers which I report below as comments to specific rebuttal sections.

---

> ### Author Response · Authors · 2025-12-03
> **Part 1/2**
>
> Thank you again for your thoughtful follow-up questions. We are happy that our earlier rebuttal addressed your main concerns and that the additional material helped clarify and strengthen our contribution. Below, we provide responses to your remaining questions.
>
> > #### Question: Why comparing to in the formula example? [...]
>
> _Why we compare to $C_2$ in the example:_
>
> The purpose of that answer is not to compare with the standard $k$-WL on graphs. It is to show the additional expressive power provided by the pairwise counting quantifier in $\mathrm{TC}_{k}$, in comparison to standard unary counting of $\mathrm{C}_k$.
>
> As it is stated in [Grohe 2021, Remark III.1], counting $\mathrm{C}_k$ can be extended from graph to arbitrary finite relational structures in a routine way: one simply adds atomic formulas for each binary relation symbol (boundary, coboundary, lower and upper), and unary relation symbols (rank). In our setting, an ACC is viewed as such a relational structure, with:
>
> Binary relations such as $E^\mathcal{N}_B(x,y)$ for boundary relation, and
> Unary predicates for ranks, colors, etc
>
> Thus, even for ACCs we may consider a “$\mathrm{C}_k$-like” logic which includes the ACC’s adjacency predicates and rank predicates, but only standard (unary) counting quantifiers. The example in the answer
>
> $ \exists^{\ge N}(x,y)\; E^{\mathcal{N}_B}(x,y)$
>
> is intended precisely to highlight what the new **pairwise** counting quantifier gives us. As a simple example, it speaks about the _number of pairs_ $(x,y)$ rather than just the number of individual cells, and thus, e.g., can distinguish ACCs that have the same number of cells of each rank but differ in how these cells are attached together along their boundaries.
>
> _Concrete structural properties captured by $(k+1)$-CCWL but not $k$-CCWL_
>
> Beyond this illustrative toy example, our manuscript now contains more structured separations:
>
> - **Example 5.5** gives a concrete pair of ACCs that $1$-CCWL cannot distinguish but $2$-CCWL can (via both logic and game arguments).
>
>
> - **Lemma D.1** provides another $1$-dimensional ACC pair with the same separation: indistinguishable by $1$-CCWL, distinguishable by $2$-CCWL.
>
>
> - **Theorem 3.2** generalizes these examples: for every $k$, there exists a pair of ACCs distinguished by $k$-CCWL but not by $(k{-}1)$-CCWL, so the hierarchy is strictly increasing in $k$.
>
>
> > #### Question: What is the intuitive reasons [...]
>
> _What “for fixed $k$” means._
>
> When we say that a property (e.g., unbounded-distance connectivity) is not expressible “for fixed $k$”, we mean the following: for every $k \in \mathbb{N}$, there is _no_ formula $\phi \in \mathrm{TC}\_{k}$ such that, for **every** structure $A$,
> $$A \models \phi \quad \text{iff} \quad A \text{ has property } P.$$ Equivalently, there is no single finite $k$ for which $P$ is definable in $\mathrm{TC}\_{k}$ on **all ACCs**. We do _not_ mean that, for a given pair of instances, there is no $k$ and no $\mathrm{TC}\_{k}$-formula that can distinguish them; the point is that no single $k$ works _uniformly_ for all instances.
>
> _Give more intuitive reasons_
>
> Intuitively, it comes from the _locality characteristic_ in finite-variable first-order logic with counting, to which our $TC_{k}$ belongs [Libkin, 2004; Gaifman & Vardi, 1985]. For any fixed $k$ and any $TC_{k}$-sentence $\varphi$ of quantifier depth $q$ (which, in our setting, corresponds to a bounded number of refinement iterations / layers), there is a finite radius $r = r(k,q)$ such that the truth of $\varphi$ on a complex depends only on the isomorphism types and counts of radius-$r$ neighborhoods. This immediately shows properties like connectivity: for example, take a large cycle $C_{2N}$ and the disjoint union $C_N \uplus C_N$. For $N \gg r(k,q)$, all radius-$r$ neighborhoods in these two graphs are just paths, so every $TC_{k}$-sentence of depth $q$ has the same truth value on both, even though one graph is connected and the other is not. Thus, when we say that such properties “fall outside $TC_{k}$ for any fixed $k$”, we mean precisely that there is no single choice of $k$ and no single $TC_{k}$-sentence that defines, e.g., connectivity on all finite ACCs; any successful definition would require formulas whose effective radius (through quantifier depth and/or number of variables) grows unboundedly with the size of the complex. Increasing $k$ enlarges the space of local patterns that can be seen, but as long as $k$ is fixed, these global, unbounded-distance properties remain out of reach.

---

> ### Author Response · Authors · 2025-12-03
> **Responses to Follow-up Questions: Part 2/2**
>
> >#### Question: In relation to Eitan et al., can the authors better explain why: many of the “blindspots” found by Eitan et al. correspond exactly to properties that need unbounded quantification and therefore fall outside of [...]
>
> In logical language, this locality can be phrased as follows.
>
> Let $\rho : \widetilde{X} \to X$ be a covering map in the sense of [Eitan et al, 2024], and let
>
> $$\widetilde{u} = (\tilde u_1,\dots,\tilde u_k) \in \widetilde{X}^k, \qquad
> u = \rho(\widetilde{u}) = \big(\rho(\tilde u_1),\dots,\rho(\tilde u_k)\big) \in X^k.$$
>
> Because $\rho$ is a local isomorphism, one can show by a straightforward induction on the structure of a $\mathrm{TC}_{k+2}$-formula $\varphi$ of quantifier rank at most $q$ that, as long as its locality radius $r(k,q)$ is below the covering radius,
>
> $$ X,u \models \varphi \quad\Longleftrightarrow\quad \widetilde{X},\widetilde{u} \models \varphi.$$
>
> Now suppose $X$ and $X'$ share a common cover $\widetilde{X}$ with covering maps
>
> $\rho : \widetilde{X} \to X$ and $\rho' : \widetilde{X} \to X'$. Applying the same argument to $\rho'$ yields
>
> $$X,u \models \varphi
> \quad\Longleftrightarrow\quad
> \widetilde{X},\widetilde{u} \models \varphi
> \quad\Longleftrightarrow\quad
> X',\rho'(\widetilde{u}) \models \varphi,$$
>
> so every $\mathrm{TC}_{k+2}$-sentence of rank $\le q$ has the same truth value on corresponding tuples in $X$ and $X'$. In Eitan et al.'s cylinder--Möbius and related examples, the common cover can be chosen with arbitrarily large girth, so for any fixed $k$ and $q$ we can make the covering radius exceed $r(k,q)$. Hence no single $\mathrm{TC}\_{k+2}$-sentence with fixed $k$ and bounded quantifier rank can distinguish these pairs; any formula that does so must have unbounded quantifier depth. This is exactly what we mean when we say that many of their ``blindspot'' properties fall outside $\mathrm{TC}\_{k+2}$ for every fixed $k$.
>
>
> > #### Question: In regards to the experiment on BREC: how is this supporting claims in the paper? I was expecting variations in for example, not in the max rank of the lifting?
>
>
> ُThanks for raising this follow-up question. We will clarify the purpose of the experiment in the paper. Our intention here is to _ground_ our framework on a well–known expressivity benchmark and sanity–check the theory in a concrete setting.
>
> Concretely, each BREC graph is lifted to an ACC and we run $1$-CCWL on this lift. The experiment supports two claims from the paper:
> (i) on $1$-dimensional data, $1$-CCWL behaves in line with the classical WL hierarchy (it does not artificially “beat” the known $1$-WL limits on BREC), and
> (ii) when we vary the lifting scheme (e.g., lifting up to rank--1 vs.\ rank--2 cells on the same input), the performance improves in line with the theory: adding higher--rank cells enriches the ACC structure and lets $1$-CCWL distinguish more pairs, without contradicting the known lower bounds of the WL hierarchy.
>
> References
>
> [Grohe, 2021] The Logic of Graph Neural Networks, LICS, 2021
>
> [Eitan et al., 2024] Topological Blindspots: Understanding and Extending Topological Deep Learning Through the Lens of Expressivity, ICLR, 2025
>
> [Libkin, 2004] Elements of Finite Model Theory., Springer Texts in Theoretical Computer Science, 2004.
>
> [Gaifman & Vardi, 1985] A Simple Proof that Connectivity of Finite Graphs is Not First-Order Definable. Bulletin of the European Association for Theoretical Computer Science (Bull. EATCS), 1985.
>
> ------
>
> ### We are very grateful for your constructive feedback throughout the review process, which has helped us sharpen both the technical contributions and their conceptual positioning.

---

### Official Review · Reviewer_qRd9 · 2025-10-31

**Soundness:** 3
**Presentation:** 2
**Contribution:** 3
**Rating:** 2
**Confidence:** 5

**Summary:**

In this paper, the authors study the so-called logical expressiveness of topological neural networks. They first define the k-combinatorial complex WL test (k-CCWL) on Page 5, via adapting the k-WL test for graphs to topological networks. Next, they show that for uniform attributed combinatorial complexes (ACCs), the output representations of two objects via an instance k-CCWL are either completely disjoint sets or completely equal sets.
Here, 'uniformity' allows us to define some notion similar to neighborhoods in graphs, and thus one can then extend message passing to such neural networks.


Next, they define topological counting logic TC-k, as well as topological k-pebble games, where the latter is a game defined to mimic the definition of topological logic.

Here are the main results: In Theorems 4.1, they prove that if for every formula in topological logic TC-(k+2), two given ACCs are consistent, then the k-CCWL coloring of them will also be the same. Next, together in Theorem 5.1 and Theorem 5.2, they show that having a winning strategy for a particular player in (k+2)-pebble game is equivalent to the topological logic TC-(k+2) assumption, completing the equivalence of different notions of expressiveness. This is given in Corollary 5.3 and Theorem 5.4. See also Eq. 6 that summarizes the main contributions of the paper.

**Strengths:**

- contributions to the theory of topological neural networks, which is of potential interest to this part of the community

- Nice and clean theoretical results, solid paper at the intersection of math and AI

**Weaknesses:**

- This paper, while having great contributions, is less accessible to the community. It is hard to follow.

**Questions:**

I completely read the main body of this work and found that it is a great paper. It contains solid mathematical contributions characterizing the expressive power of topological neural networks via different approaches: (1) methods similar to k-WL for graphs, (2) the introduced notion of pebble games, (3) the definition of the topological logic classes of order k. This contributed to the theory side of geometry and topology in neural networks.



Unfortunately, this paper, while having great contributions, has a major problem. It is less accessible to the community. As ICLR and even the geometry community within AI have different backgrounds, it is necessary to make sure that a paper with that level of great math contributions is accessible. The author should make sure that a reader, even if they barely know geometry and topology, could understand the main contribution and the message. The first few pages of the paper have to deliver the message to people who do not have much background in theory, yet they want to know what the contributions of the paper are. For instance, for a practitioner, the paper is absolutely difficult to read and follow.


I suggest that the author extensively reconsider the first few pages of this paper and rewrite them so that it is more accessible. That way, the paper will get more audience and will have more impact, given its great contributions to the math of AI. As a result of this, for the current version of the paper, I recommend rejection.



- Typo, Line 91: There is nothing called 'Theorem 3.1' in the paper, which is cited multiple times in the paper. Probably the authors meant 'Proposition 3.1.' Please make sure to correct all such typos in the paper.

---

> ### Author Response · Authors · 2025-11-23
> **Part 1/2**
>
> Thank you for taking the time to read our paper and for acknowledging the strength of the mathematical contributions. Below, we offer our perspective on the accessibility concerns you raised. Based on your feedback, we have improved the presentation of our results and expanded the “Contributions” paragraph in the introduction to provide a clearer, high-level overview of the key ideas.
>
> > Weakness: This paper, while having great contributions, is less accessible to the community. It is hard to follow.
> Unfortunately, this paper, while having great contributions, has a major problem. It is less accessible to the community. As ICLR and even the geometry community within AI have different backgrounds, it is necessary to make sure that a paper with that level of great math contributions is accessible [...]
>
> We appreciate your feedback and agree that clarity and accessibility are important. Below, we explain the steps taken to ensure readability and why we believe the paper is suitable for the ICLR audience.
>
> **Story is easy to follow**. The Introduction provides an accessible narrative. In particular, we begin with an overview of GNNs and their well-known connection with the 1-WL isomorphism test, highlighting works that use first-order logic to characterize the expressivity of GNNs. We then introduce TNNs (topological neural networks), the central object of study, as a natural response to the limited expressive power of vanilla GNNs. Also, we pose our research question: “What is the logical expressivity of TNNs?” Next, we introduce our contributions, namely: the higher-order isomorphism tests related to TNNs; the novel counting logic we introduce, and its associated game. Importantly, it is well-known in the field that ($k$-)WL test has the same expressivity power as $C\_{k+1}$ counting logic, and also can be characterized by the ($k$+1)-pebble game. Thus, these are not completely new notions for people interested in the Theory of GNNs.
>
> **Contributions are summarized in Introduction**. Importantly, we finish Section 1 (Introduction) by summarizing the key results of the paper to help readers understand the structure of our results. This provides a high-level idea of our theoretical results and where to find them in the paper.
>
> **Background introduces all required notions**. Our background covers all key notions and results. In particular, we introduce TNNs and combinatorial complexes — these models have become very popular in ML [e.g., 1,2,3, 4]. Then, we define the k-WL algorithm. Finally, we introduce counting logic, key results regarding the logical expressivity of k-WL by Cai et al. (1992), and the pebble game — a notion used to characterize counting logic.
>
> **The paper does not assume advanced topology or geometry**. We would like to emphasize that our mathematical constructions are intentionally kept lightweight: we only introduce basic combinatorial complexes (as finite sets of elements) and four adjacency notions all phrased in elementary discrete terms, before any logic or games appear. These definitions require no prior knowledge of topology or geometry; they mainly rely on standard linear algebra and discrete mathematics, and most of the proofs are induction arguments. Also, this is very much in line with the theoretical works on GNN logical expressivity previously published at ICLR.
>
> **ICLR is home to relevant related works**. We note that relevant works related to ours have been published at ICLR, namely:
>
> - Barceló et al. “The Logical Expressiveness of Graph Neural Networks.” International Conference on Learning Representations (ICLR), 2020.
>
> - Geerts and Reutter. “Expressiveness and Approximation Properties of Graph Neural Networks.” International Conference on Learning Representations (ICLR), 2022.
>
> - Xu et al. “How Powerful are Graph Neural Networks?” International Conference on Learning Representations (ICLR), 2019.
>
> The first two works leverage logic to understand and characterize the expressive power of GNNs, whose connection with 1-WL was established in the last work.
>
> > I suggest that the author extensively reconsider the first few pages of this paper and rewrite them so that it is more accessible.
>
> Thanks for your comment. **Based on your feedback, we have expanded the Contributions paragraph (Introduction) to include a high-level overview of the paper, aiming to make the key ideas and contributions broadly accessible.**
>
> Additionally, we would be grateful if the reviewer could provide more specific suggestions or examples of points that felt difficult to follow, so we can further improve the presentation.

---

> > ### Author Response · Authors · 2025-11-23
> > **Part 2/2**
> >
> > > Question. Typo, Line 91: There is nothing called 'Theorem 3.1' in the paper, which is cited multiple times in the paper. Probably the authors meant 'Proposition 3.1.' Please make sure to correct all such typos in the paper.
> >
> > Thank you for catching this. You are absolutely right that there is no “Theorem 3.1” in the paper and that those references were inconsistent with the actual statement, which is a proposition. In the revised version, we have fixed the cross-references so that all former mentions of “Theorem 3.1” now correctly refer to Proposition 3.1. We also systematically checked all theorem/lemma/proposition references in the manuscript to ensure that labels, numbering, and types are now fully consistent throughout.
> >
> > References:
> >
> > [1] Position: Topological Deep Learning is the New Frontier for Relational Learning, ICML, 2024
> >
> > [2] TopoTune : A Framework for Generalized Combinatorial Complex Neural Networks. ICML, 2025
> >
> > [3] Weisfeiler and Lehman Go Cellular: CW Networks. NeurIPS, 2021.
> >
> > [4] Architectures of Topological Deep Learning: A Survey on Topological Neural Networks. ArXiv e-prints, 2023
> >
> > ---
> > We hope this response alleviates your concerns. If our clarifications address your accessibility question, we would sincerely appreciate your reconsideration of the score. Otherwise, we remain happy to discuss further.

---

> > > ### Comment · Reviewer_qRd9 · 2025-11-23
> > >
> > > Thanks for your response. I will update you shortly about the rebuttal. Sorry for the delay and thanks for your patience

---

### Official Review · Reviewer_wsPE · 2025-11-02

**Soundness:** 3
**Presentation:** 3
**Contribution:** 3
**Rating:** 6
**Confidence:** 3

**Summary:**

The paper develops a rigorous theory of the logical expressiveness of topological neural networks (TNNs) operating on attributed combinatorial complexes (ACCs). It introduces:

- k-CCWL: a higher-order Weisfeiler–Leman-style test for combinatorial complexes that refines colors on k-tuples via a “double shift sequence,” aligning with message passing that aggregates via pairs across upper/lower neighborhoods.

- TCk: a new finite first-order logic fragment endowed with a pairwise counting quantifier designed to capture the pairwise aggregation semantics inherent to TNNs.

- Topological pebble game: a game-theoretic characterization mirroring TCk, but with rules adapted to pairwise placements/relations in complexes.

The central theoretical result is the exact equivalence among these three viewpoints.
This yields a clean expressiveness characterization akin to the classical WL–counting logic–pebble game triad for GNNs, and explains why expressivity increases with k for higher-order TNNs. The paper also proves a “broadcast anchor” property for uniform ACCs, establishing an identical-vs-disjoint global signature phenomenon that strengthens the isomorphism testing narrative.

**Strengths:**

- Originality: The paper introduces a novel pairwise counting quantifier and a tailored topological pebble game, both tailored to the unique mechanics of TNN message passing over combinatorial complexes. This goes beyond standard graph WL/FOC frameworks to a topological domain with upper/lower adjacency and boundary/co-boundary relations. The “double shift” construction and the logic–game–algorithm triad for TNNs are original and conceptually unifying.

- Quality: The theoretical development is systematic, starting from ACCs and neighbourhood systems, building k-CCWL from atomic types and multiset refinement contexts, and then matching these to TCk+2 and the pebble game through careful equivalence arguments. The appendices provide detailed proofs, and the broadcast anchor gadget for uniform ACCs is a nice technical device that clarifies the isomorphism testing behaviour.

- Clarity: The paper clearly distinguishes the four adjacency types and motivates why pairwise aggregation in TNNs necessitates pairwise counting in the logic. The stepwise definitions for atomic types, initialisation, refinement, and stabilisation are well structured, and the use of quantifier depth and alignment of refinement depth with logic/game rounds help readability.

- Significance: Establishing a precise logical expressiveness theory for TNNs is timely and valuable. It puts TNNs on similar theoretical footing to GNNs and higher-order WL variants, enabling principled reasoning about what architectures can or cannot decide. The equivalence result is likely to become a reference point for future work in topological deep learning and higher-order message passing.

**Weaknesses:**

- Scope restriction to uniform ACCs: Several key claims and the broadcast-anchor property depend on uniform ACCs. While understandable for technical control, it would strengthen the discussion of how results extend to non-uniform complexes and the minimal conditions needed.

- Complexity and practicality: The theoretical expressiveness results are strong, but practical guidance is limited. There’s little discussion of the computational complexity of k-CCWL on ACCs, its scaling with k and complex order, or efficient approximation methods. Concrete implications for designing more powerful TNN layers, such as beneficial pairwise aggregations or architectural motifs, could be elaborated. TNNs are often criticised for practicality due to complexity, so the increased complexity on higher orders may hinder the impact of the theoretical results.

- Empirical or constructive exemplars: Even small, instructive examples demonstrating where k-CCWL/TCk+2 distinguish structures that standard WL/GNNs can’t, especially in real TNN use-cases (e.g., hypergraphs, simplicial complexes in molecules), would increase accessibility and impact.

- Contextualisation vs. related theory: The connection to prior works on higher-order WL, counting logic, and pebble games is noted, but a more granular comparison to existing expressiveness results for cellular/cellular sheaf networks or recent “topological blind spots” analyses would better position the novelty. Making explicit how the new pairwise quantifier relates to prior graded/counting logics would help readers from finite model theory.

- Notation density: Some sections (e.g., Definition and refinement contexts with `D(t)_k(x)` and the double shift sequence) are notation-heavy. Additional figures or worked examples could reduce cognitive load.

**Questions:**

1.  Why do you need the broadcast-anchor? Does the isomophism test work without it?

2. Is k-CCWL guaranteed to converge? I couldn't find relevant info in the proof.

1. Beyond uniform ACCs: Which parts of the equivalence hinge critically on uniformity? Can the broadcast-anchor identical-vs-disjoint result be generalized with weaker conditions (e.g., local facet constraints)? If not, can you provide counterexamples?


2. Tightness of the k+2 overhead: The need for k+2 variables/pebbles arises from the double shift sequence. Is k+2 provably tight, or could specific subclasses of ACCs/TNNs be captured by TCk+1? A formal lower bound on variables needed would be informative.

3. Complexity bounds: What are the computational costs of k-CCWL (and its stabilization) on ACCs of order ρ and size |X|? Are there known polynomial bounds in k, ρ, and |X|? Can you propose practical approximations that preserve the expressiveness class for typical TNN architectures?

4. Design guidance for practitioners: Given the pairwise quantifier motivation, which concrete TNN aggregation patterns (e.g., pairwise interactions via NB/NC intersections) are minimally necessary to realize the TCk+2 power? Could you sketch an architecture template that is complete for TCk+2 under injective maps?

5. Illustrative case studies: Could you add small synthetic examples where k-CCWL separates complexes that 1-WL or even k-WL on graphs cannot? For instance, hypergraph configurations or simplicial complexes where upper vs. lower adjacencies are crucial.

6. Relations to sheaf/CW networks: How does k-CCWL/TCk+2 apply to CW networks (Bodnar et al., 2021) and cellular sheaf-based models? Are there natural adaptations of the atomic type to encode sheaf morphisms/twists, and would the pairwise counting extend?

7. Limits and blind spots: Are there graph/topological properties you can prove are still not definable in TCk+2 for fixed k (i.e., requiring unbounded k)? A short “limitations” subsection would sharpen the theoretical picture.

8. Anchors and robustness: The broadcast-anchor gadget is central. How robust is this construction to noise or attribute perturbations? If attributes are continuous and then discretized, does the identical-vs-disjoint property survive?

---

> ### Author Response · Authors · 2025-11-23
> **Part 1/3**
>
> Thank you very much for the thoughtful and constructive review. We are updating the PDF to incorporate the suggestions from all reviewers. Below, we provide point-by-point responses to your comments.
>
> > "Question 1. Why do you need the broadcast-anchor? Does the isomophism test work without it? "
>
> _Does the isomorphism test work without it?_ Yes! We note that broadcast-anchors allows us to define fresh colors to 0 and 1-rank cells so they can be distinguished from the original ones in two input CCs, A and B. To define a CC-isomorphism test between A* and B* (i.e., A and B with anchors), we must map the anchor in A* to the anchor in B*. Therefore, there exists a CC-isomorphism between A* and B* iff there is a CC-isomorphism between A and B. Thus, **adding broadcast-anchor cells does not affect whether A and B are CC-isomorphic**.
>
> _Why do we need a broadcast-anchor?_  The intuition is that the broadcast-anchor allows for capturing color differences between CC elements and propagating them to other cells. The broadcast-anchor is introduced to prove the _identical-vs-disjoint_ property for uniform ACCs (Proposition 3.1). This property is important for two reasons:
>
> - It is important for the proof of the equivalence between $k$-CCWL and TCLogic{k+2} on uniform ACCs (Theorem 5.4). In particular, it lets us connect equality of global stable colors of two ACCs with agreement vs. disagreement on all sentences of TCLogic{k+2} (i.e., formulas without free variables). Intuitively, this gives a clean structure-level perspective on the expressive power of $k$-CCWL and TCLogic{k+2} by eliminating the need to reason about valuations.
>
> - It also leads to an efficiency improvement for the $k$-CCWL isomorphism test. With the anchor and the identical-vs-disjoint property in place, we can run $k$-CCWL on each ACC once and then, after stabilization, inspect the color of a single global tuple in one ACC and only check whether this color occurs in the other ACC: if it does, then the two sets of global stable colors coincide, hence $k$-CCWL cannot distinguish the complexes; if it does not, then the sets are disjoint and $k$-CCWL does distinguish them. Without the anchor, even if one color appears in both complexes, we still have to search for all other colors (and their multiplicities). To see this, consider two graphs $A=K_2 \cup K_{1,3}$ and $B=K_2 \cup P_4$  --- here, $K_{n,m}$ is a bipartite graph with partitions of size $n$ and $m$, and $P_4$ is a path graph. Here, we have that their multisets of stable colors overlap but are different.
>
> > "Question 2. Is $k$-CCWL guaranteed to converge?"
>
> Yes, in our setting, $k$-CCWL is always guaranteed to converge, for the same reason as the classical $k$-WL algorithm on finite graphs. In all algorithmic parts of the paper (CCWL and $k$-CCWL), we consider finite ACCs, i.e., the underlying cell set $X$ is finite. We will make this finiteness assumption explicit in the revised version. For clarity, we have added Lemma B.1 in the Appendix, which shows that on any finite ACC the $k$-CCWL refinement sequence stabilizes and $\chi_k^{(\infty)}$ is well-defined.
>
> > "Question 3. Beyond uniform ACCs: Which parts of the equivalence hinge critically on uniformity? Can the broadcast-anchor identical-vs-disjoint result be generalized with weaker conditions (e.g., local facet constraints)?" (related to Weakness 1)
>
> We thank the reviewer for raising this point.
>
> We highlight that we only require uniformity for Proposition 3.1 (the disjoint-vs-disjoint property) and Theorem 5.4 (Structure-level isomorphism tests). Therefore, many key results do not rely on that assumption, such as the core logic/game correspondence, the translation between $k$-CCWL colors and $\mathrm{TC}_{k+2}$ formulas, and the topological $(k{+}2)$-pebble game.
>
> **Regarding a weaker condition.** Indeed, we can relax the uniformity condition. Uniformity was used to ensure a “flow of information’’ between every cell and the fresh rank-$0$ anchor cell. Technically, uniformity ensures that, for every cell $c$, there exists a finite sequence
>
> $$
> c = x_0, x_1, x_2, \dots, x_m
> $$
>
> such that each $x_i$ lies in one of the CCWL neighborhoods of $x_{i+1}$ (boundary, coboundary, lower or upper), and the sequence ends at a $0$-rank cell. Along such a path, CCWL refinement can propagate information from $c$ to the broadcast-anchor and back.
>
> Based on this, in our revised manuscript, we provide an alternative definition of uniformity in an ACC (see Background in the updated PDF) such that for each cell, there should be a sequence of adjacent cells that ends in a $0$-rank cell. Importantly, this aligns with the sufficient and necessary conditions we need to prove our results.

---

> ### Author Response · Authors · 2025-11-23
> **Part 2/3**
>
> > "Question 4. Tightness of the $k+2$ overhead: Is $k+2$ provably tight, or could specific subclasses of ACCs/TNNs be captured by \TCLogic{k+1}? A formal lower bound on variables needed would be informative."
>
> _Why is the "+2 variabale" gap tight?_ Thanks for your question. In the revised version, we now show that the "+2 variable" gap is not only sufficient but also necessary. In Section D, we first define 1-dimensional ACC of a graph, denoted as $A = G^{\mathrm{CC}}$. Then, we prove that $k$-CCWL on $G^{\mathrm{CC}}$ has the same expressivity power as $k$-WL on $G$. Basically, we show:
>
> $$
> \text{$k$-WL distinguishes $G$ and $H$} \iff \text{$k$-CCWL distinguishes $G^{\mathrm{CC}}$ and $H^{\mathrm{CC}}$}.
> $$
>
> We can now use a result by Cai et al. (1992), which shows that for any $k \geq 1$, there is a pair of graphs $G_k$ and $H_k$ such that $k$-WL can distinguish, but $(k-1)$-WL cannot. Therefore, we can use $G_k^{\mathrm{CC}}$ and $H_k^{\mathrm{CC}}$  to obtain the separation we are looking for, i.e., that **$k$-CCWL \ $TC_{k+2}$ is strictly more powerful than $(k-1)$-CCWL** \ $TC_{k+1}$. This is now established in Theorem D.3 of the revised manuscript.
>
> > "Question 5. What are the computational costs of $k$-CCWL (and its stabilization) on ACCs of order $\rho$ and size $|X|$? Are there known polynomial bounds in k, $\rho$, and $|X|$?" [Related Weakness 2]
>
> We thank the reviewer for this suggestion. In the revised version (Section B), we now state these bounds explicitly along with a proof.
>
> Let $X$ be the set of cells, $|X|=n$, and let $k$ be fixed.
>
> **Per-round cost.** By definition, in each refinement round $t \to t{+}1$ we iterate over all $k$-tuples $x \in X^k$ and, for each $x$, range over all $(\alpha,\beta) \in X^2$ in the double-shift. A straightforward implementation, therefore, has a worst-case cost
>
> $$
> O\bigl(|X|^k \cdot |X|^2\bigr) = O\bigl(n^{k+2}\bigr)
> $$
> per round, which we now make explicit in the paper.
>
> **Rounds to stabilization.** As in standard $k$-WL, each non-trivial round strictly increases the number of distinct colors on $X^k$, and there are at most $|X|^k$ tuples. Hence, the number of rounds to stabilization satisfies
>
> $$
> t_{stab} < |X|^k.
> $$
>
> Putting this together gives a worst-case bound
> $$
>     \mathrm{TIME}(k\text{-CCWL})
>     \leq O\bigl(T \cdot n^{k+2}\bigr)
>     \subseteq O\bigl(n^{2k+2}\bigr),
> $$
>
> with space complexity $O(n^k)$ to store tuple colors. In addition, we need the space for the ACC itself (incidence, ranks, attributes), which is at most quadratic in $n$ for fixed rank. Thus, the overall space complexity is $O(n^k)$ for fixed $k$.
>
> > "Question 6. Which concrete TNN aggregation patterns (e.g., pairwise interactions via NB/NC intersections) are minimally necessary to realize the TC{k+2} power? Could you sketch an architecture template that is complete for TC{k+2} under injective maps?" [Related to Weakness 2]
>
> Thanks for your question. In graph ML, injective aggregation and update functions are typically implemented using MLP-based updates together with DeepSets-style aggregators, as in Graph Isomorphism Networks (GIN). Cell Isomorphism Networks (CINs) [Bodnar et al., 2021] follow exactly this design to obtain injective layers; as a consequence, when restricted to cell complexes, their expressive power matches that of 1-CCWL.
>
> > "Question 7. Illustrative case studies: Could you add small synthetic examples where $k$-CCWL separates complexes that $1$-WL or even $k$-WL on graphs cannot?" [Related to Weaknesses 3 and 5]
>
> As suggested, in the revised version, we have added Example 5.5, which presents a pair of ACCs that are indistinguishable by 1-WL, 1-CCWL, and Simplicial WL but can be separated by 2-CCWL. These examples rely on the fact that 1-WL and 1-CCWL have identical expressive power on 1-dimensional ACCs (Theorem D.2). We demonstrate separability in two ways: i) by explicitly providing a formula $\phi \in TC_4$ that distinguishes the pair; and ii) by exhibiting a winning strategy for Player I in the topological 4-pebble game corresponding to $\phi$.
>
> > "Relations to sheaf/CW networks: How does $k$-CCWL/\TC{k+2} apply to CW networks (Bodnar et al., 2021) and cellular sheaf-based models?" [Related to Weakness 4]
>
> Thanks for your question. We note that 1-CCWL characterizes the expressive power of CW Networks [Bodnar et al., 2021] when these models employ injective aggregation and update functions. In particular, Bodnar et al. (2021) introduce Cell Isomorphism Network (CIN) (see their Appendix E3), which has expressive power equivalent to that of 1-CCWL when restricted to cell complexes.
>
> Regarding sheaf-based models, to the best of our knowledge, there is no work that applies sheaves to higher-order domains (e.g., cell complexes). For instance, Neural Sheaf Diffusion [Bodnar et al., 2021] only exploits cellular sheaves over graphs. Extending and evaluating sheaf-based constructions for higher-order domains is an interesting research direction.

---

> ### Author Response · Authors · 2025-11-23
> **Part 3/3**
>
> > "Question 9. Limits and blind spots: Are there graph/topological properties you can prove are still not definable in $TC_\{k+2\}$ for fixed k (i.e., requiring unbounded k)? A short “limitations” subsection would sharpen the theoretical picture."
>
> We thank the reviewer for raising this important point. In the revised version, we now add a short ``Limitations'' paragraph to make blind spots explicit.
>
> In particular, for every fixed k, $TC_k$ is still a finite-variable counting logic. Therefore, it inherits the usual locality limitations of $k$-WL. Moreover, many global graph/topological properties provably need unbounded k, and it is not possible to define them in $TC_k$. There are some examples, such as:
> - Global counting and majority-level properties: "More than half of the cells have the property $P$".
> - Standard global graph properties that are impossible for k-WL to catch for all fixed k: Connectivity, existence of a Hamiltonian path, or the existence of a perfect matching.
> - Global topological invariants that depend on the overall pattern of ``holes'' and voids across the entire complex, where the difference between two complexes may only appear in very large cycles that cannot be detected from any fixed-radius local neighborhood.
>
> > "Question 10. Anchors and robustness: The broadcast-anchor gadget is central. How robust is this construction to noise or attribute perturbations? If attributes are continuous and then discretized, does the identical-vs-disjoint property survive?"
>
> Our broadcast-anchor is defined purely on the structure of the ACC. The identical-vs-disjoint property in Proposition 3.1 depends only on the structure of the ACC and whether two attributes are equal or not, and it does not depend on whether the attribute function is discrete, continuous, or obtained by discretizing continuous features.
>
> Although the reviewer finds the gadget to be central, we only use it to prove Theorem C.14. There, the broadcast-anchor lets us remove the partial valuations on uniform ACCs. Note that our main tuple-wise expressivity result in Corollary 5.3 does not rely on broadcast-anchor and already holds without the gadget.
>
> References:
>
> [Bodnar et al., 2021] Weisfeiler and Lehman Go Cellular: CW Networks. NeurIPS, 2021.
>
> [Cai et al., 1992] An Optimal Lower Bound on the Number of Variables for Graph Identification. Combinatorica, 1992.
>
> ---
>
> Thank you again for the constructive feedback. We hope our response has fully addressed your concerns, and we would greatly appreciate your strengthened support for this work. If you have any further questions or suggestions, please let us know.

---

### Author Response · Authors · 2025-12-03
**General Response**

We sincerely thank you and all four reviewers for the careful evaluation, constructive questions, and helpful discussion. We are encouraged that reviewers specified all of their concerns were resolved and the discussion substantially strengthened the paper (Reviewers bQiY and 6rEA).

We appreciate the reviewers’ positive feedback (Reviewers bQiY, 6rEA and wsPE). They agree on the significance and strength of our contributions and note that the revisions have improved the clarity and depth of the paper (bQiY and 6rEA). They emphasized the novelty and potential impact of our approach, and highlighted the stronger theoretical analysis and clearer figures. We believe we fully addressed the earlier concerns and further strengthened the work, and we are hopeful that this will be reflected in the final evaluation.The comments focused on clarity and exposition rather than on the substance of the results.

Below we summarize how we addressed the main concerns:

### **Accessibility and intuition (Reviewers qRd9, bQiY).**
We have significantly revised the introductory sections to make the paper easier to read and more intuitive:

-   We added a paragraph, “Why pairwise counting on ACCs?”, that explains in simple terms why ACC relations are mediated by third cells and why boundary–coboundary flows are naturally pairwise.

-   We introduced several new figures (Figs. 2–5) to visually explain the isomorphism test, logical distinguishability, and the pebble game, making these ideas easier to follow.

-   We added a short “Limitations” paragraph that intuitively describes which properties are not definable for fixed $k$ because of locality.

### **Tightness of the (k+2) overhead and strict hierarchy (Reviewers 6rEA, wsPE).**

-   Appendix D now shows that the $((k{+}2))$-variable overhead is necessary: on 1D ACCs obtained from graphs, $(k)$-CCWL has exactly the expressive power of $(k)$-WL, and together with the Cai–Fürer–Immerman lower bound this yields Theorem D.3.

-   Example 5.5 and Lemma D.1 now give concrete pairs that separate $(k)$-CCWL from $((k{-}1))$-CCWL, and Theorem 3.2 generalizes this to all $(k)$.

-   In response to follow-up questions, we also explain how these separations can be “thickened’’ to genuinely higher-dimensional ACCs by replacing each node with a higher-dimensional gadget.

### **Broadcast anchor and uniform ACCs (Reviewers wsPE, bQiY, 6rEA).**

-   We clarify that the broadcast anchor is used only for the global, structure-level result (Theorem 5.4); the main tuple-level equivalence (Corollary 5.3) holds without it.

-   We relaxed uniformity in terms of a simple incidence-connectivity condition (every cell can reach a $0$-rank cell through defined adjacencies), which is exactly what is needed for information flow to and from the anchor.

-   We add an explicit non-uniform counterexample (Example B.8) where the identical-vs-disjoint property fails, showing that some structural assumption is indeed **helpful**.

### **Complexity, convergence, and practice (Reviewers wsPE, 6rEA, bQiY).**

-   Appendix B now gives explicit bounds: for fixed $(k)$, one round of $(k)$-CCWL on a finite ACC with $(|X|)$ cells costs $(O(|X|^{k+2}))$, and stabilization occurs in at most $(|X|^k)$ non-trivial rounds.

-   We implemented $(1)$-CCWL on ACCs constructed from the Basic and Regular BREC benchmarks; on all pairs distinguishable by $(1)$-CCWL, separation occurred in fewer than $5$ iterations.

### **What ($\mathrm{TC}_k$) / $(k)$-CCWL can and cannot express; relation to Eitan et al. (Reviewer bQiY).**

-   We clarify the standard “for fixed $(k)$” notion: there is no single $(\mathrm{TC}_k)$ sentence that defines connectivity, etc. on all finite ACCs. The locality radius ($r(k,q)$) (for $(k)$ variables and depth $(q)$) bounds what these logics can see, ruling out global connectivity, and certain homology properties.

-   We add a discussion of Eitan et al.’s covering-based “blindspots”. Their pairs (e.g., cylinder vs.\ Möbius strip with long fundamental cycles) can be made arbitrarily large while remaining locally indistinguishable. This matches the usual view that such non-local invariants lie outside fixed-$(k)$ counting logics, and hence outside fixed-$(k)$ CCWL.


### **Scope of architectures (Reviewers bQiY, 6rEA).**

-   We clarify that $(1)$-CCWL matches the expressive power of CW Networks with injective aggregators (e.g., CIN) and upper-bounds many other ACC-based TNNs, including simplicial message-passing networks and UniGNN-style models.

-   We discuss how we can extend the proposed architecture to sheaf-based updates, persistence-guided lifting, and attention-based TNNs.

Overall, these changes tighten the main results, clarify limitations and global blindspots, and better connect the theory to existing and emerging TNN architectures. We hope they will be helpful in your final assessment, and we are grateful for the time you have devoted to our submission.

---

### Meta-Review · Area_Chair_1iK3 · 2026-01-05

**Summary:**

This is a mathematically technical paper that establishes a correspondence between the expressive power of topological neural networks, a topological counting logic introduced in this paper, and a topological extension of the pebble game.

The original scores were (6, 2, 6, 4), but there was a fair amount of discussion between the authors and the reviewers that shows that the reviewers value the contributions of the paper, and were overall satisfied with the authors' responses.

The most negative reviewer mainly criticized the paper's suitability for ICLR, saying that the paper is not accessible to the ICLR audience. Other reviewers also commented on the paper's presentation being hard to follow. The authors responded by expanding the introduction and mentioning similar papers that were previously published in ICLR. The reviewer wasn't able to respond whether they were satisfied with the answers provided.

I agree with reviewer qRd9 that the paper is very technical. However, given the positive reviews and the positive exchanges between the authors and the reviewers, I lean towards accepting the paper.

**Reviewer Concerns:**

The reviewers concerns are fairly summarized in the authors' global response.

**Reviewer Scores:**

The authors gave thorough responses to all the reviewers. The reviewers who responded were satisfied with the answers provided. Reviewer 6rEA did not respond, but they had technical questions/weaknesses that I believe were properly addressed.
Reviewer qRd9 did not respond either. I am not sure if their criticisms were resolved.

---

### Decision · Program_Chairs · 2026-01-26

Accept (Poster)